# Dance of SNN and ANN: Solving binding problem by combining spike timing and reconstructive attention

**Hao Zheng**[†]    **Hui Lin**[†]    **Rong Zhao**    **Luping Shi**[*]
Department of Precision Instrument
Center for Brain-Inspired Computing Research
Tsinghua University, Beijing 100084, China
{zheng-h17,linhui21}@mails.tsinghua.edu.cn
{r_zhao,lpshi}@tsinghua.edu.cn

## Abstract

The binding problem is one of the fundamental challenges that prevent the artificial neural network (ANNs) from a compositional understanding of the world like human perception, because disentangled and distributed representations of generative factors can interfere and lead to ambiguity when complex data with multiple objects are presented. In this paper, we propose a brain-inspired hybrid neural network (HNN) that introduces temporal binding theory originated from neuroscience into ANNs by integrating spike timing dynamics (via spiking neural networks, SNNs) with reconstructive attention (by ANNs). Spike timing provides an additional dimension for grouping, while reconstructive feedback coordinates the spikes into temporal coherent states. Through iterative interaction of ANN and SNN, the model continuously binds multiple objects at alternative synchronous firing times in the SNN coding space. The effectiveness of the model is evaluated on synthetic datasets of binary images. By visualization and analysis, we demonstrate that the binding is explainable, soft, flexible, and hierarchical. Notably, the model is trained on single object datasets without explicit supervision on grouping, but successfully binds multiple objects on test datasets, showing its compositional generalization capability. Further results show its binding ability in dynamic situations.

## 1   Introduction

Learning disentangled and distributed representation of generative factors of the world is believed to benefit compositional generalization, because those invariant features can be reused as symbols to build exponentially larger amounts of objects with higher complexity [1, 2, 3, 4]. However, disentangled representation can suffer from a fundamental difficulty when the inputs are composed of multiple objects. Since the features are disentangled and invariant to specific objects, multiple coexist feature representations interfere and lead to superposition catastrophe (Fig.1a). This is conceptualized as the binding problem by von der Malsburg (1981)[5, 6] and extensively debated in neuroscience and psychology literature [7]. Recently, the binding problem is also documented as a fundamental reason behind various inabilities of artificial neural networks (ANNs) like data inefficiency and limited transfer capability [3].

Segregating and representing symbol-like entities from raw data is challenging and often requires supervision [8, 9, 10, 11, 12, 13] or task-specific design[14, 15, 16, 17]. Thus, the problem is usually ignored by assuming there is only a single object of interest or mediated by using static localized binding [18, 19] at the cost of expensive computational demand. All the above methods in ANNs cannot fundamentally solve the binding problem. In contrast, slot models[3, 20, 21, 22, 23, 24, 25,

---

[*]Corresponding author, [†] Contribute equally

36th Conference on Neural Information Processing Systems (NeurIPS 2022).

26, 27, 28, 29, 30, 31, 32] in ANNs seems to provide a promising approach, which uses various types of computational slots to inform the object label. However, current slot models are not so flexible and dynamic as human perception partly because they require pre-defined, fixed, discrete number of slots to place a hard label to each object. Besides, the slot is usually entangled with certain dimension of features[33] and the set of features often need to be duplicated for each slot[25].

To overcome this challenge, we brought ideas from neuroscience. In neuroscience, temporal coordination is an alternative solution to support the binding in a human brain. The solution is based on spike timing correlation in a network of spiking neurons (SNNs), referred to as temporal binding theory[5] or correlation brain theory[34]. The theory assumes a hybrid code. Each neuron has its receptive field of firing rate tuned to certain features. Its spike timing can additionally be modulated to express grouping information, so that assemblies encoding features belonging to the same object fire coherently in time (Fig.1b). Grouping in time dimension has five important advantages. First, object representation based on additional dimension of time will naturally be in a common format[3], essential for systematic generalization to unseen situations or more objects. Second, unlike slot models, a single set of features are shared among all object representations. Third, the grouping information is usually continuous thus may help to encode uncertainty about the grouping. Fourth, unlike supervised method, temporal binding is an unsupervised process, consistent with the multi-stability property of the grouping. Fifth, the number of groups is not needed to be explicitly pre-defined, but self-organized dynamically according to temporal coherence. Despite the virtue of temporal binding, however, how to realize spike timing correlation in various computational demanding situation remains an open and challenging problem. As far as we know, the temporal binding ability of traditional SNN models is only tested on very simple stimulus patterns[35, 36, 37, 38, 39, 40], not as complex as datasets in the field of ANNs. Besides, since the concept of spike and spike timing is incompatible with the ANN framework, very few studies have incorporated the promising SNN solution into ANNs[3].

In this work, we introduce a hybrid neural network architecture (HNN), combining spike timing code within the SNN module with delayed reconstructive feedback from the ANN module (Fig.1c). First, the augmented spike coding space exploits time dimension to softly group multiple objects with a natural common format and a single set of features. Second, the denoising autoencoder (DAE)[41, 42] feedback important modulatory signals to coordinate the spike timing into coherent states. Third, the spiking dynamics induces temporal competition among multiple objects to group them at separate timings. Taken together, SNNs expand the representation space of ANNs, and ANNs provide the complex computations necessary for flexible binding. Two parts interact iteratively. Thus, we name the resulting architecture as **DASBE** (short for " **D**ance of **A**NN and **S**NN " **B**inding n**E**twork). We also provide an informal proof of the convergence of the model.

Evaluating binding is intrinsically challenging due to multi-stability [3]: scenes can be ambiguous and afford multiple grouping states, some of which are even unknown. Thus, we combine quantitative and qualitative analysis for evaluation. The former is mainly to provide a partial measure of the effectiveness based on pre-defined ground truth while the latter provide important additional proof of the binding. Specifically, (1) We quantitatively evaluate the binding in **DASBE** by the mean adjusted mutual information (AMI) score [43] between clustering result of spiking patterns and ground truth segmentation. The **DASBE** achieves more than 0.50 on all dataset and 0.80 on 3 out of 5 datasets, demonstrating its grouping capability on different types of stimuli. (2) We further visualize the spike representation and analyze the properties of the representation on the Shape dataset, showing that the grouping is continuous and based on spike synchrony, which is measured based on Victor-Purpura metric [44] of spiking patterns. Interestingly, the bound assemblies are seated on the peak of emerged gamma-like oscillations, analogous to the brain[45]. (3) Besides binding in pixel-level features, we also show that features of hierarchical levels can be bound in **DASBE**. (4) Lastly, it is demonstrated that **DASBE** can even track multiple moving objects and detect pop-up objects in the dynamic situation without explicit supervision. Taken together, we hope integrating ANNs and SNNs properly can provide new possibilities to solve binding problem in a brain-inspired manner.

## 2 Contribution

(1) We introduce the spiking dynamics into ANNs to solve the binding problem, so that the binding is unsupervised, soft, in a common format and keeps a single set of features. (2) We introduce reconstructive attention from trainable DAE into SNNs so that the capability of temporal binding is enlarged beyond that of traditional fixed connection models (3) We develop neuroscience-inspired

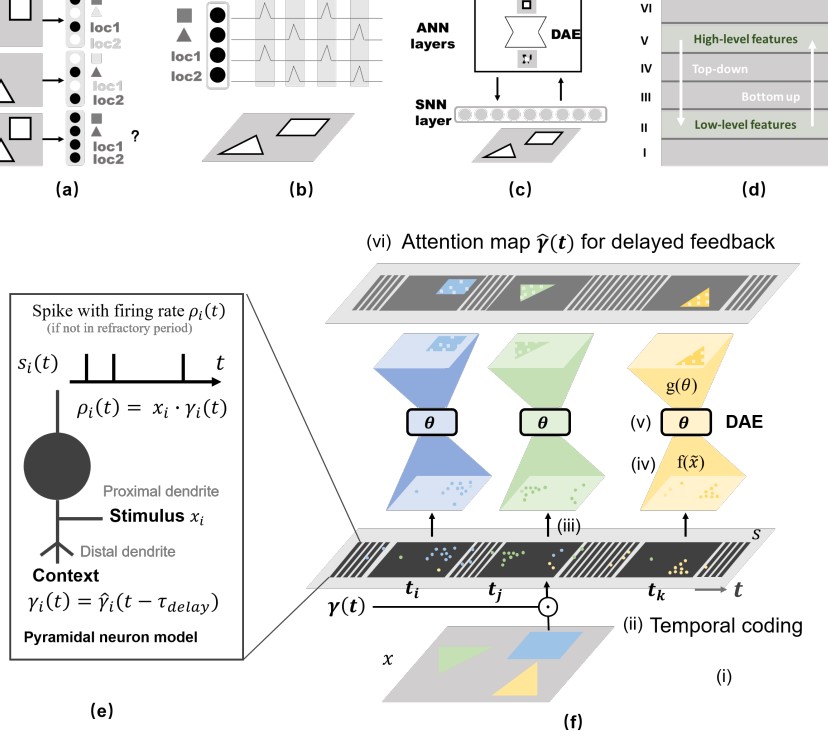

Figure 1:  (a) The binding problem.  (b) Temporal binding theory.  (c) Overview of **DASBE** architecture. (d) Biological relevance. (e) Single neuron model in **DASBE**. (f) Detailed information flow of **DASBE**. (i) input image (ii) SNN coding space (iii) coincidence detector (iv) Encoder (v) latent space (vi) attention map given by decoder. Feedback flow is not shown for clarity

analysis method to evaluate and visualize the property of spiking representation in a single level or hierarchical levels, in static or dynamic situations. Gamma-like oscillation is observed.

## 3   Model

How can temporal binding be incorporated in ANNs to enhance the compositional representation ability of ANNs? We provide an answer by augmenting an ANN with a spike coding space (by SNNs), so that the time dimension can possibly be used to group different objects. On the contrary, how can each spiking neuron "know" its grouping companions to fire with? We provide an answer by a top-down modulation from a denoising autoencoder (DAE). Lastly, how can different object be grouped at separate timings? We provide a minimal answer with neural refractory dynamics to induce a mandatory temporal competition.

The model is composed of four parts: a spike coding space (SCS) of low-level features, an ANN encoder, a latent space of higher-level features, and an ANN decoder (Fig.1c). Four parts are connected end to end, so that the ANNs and SNNs interact dynamically in an iterative way (with delay). The biological interpretation of each part is shown in Fig.1d.

### 3.1   Encode binary stimulus by topographical mapping

Inspired by topographical mapping in primary cortical layers [46], the dimension of spike coding space (SCS) is set to be of the same as the input image so that pixels in the image have one-to-one correspondence to neurons in the SCS (Fig.1f(i)). The encoding is binary. $x \in \{0, 1\}^N$ and $s \in \{0, 1\}^N$ where $x$ is the image stimulus and $s$ is the spiking response of neurons in SCS, $N$ is the size of image.

## 3.2 Spike coding space (SCS)

The single neuron model in SCS (Fig.1f(ii)) is the two-compartment model of pyramidal cells [47](Fig.1e). The proximal dendrite [48] of neuron i receives binary driving signals from external stimuli $x_i(t)$. The distal dendrite [49] of neuron i receives smooth modulatory signals $\gamma_i(t)$ from delayed top-down feedback.

Since the distal and proximal current usually interact in a non-linear manner in vivo [47, 50], the product of two terms determines the firing rate of the active neurons (not in refractory period).

$$\rho_i(t) = x_i \cdot \gamma_i(t), \tag{1}$$

where $i$ is the neuron index; $\rho_i(t)$ is the firing rate. If the neuron generates a spike, $s_i = 1$, otherwise $s_i = 0$. After firing a spike, the neuron undergoes a period of refraction $\tau_{rfr} \in \mathbb{N}$. In total,

$$s_i(t) = \texttt{Ber}(\rho_i(t)), i \in A. \tag{2}$$

where $A$ is the set of active neurons that are not in refractory periods $\tau_{rfr}$, and $\texttt{Ber}()$ is a sampling step according to Bernoulli distribution with mean $\rho_i(t)$. Thus, the firing event $s_i = 1$ reflects the integrative effect of external stimuli, top-down modulation from ANNs and spiking dynamics in SNNs.

## 3.3 Bottom-up processing by the encoder of DAE with coincidence detectors

The spiking events in SCS are readout by coincidence detectors (CD, Fig.1f(iii)) [51], which only response to a very limited temporal window $\tau_w < 3$ with stochastic decay $\alpha_{(i,p)}^\tau \in \{0, 1\}$. The CD result is further fed to the autoencoder.

$$\tilde{x}_i(t) = \Theta(\sum_{\tau=0}^{\tau_w} \alpha_{i,p}^\tau \cdot s_i(t - \tau) - V_{th}). \tag{3}$$

$\Theta$ is a step function with threshold $V_{th}$, standing for the activation function of the CD; stochastic decay $\alpha_{i,p}^\tau \sim \texttt{Ber}(p^\tau), p < 1$. Then the DAE encoder extracts out higher-level features of the objects in the images as the latent representation $\theta(t) = T(f(\tilde{x}(t)))$. Here, $f$ is the encoder and $\theta$ is the latent vector (Fig.1f(iv,v)). T is an additional spike sampling operation if latent representation is binary, otherwise it is just an identity map.

## 3.4 Delayed modulation of attention map given by the decoder of DAE

The output of DAE decoder is fed back to the pyramidal neuron in SCS through distal dendrite as top-down modulation, but with a delay period.

$$\gamma(t) = g(\theta(t - \tau_{delay})) = \hat{\gamma}(t - \tau_{delay}). \tag{4}$$

$g$ is the decoder of DAE. $\hat{\gamma}$ refers to the attention map given by the DAE (Fig.1f(vi)) and $\tau_{delay} \in \mathbb{N}$ refers to the delay period.

For clarity, the overall binding process is shown in Algorithm 1. We identify three key properties of the **DASBE** binding process: First, the positive feed-back structure of DAE, which is termed as the "folded" DAE here, embeds each single object as its attractive fixed points.Those fixed points act as associative memories of symbol-like entities. Second, The superposed state of multiple objects is not such a fixed point. Thus, superposition is not a stable representation of "folded DAE". However, "folded" DAE may focus on one object and ignore others. Third, if augmenting "folded" DAE with refractory dynamics of SNN, and delayed feedback, then each object becomes only transient states due to temporal competition. However, alternations of objects in turn is a possible stable trajectory. More formally,

**Proposition 1.** Assume the mapping $f_{DAE}$: $R^d \rightarrow R^d$ has learned to reconstruct vectors of single objects $\{x_n\}$, $x_n \in \{0, 1\}^d$ perfectly. Then, (1) (Fixed point) $\forall x_n \in \{x_n\}$, $x_n = f_{DAE}(x_n)$ and

---

**Algorithm 1** Temporal binding process in **DASBE**. The input $x$ is a binary vector of dimension $D_{image}$. The attention map (`attn`) is a real-valued vector of dimension $(T + \tau_{delay}) \times D_{image}$. $T$ is the length of binding process. We initialize `attn`$[t]$, $t \in [-\tau_{delay}, 0]$ as independent positive samples (`attn` $> 0$) from standard Gaussian distribution $\mathcal{N}(0, I)$. Refractory variable `rfr` is initialized as $0$. $\tau_{delay}$, $\tau_w$, $\tau_{rfr}$ are time-scale hyper-parameters. In our experiments, we set $\tau_{delay} \in [20, 60]$, $\tau_w = 3$, $\tau_{rfr} \in [0, 15]$, $T \approx 1000$. The hyper-parameters for CD: $V_{th} = 1$, $p = 0.5$

---

1: **Input**: `inputs` $\in \{0, 1\}^{D_{image}}$, `attn`$[-\tau_{delay} : 0] \sim |\mathcal{N}(0, I)| \in \mathbb{R}^{\tau_{delay} \times D_{image}}$
2: **Layer params**: $\text{MLP}_f$, $\text{MLP}_g$: encoder and decoder of DAE.
3: **Other layer operations**: `Norm`; `Softmax`; `CD`; `Ber`, `T`: Bernoulli sampling operation for binary latent space or identity map for real-valued latent space.
4:    `attn`$[-\tau_{delay} : 0] = \text{Norm}(\text{attn}[-\tau_{delay} : 0])$;      # normalization of initial attention map
5:    **for** $t = 0 \ldots T$
6:       `context` $= \text{attn}[t - \tau_{delay}]$;
7:       `firing_rate` $= \text{inputs} \odot \text{context} \odot (\text{rfr} == 0)$;      # $\odot$: element_wise product
8:       `spike`$[t] \sim \text{Ber}(\text{firing\_rate})$;   # Ber: sampling according to Bernoulli distribution
9:       `rfr`$+= \text{spike}[t] \cdot \tau_{rfr}$;      # set spiking neuron into refractory period
10:     `rfr` $= \max(\text{rfr} - 1, 0)$;      # update refractory variable
11:     `input2dae` $= \text{CD}(\text{spike}[t - \tau_w : t])$;      # CD: coincidence detector
12:     `encode` $= \text{Softmax}(\text{MLP}_f(\text{input2dae}))$;
13:     `latent` $= \text{T}(\text{encode})$;      # latent space can be real-valued or binary
14:     `attn`$[t] = \text{Softmax}(\text{MLP}_g(\text{latent}))$;
15:   **return** `spike`

---

(2) (Attractive) Under dynamics $x(t + 1) = f_{DAE}(x(t)) : \exists \delta > 0, \alpha \in (0, 1)$, s.t., $\forall \epsilon \in \{0, 1\}^d$, s.t. $\|\epsilon\| \leq \delta$, then, $\|f_{DAE}(x_n - \epsilon) - x_n\| < \alpha \cdot \|(x_n - \epsilon) - x_n\|$, "$-$" is constraint in the binary space of $x_n$

**Proposition 2.** $\exists x' = \sum_{i=1}^{K} x_{n_i}$ , $\|f_{DAE}(x') - x'\| > 0$ (superposed state is not a fixed point), "$\sum$" is constraint in the binary space

**Proposition 3.** Under delayed dynamics: $x(t + 1) = f_{DAE}(x(t - \tau_{delay}))$, "$x(t) \equiv x_n, t \in [0, T]$ " is a possible attractive solution.

**Proposition 4.** Taking refractory dynamics and delayed feedback together: $x(t + 1) = f_{DAE}(\text{Ber}(x(t - \tau_{delay}) \odot I(rfr = 0))$ , then "$x(t) \equiv x_n$ , $t \in [0, T]$" is not a solution. Given $x' = \sum_{i=1}^{K} x_{n_i}$ , $\exists x_{tra} : [0, T] \to \{0, 1\}^d$ , $x_{tra}(t) \in \{x_{n_i}\} \cup \{\mathbf{0}\}$ is a possible stable trajectory. ($I(x)$ is an indicator function)

The proof is in the Appendix. Note that the propositions are idealized and therefore may deviate from situations in practice. However, the feasibility of above properties is very likely to hold as can be seen in the experiments (Also see Appendix), which heuristically imply the binding ability of the model.

## 4 Experiment and results

### 4.1 Experiment setup

We test the binding performance on benchmark datasets[2], including Bars, Shapes, Corners, Multi-MNIST and MNIST+Shape. All these datasets are synthetic for convenience of evaluation and analysis. Further visualization are only analyzed in Shapes dataset for clarity, while results on other datasets can be found in Appendix.

Table 1: AMI score (%)

| Dataset | **DASBE** | Folded DAE | PCNN |
|---|---|---|---|
| Bars | **96.5** $\pm$ 0.03 | 9.6$\pm$0.04 | 9.3$\pm$0.04 |
| Shapes | **90.2** $\pm$ 0.07 | 47.5$\pm$0.04 | 44.7$\pm$0.15 |
| Corners | **80.6** $\pm$ 0.08 | 54.2$\pm$0.15 | 28.5$\pm$0.08 |
| Multi-MNIST | 55.5 $\pm$ 0.06 | 43.0$\pm$ 0.08 | 23.0 $\pm$ 0.11 |
| MNIST+Shape | 53.3 $\pm$ 0.26 | 38.3$\pm$0.03 | 12.9$\pm$0.08 |

## 4.2 Results

### 4.2.1 Quantitative evaluation of binding

We evaluate the **DASBE** binding performance on a list of benchmarked artificially generated datasets consisting of binary images of different complexity. For each dataset, a DAE is trained to remove the salt&pepper noise on images of single objects. The best trained DAE is used for binding on images containing multiple objects. The binding quality is evaluated based on ground truth object identity.

After the simulation, we use the K-means clustering to cluster spiking neurons into different groups according to their spike train pattern. Since the ground-truth segmentation is available for generated datasets, we compare the clustering result and ground-truth by adjusted mutual information (AMI) [43] as a measure for temporal feature binding performance. The AMI score is 1 for perfect grouping, and 0 for chance level. Although K-means uses Euclidean distance to compare each spike train with an averaged firing density, thus, in principle, do not explicitly evaluate timing difference, it will be shown that the clustering is actually based on spike timing code instead of firing rate difference (See Sec4.2.3 or Fig.3). The model achieves more than **0.50** on all datasets and **0.80** on 3 out of 5 datasets (5 random seeds, 6000 samples), implying their successful binding ability (Table 1).

To show the synergistic effect of combing ANN and SNN part, we reduce the hybrid **DASBE** into SNN (PCNN) and ANN ("folded" DAE) version as baseline. Pulse-coupled neural network (PCNN) is a traditional temporal binding model in neuroscience [52, 53, 54], and can be considered as replacing the trainable ANN(DAE) part by the fixed recurrent connections. Since original PCNN binds objects based on connectivity only, it gets into trouble when objects overlap (Bars, MNIST+Shape) or when unconnected parts compose a whole (Corners). The "folded" DAE (as defined in Proposition 3 and Appendix) can be considered as removing the spiking dynamics or replacing the SNN coding space by a non-spiking layer ($\tilde{x}_i(t) = x_i \cdot \gamma_i(t)$, where $\tilde{x}(t)$ is the non-spiking input to the DAE module). As shown in Proposition 3, "folded" DAE may focus too much on one object and ignore others. As a result, both control models perform much worse than the **DASBE** (Table 1).

### 4.2.2 Qualitative analysis

Visualization of spike timing pattern in SCS on Shapes dataset is shown in Fig.2 b.c.d. The randomly selected image consists of a square and two triangles located at different positions. Starting from random scattered firings, feature neurons responsive to the same object gradually synchronize themselves and keep out of phase to that of other objects. Object-related assemblies are seated on the peak of emergent gamma-like oscillatory population activities[45] (Fig.2 d). More details can be found in Appendix.

### 4.2.3 Evaluating synchrony, coding scheme and convergence

To get more insights of spiking code, we develop a method to evaluate synchrony level. Here, we define synchrony as a measure of temporal coherence of spike firing in each group. Such measure requires two things. (1) a coherence measure of clusters and (2) a metric explicitly dependent on spike timing difference. In this spirit, we define synchrony score as the silhouette score [55] of clusters based on Victor-Purpura metric [44] of spike trains. The Victor-Purpura metric is non Euclidean and has a time-scale parameter $q$. At its extreme, the metric explicitly measures the spiking rate when $q \to 0$ and spiking synchrony when $q \to +\infty$ (see Appendix). We define synchrony score and rate score by large and small $q$ value. It can be observed that, along the temporal binding process on Shapes, synchrony score is increased from 0.2 to 0.9, indicating the neurons converge to an object-dependent synchronous state from random guess (Fig.2a). In contrast, rate score remains

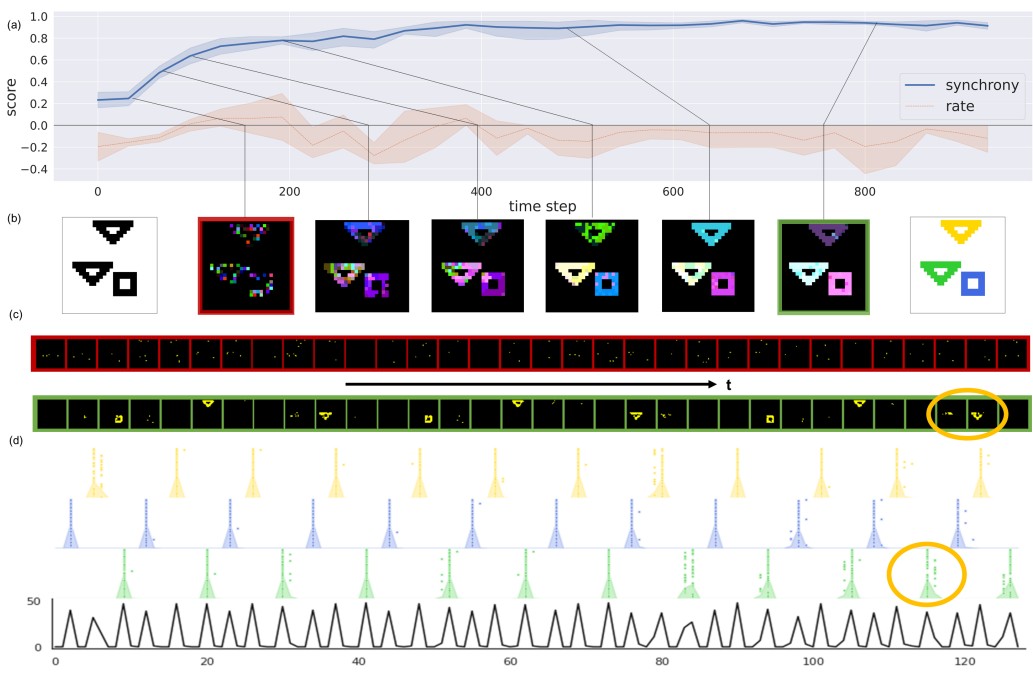

Figure 2: Temporal binding process in SCS. (a) Evolution of averaged synchrony score and rate score in five random initial condition. (b) left most: input image; right most: ground truth; middle five: firing phase of each feature neuron during simulation, indicated by color. The initial phase indicated by red box and bound phase indicated by green box (c) Zoomed in firing pattern of initial phase (red box) and bound phase(green box) in (b), respectively. (d) spike recording (color dots) of each neuronal group in last 120 time steps. Shallow shadow is the firing rate of each group. Black line is the total population activity, implying gamma-like oscillation.

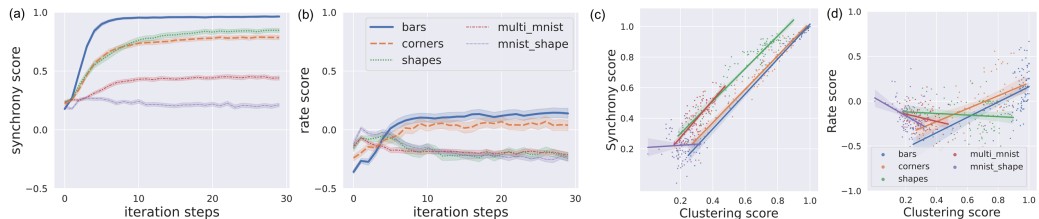

Figure 3: Synchrony score, rate score and clustering score of K-means. (a) Temporal evolution of synchrony score (b) Rate score (c) Synchrony score vs clustering score. (d) Rate score vs clustering score.

around zeros. Due to consistency between synchrony score and clustering score during the binding (Fig.3c green line), spike timing is indeed used by both binding process and clustering evaluation. These properties are roughly consistent across all datasets (Fig.3).

It is interesting to note that the grouping information is continuous (Fig.2 c.d, indicated by yellow circles): each feature (spikes of feature neurons) is not hard-paired to a discrete slot, but "softly" bound to one or multiple limited contingent regions in time dimension. Thus, firing dispersion seems to continuously encode the uncertainty of the binding in a natural way, acting like a coherence measure of grouping. The number of coherent states is not fixed by a pre-defined number $K$, but as an emergent phenomenon during the dance of ANN and SNN parts. These are special features of temporal binding both in **DASBE** and in the brain.

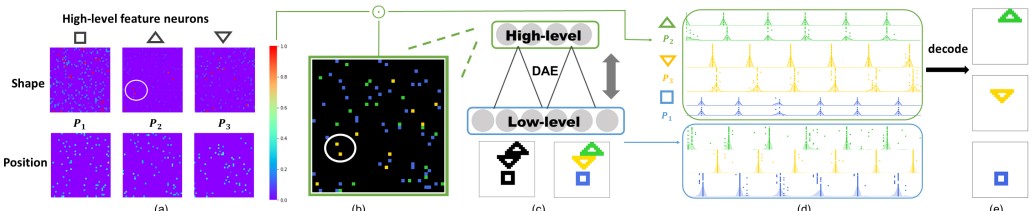

Figure 4: Hierarchical temporal binding in **DASBE**. (a) Feature neurons whose receptive field is related to certain features and independent of opposite type of features (b) Firing activity in hidden layer, firing time indicated by grouping color. Exemplar of found feature neurons are circled. (c) **DASBE** with hierarchical spike coding space (d) Spiking recording of low-level (bottom) and high-level (top) feature neurons (e) Decoding of synchronous firing neurons in the hidden layer

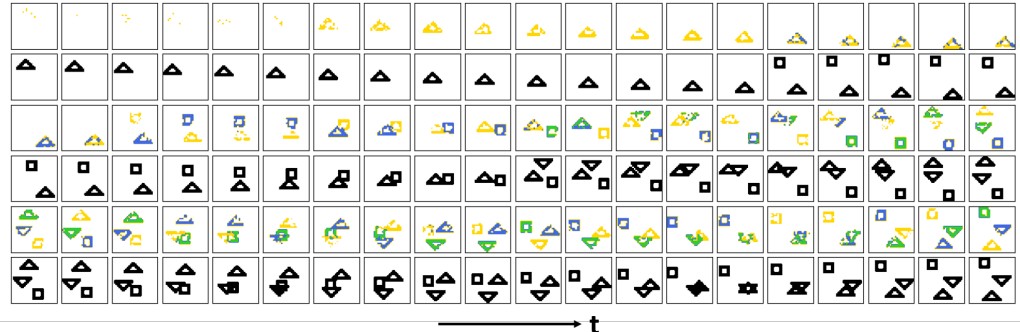

Figure 5: Binding in moving situation. Black-white image is the input and color image is the clustering result of latest spikes (1 delay period). Time evolves left-to-right and up-to-bottom.

### 4.2.4 Hierarchical binding in DASBE

Features in both the brain and ANNs has a hierarchical organization. Thus, we further study whether **DASBE** can bind hierarchical features in time. To verify it in shape dataset, we realized a spiking latent coding space (Fig.4c) in the DAE by making T a Bernoulli sampling process (Sec3.3; Alg1). Besides, we train the DAE with an additional supervised contrastive loss[56] only to learn a disentangled and distributed latent representation so that the result is more comparable to that studied in the original binding problem. Feature neurons are identified if they invariantly response to certain generative factors (shape and position in this simple case, Fig.4a). Note that, such supervision is not necessary for hierarchical binding in general cases.

The results show that the shape and position, two generative factors of the shape dataset, is represented in an approximately distributed and disentangled manner in the latent spike coding space of DAE (Fig.4 a.b). After the binding converges, both the low-level features(object-related pixels neurons) and high-level features (shape and position neurons) synchronize together (Fig.4 d.e). It is interesting to note that a hierarchy tree of features of different levels can be readout by following such spiking synchrony.

### 4.2.5 Binding in moving situation

We further study whether the simple **DASBE** can bind in dynamic situation. To verify, we generate the moving shape dataset (Fig.5), where each object can move along a fixed direction during the experiment with a constant speed (1 pixel per delay period) . Besides, a new DAE with a recurrent hidden layer is trained to predict next step of single object. Surprisingly, the **DASBE** can not only track multiple moving objects robustly even in overlap situation but also detect pop-up objects. The result shows that temporal binding in **DASBE** is flexible since the number of groups is not fixed, and dynamic since the model can re-correct itself based on the on-going situation.

# 5 Related work

The concept of synchrony was first illustrated in the ANNs by complex-valued Boltzmann machine [57], and latest developed into complex-valued auto-encoder[58]. However, these works abstract away firing rate/firing timing to the amplitude/phase of a complex number. Due to the limited range of phase values, approaches using complex-valued activation can only bind a small number of objects. On the contrary, the DASBE can flexibly use temporal correlation to bind large number of objects (See Appendix, examples for binding Bars).

Most work attempting to solve the binding problem in ANNs focus on one particular representational format: slots[2, 21, 22, 23, 24, 25, 27]. These slots are predefined in the network structure and explicitly separate the latent space for different objects. In contrast, the DASBE implicitly and softly assign grouping information to objects in a self-organized process.

Some earlier works solve temporal binding problem by attractor dynamics [36, 40] that memorizes the single objects. Here, the **DASBE** is also a dynamical system as a whole. The interesting point is that the pre-trained DAE seems to parameterize the attractor dynamics and explicitly plant the fixed point of this system (Proposition1) in an equivariant and structured way. Equivariance refers to that retrieved memories can adapt itself in feature space along with the change of stimulus, instead of a discrete converged prototype. Besides, the structure of the attractor is encoded in latent layers of the DAE, like shape and position. Thus, the capacity of SNN binding model get expanded so that they can be tested beyond toy patterns.

The top-down modulation may remind us of the attention mechanisms in solving the binding problem [20, 26, 59]. It also has been argued that temporal binding and attentional binding may be the two sides of the same coin and it seems the brain uses both methods [3, 60]. Thus, this model can be considered as bridging the temporal binding and the attention mechanism together.

# 6 Discussion

Can the **DASBE** provides insights to the temporal binding process in the brain? In a point of view of modeling, the model can be imagined as a kind of simplified cortical circuits (Fig.1d): The SCS composed of refractory pyramidal neurons with proximal and distal dendritic sites [47, 48, 49] mimics the layer II/III. The delayed loop of bottom-up/top-down processing[61, 62] in higher cortical layers (IV, V, VI) is formulated as a folded denoising autoencoder[63]. In this way, **DASBE** suggests a theory of temporal binding, which highlights the role of top-down modulation from pre-wired cortical circuits: the pre-wired cortical pathway, from thousands of years of evolution, encodes priors of prototypes of plenty of single objects. This prior is the essential cue for actively sampling the input and providing preparing signals for grouping. This prior is encoded in the decoder of DAE in **DASBE** during training. The preparing modulatory signals and driving signals act on pyramidal neurons at different dendritic locations in a non-linear manner [50]. The driving signals at the proximal sites dominate as gating signals while modulatory signals select which neuron to finally be activated [49]. The refractory period provides important competition in temporal dimension, thus firing alternates among different groups. Coincidence detectors [51] readout the synchrony events and feed to the higher cortical areas, where higher-level features are extracted. The feedback, carrying the prior cue, is delayed due to signal transmission along the pathway, of tens of milliseconds [64]. This delay period provides a window where grouping can alternate on gamma waves. Thus, the model highlights the role of the priors in pre-wired cortical circuits, and the interaction between refractory dynamics and the delay of feedback. In our model, it is the dance between SNNs and ANNs.

The aim in this paper is to demonstrate that the combination of top-down modulation and spike timing in an HNN framework [65] is a promising way to solve the binding problem. Since this framework is conceptually consistent with many essential properties of the binding capability, like unsupervision, common format, and multi-stability[3], while structurally compatible with both ANNs and SNNs, DASBE can be either enhanced by advancing ANN architectures to reach more competitive binding capabilities or added with more biological ingredients for modeling binding phenomenon in the neural system. We are convinced that DASBE can help to bridge insights from various aspects to solve binding problem in the future.

## 7 Conclusion

We propose a hybrid neural network architecture to combine top-down modulation and spike timing representation to solve the visual binding problem. The top-down feedback from the output of DAE modulates the spike timing of spiking neurons. Therefore, the solution to the binding problem is formed through iterative top-down/bottom-up processing and represented by neuronal synchrony.

## Acknowledgments and Disclosure of Funding

We would like to thank Mingkun Xu, Wenli Zhang and Faqiang Liu for general advice and feedback on the paper. We would like to acknowledge the anonymous reviewers for valuable suggestions and comments. This work was supported by National Nature Science Foundation of China (No. 61836004, No. 62088102), National Key Research and Development Program of China (Grant No. 2021ZD0200300), IDG/McGovern Institute for Brain Research at Tsinghua University, CETC Haikang Group-Brain Inspired Computing Joint Research Center.

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
