# A Appendix

In Section A.1, we highlight some limitations of our work as well as potential directions for future work. In Section A.2, we discuss the possible negative consequences of our work and ways to resolve them. In Section A.3,we show the computing resources and code availability. In Section A.4, we report the training details of all experiments. In Section A.5, we introduce the metrics or measures we use in the paper including AMI score, clustering score, synchrony score, rate score, and Victor-Purpura metric. In Section A.7,A.8,A.9, we report the experiment details one by one. In Section A.10, we report further qualitative results on all five datasets (results with different AMI score are selected). The temporal structure can have diverse properties and interesting features. Lastly, in Section A.11, we give the proof for the Proposition 1∼4.

## A.1 Limitations

We highlight several limitations that could possibly be addressed in future works.

**Input type**. In this paper, the input image is binary and $x_i$ as a binary gating variable to SNN layer. It is desirable to allow real-valued pixels to be the driving signal so that binding can be tested in real-world situations. It may be solved by soft gating mechanism or other binary encoding strategies in the future.

**Performance**. In this paper, we mainly illustrate the idea that properly combining ANNs and SNNs can lead to emergence of temporal binding representation. But we do not search the hyperparameters to optimize the binding performance and we only use minimal realization of ANNs (DAE). As a result, we do not compare the model with other state of art ANN models on unsupervised object-centric representations. But we believe it is possible that by combining more advancing ANNs models, the binding performance can be improved as well.

**ANN Architecture**. We highlight that current **DASBE** is not the only way to bridge temporal binding and ANNs. Actually, DAE is just a minimal realization. Introducing SNN coding space into various ANNs prototypes (dynamic routing in Capsule net[1, 2], Attention slot[3], message passing in GNN[4]) can possibly develop more possibilities of temporal binding models.

**Learning while binding**. In this paper, the binding process is based on pretrained DAE. In futures works, we hope the model can learn and infer at the same time, perhaps by introducing other ANN architectures into temporal binding model (Capsule net, Prednet, AttentionSlot) or by designing training protocol (eg. train if synchrony score is high enough)

**Biological modelling**. Although the model has several bio-related features, including temporal binding, phase precession, gamma wave, we do not make quantitative comparison with neuroscientific data. But the future work will illustrate whether **DASBE** binding can be a reasonable model for biological perception.

**Evaluation of binding**. In the paper, the binding evaluation (AMI score) is based on ground truth labels. However, in principle, binding (or segregation) is fundamentally multi-stable and do not have a "ground truth label" at all. In our experiment, we find several situations that the binding is reasonably convincing and even creative. But the AMI score is low just because the binding differs from the ground truth. Evaluation of binding is a hard problem and we hope future work can be based on scores of more flexibility.

**Dynamics of SNN**. In this paper, we stress the role of feedback attentions on modulating the SNN behaviors. Thus, inner-layer SNN dynamics is realized in a minimal way, only taking spike firing and self-refraction into account, ignoring inner-layer recurrent dynamics and Hebbian learning rule like STDP[5]. But it is interesting to compare the role of inner-layer dynamics against top-down modulation. Besides the refraction is based on a unified timescale. It is desirable to explore the functional role of heterogeneous refractory period in future works.

## A.2 Broader impact

Combining ANNs with a spike coding space(SCS) allows to flexiblly bind symbol-like entities from the perceptual input. It is a general architecture that can be initialized in various forms and be used in a wide range of domains. In our paper, we only consider binding on simple artificially generated

images. However, in principle, it is possible to develop its variant that can bind real-world data and learn further representations at the same time. And we found that the model can sometimes bind objects creatively beyond the groundtruth of benchmark. After all, binding itself is an open question, or even subjective. Thus, to assess whether the model binds or learns in unwanted ways, one can visualize the spiking representations as we did in the paper. However, if the representaion goes deeper within the hierarchy or the scene is not human interpretable, more work is required to develop the general visualization or analysis method, which can serve as a step towards more transparent and interpretable binding.

## A.3 Experiment resources and code availability

All experiments have been performed on ubuntu1$\sim$16.04.12 with device: CPU(Intel(R)Xeon(R) CPU E5-2640 v4 @ 2.4GHz) and 4$\times$GeForce RTX 2080 Ti. The python version is 3.6.3.

The code for results in this paper can be found on Github: https://github.com/monstersecond/DASBE

## A.4 Training details

The details of training neural networks for temporal binding are as follows:

1. *Multi-MNIST* uses contractive-autoencoder. *Hierarchical Feture Binding* ueses autoencoder with spiking hidden layer. *Moving Shapes* uses recurrent hidden layer which hidden layer $h_{t+1}$ at step $t$ gain inputs from the encoder network $\texttt{Encoder}(X)$ and the previous value of hidden layer $h_t$. The function of its recurrent hidden layer can be writen as $h_{t+1} = \texttt{Sigmoid}(W_h \cdot h_t + W_x \cdot \texttt{Encoder}(x) + b)$.

2. Loss functions are all Binomal Cross Entropy Error (between input and reconstruction) except *Multi-MNIST* which uses an additional contractive loss [6] as regularization and *Moving Shapes* which uses mean-squared loss.

3. All networks are trained with stochastic gradient descent (SGD).

4. Minibatch size is show in table 1. Sepecially, for RNN used in *Moving Shapes*, its minibatch size is 32 and the time step length of each sample is 10 (shorted for 32(20) in the table 1).

5. Encoder networks, decoder networks are set according to table 1 with Sigmoid output layer and ReLU for internal activation funciton.

6. Learning rate is set according to table 1.

7. Noise is set according to table 1. The meaning of 0.6$\sim$0.8: First, randomly choose a probability $p$ between 0.6 to 0.8 according to uniform distribution; Second, randomly change 1 to 0 with probability $p$. For noise setting in *Moving Shapes*, P(remove)=0.2 means randomly remove a frame with probability 0.2, P(1$\rightarrow$0)=0.5 means randomly change 1 to 0 with probability 0.5, P(0$\rightarrow$1)=0.05 means randomly change 0 to 1 with probability 0.05.

8. All datasets early stop when validation loss does not decrease for more than 40 epochs except *Moving Shapes* which is 25 epochs.

9. We use Back Propagation Through Time (BPTT) to train the spiking neural networks [7].

## A.5 Metric

### A.5.1 AMI score

Similar to the earier work [8], we use the Adjusted mutual information (AMI)[9] score to measure the binding performace because AMI is a score that measures the clustering similarity with invariance to permutations of grouping labels. In this work, the AMI score compares the clustering result of K-means against the ground truth segmentation. The K-means clusters each spiking neuron into one of the $K + 1$ groups (an additional group for background) according to their spike train with latest $N_{back} \times \tau_{delay}$ time steps. Each spike train is pre-processed by smoothing with an exponential filter with $\tau_s mooth = 1$ and decay factor 0.5, so that the distance measure between spike trains can be tolerant to slight spike timing shift within timescale $tau_{smooth}$. $N_{back} = 10$ in quantitative evaluation. The dataset and ground truth are all synthetic and the code can be found in Github. Origin code in [8] is available at https://github.com/Qwlouse/Binding

Table 1: Details of training neural networks

| Dataset | Encoder | Decoder | learning rate | noise | minibatch size |
|---|---|---|---|---|---|
| Bars | FC(400, 100) Sigmoid() | FC(100, 400) Sigmoid() | 1e-2 | 0.6~0.8 | 1024 |
| Shapes | FC(784, 512) ReLU() FC(512, 400) Sigmoid() | FC(400, 512) ReLU() FC(512, 784) Sigmoid() | 1e-2 | 0.6~0.8 | 1024 |
| Corners | FC(784, 100) Sigmoid() | FC(100, 784) Sigmoid() | 1e-2 | 0.6~0.8 | 1024 |
| MNIST+Shapes | FC(784, 250) Sigmoid() | FC(250, 784) Sigmoid() | 3.1685e-2 | 0.6 | 128 |
| Multi-MNIST | FC(2304, 500) Sigmoid() | FC(500, 2304) Sigmoid() | 1e-4 | 0.6 | 1024 |
| Hierarchical Feature Binding | FC(784, 600) ReLU() FC(600, 400) ReLU() FC(400, 350) ReLU() FC(350, 1600) Sigmoid() | FC(1600, 350) ReLU() FC(350, 400) ReLU() FC(400, 600) ReLU() FC(600, 784) Sigmoid() | 1e-3 | 0.6 | 512 |
| Moving Shapes | FC(784, 600) ReLU() FC(600, 300) Sigmoid() | FC(300, 600) ReLU() FC(600, 784) Sigmoid() | 1e-3 | P(remove) = 0.2 P(1→0) = 0.5 P(0→1) = 0.05 | 32 (20) |

## A.5.2 Silhouette score

Silhouette coefficient[10] is a score to evaluate the quality of clustering by measuring the inner-group coherence. The score is calculated using average intra-cluster distance (a) and average nearest-luster distance (b). The score is computed as $(b - a)/max(a, b)$. The distance is Euclidean distance by default and can be rewritten. We use Euclidean distance to measure the "clustering score" or "K-means score", which describe the clustering quality of K-means (eg. at the begining phase of binding, the clustering quality is low and at the convergent phase, the clustering quality is higher). The document can be found at https://scikit-learn.org/stable/modules/generated/sklearn.metrics.silhouette_score.html.

## A.5.3 Victor-Purpura metric

Victor-Purpura metric is a kind of edit distance introduced by [11] in 1996 to evaluate temporal coding in cortex recording. The distance between two spike trains is defined as the minimum transformation cost from one to the other. The transformation is consist of three elementary operation:

1. Add: the cost of adding a spike at certain time point is $1$.

2. Delete: the cost of removing a spike at certain time point is $1$.

3. Shift: the cost of moving a spike from one time point to another time point is $q \cdot \Delta t$. (The $q$ is an important parameter and $\Delta t$ is the length of time shift.)

The minimum is computed across all possible transformation path between two spike trains. The operation of adding and deleting ensure that there are at least one legal path. Examples can be found in [11]. It can be proven that Victor-Purpura metric satisfy the principle of (1) positivity, (2) symmetry and (3) triangle inequality[11]. Thus, with Victor-Purpura metric, spike train of different length construct a metric space. Unlike Euclidean metric, the Victor-Purpura metric is non Euclidean and

does not require a unified length of vectors. And it explicitly measures the timing difference by the Shift operation.

It is notable that the parameter $q$ can determine what quantity the metric measures. On one hand, if $q = 0$, then shifting the spike will not cause any cost and the metric only measures the spike count (independent of firing time). On the other hand, if $q \to \infty$, then shifting spikes always cause larger cost by "first adding then deleting". Thus, the metric count the number of spikes that are not "perfectly" synchronized, a harsh measure of synchrony. Thus, $q$ can be regarded as a time-scale parameter, modifying the metric between pure firing rate measure and pure synchrony measure. We realize computing this metric by dynamic programming strategy.

### A.5.4 Synchrony score and rate score

To evaluate the temporal binding and analyze time coding representation, it is essential to know the temporal structure of the spikes. In our case, it is the spiking synchrony. And the synchrony events are not global, but related to the grouping of features. The challenge of the evaluation is the following: (1) the spiking patterns themselves do not construct a linear space. And commonly used metric (like Euclidean metric) can not distinguish temporal structure from others (like firing rate) (2) The synchrony may not be perfect, but has a relatively small error range. The metric should be able to bear with such case. (3) The synchrony is not global, but more like the cluster synchrony[12].

Motivated by the challenges above, the synchrony in the binding process is measured by Silhouette coefficient based on Victor-Purpura metric ($q$ large). Here, spiking synchrony is defined as the temporal coherence level inside each cluster and such timing structure is explicitly measured by the Victor-Purpura metric. On the contrary, the rate score is measure by Silhouette coefficient based on Victor-Purpura metric when q is 0. We find that "$q = 1/3$" is good enough to measure the synchrony level in this work (synchrony score changes little for $q >> 1/3$, partly because 3 is a characteristic time scale in this work.). In sum, the K-means clustering is performed first and then synchrony score is computed based on the grouping result.

### A.6 Details for Table1

### A.6.1 Dataset

The examples of five datasets we use in training and testing are demonstrated in Fig.1.

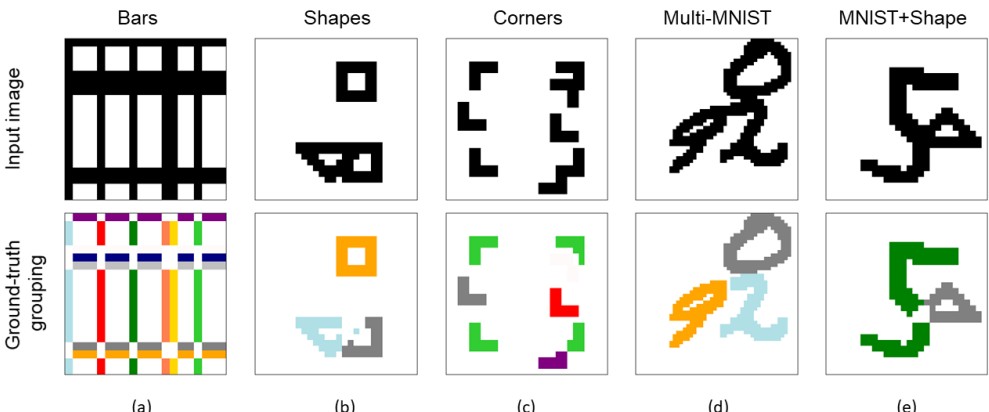

Figure 1: Examples of five datasets.(a) Bars (b) Shapes (c) Corners (d) Multi-MNIST (e) MNIST+Shape. The upper black-white figure is input image and the bottom color figure is the ground-truth grouping

**Bars** (Fig.1a), introduced by [13], are to demonstrate unsupervised learning of independent components of an image. Here, we use the modified version from [14, 8], which place 6 horizontal and 6 vertical full-length lines in random position in the image. Since the number of object is large, this dataset can test the binding ability of relatively larger number of objects

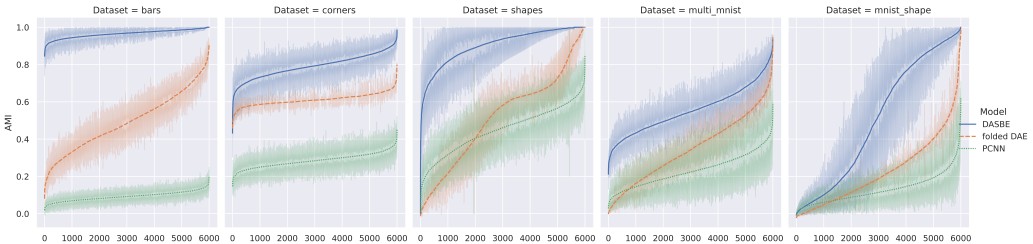

Figure 2: AMI score of DASBE, ANN baseline and SNN baseline on five datasets with 6000 samples. Five random seeds are used. The samples are sorted according to the mean AMI score (based on five random seeds)

**Shapes** (Fig.1b), taken from [14, 8], randomly place three types of shapes in an image, possibly with different level of overlap. This dataset (with our variant version) can test binding ability of different levels of features under static, overlap, or moving situation.

**Corners** (Fig.1c), introduced by [14, 8], consist of 8 corner shapes in random locations and orientations, and four of them are aligned to form a (unconnected) square. Thus, this dataset can test whether the binding ability is dependent on connected-ness.

**Multi-MNIST** (Fig.1d), is also taken from [8]. Three random MNIST digits are randomly placed in a $48 \times 48$ image. This dataset is more challenging because each prototype (digits) can have slight different shapes, so that features can have large amount of values.

**MNIST+Shape** (Fig.1e), taken from [14, 8], combine a random shape from the Shapes dataset with a single MNIST digit. Since the two object in the image have very different (1) type, (2) size, (3) diversity and the overlap is often present, this challenging dataset is useful to test binding in more complex situation.

### A.6.2 Experiment

We follow identical procedure on each dataset above. On each dataset, we first randomly generate single object dateset using relevant generative factors. Secondly, a DAE is trained to reconstruct the image disturbed by salt&pepper noise of different levels. Details can be found in the next section A.6.3. Third, after the DAE is trained, we combine it into the **DASBE** to test the binding ability in relevant multiple-object dataset. We divide the whole binding period into iterations of delay loops (eg. $T$=number of iterations$\times\tau_{delay}$). Fourthly, we use K-means to cluster the spiking patterns in SCS (only latest spikes are considered). The number of clusters in K-means is consistent with ground truth, which is K(number of objects)+1(background). Then we compare the clustering result with ground truth by the AMI score on each image and average them across the dataset of 6000 samples. Five random seeds are used to measure the error bars of the mean AMI score and we found that the mean AMI score is so robust that deviation is extremely low.

The distribution of AMI scores across all 6000 samples in the dataset is shown in Fig.2. The samples are sorted and reordered based on the AMI-score. It can bee seen in either table1 in the main text or Fig.2 that DASBE achieve high AMI score fastly in Bars dataset, which contain 12 bars in each image. As shown in 17, the DASBE indeed is able to bind objects with flexibly in temporal dimension. Such flexibility is rare in complex-value based models due to limited range of phase value.

Comparing with Bars/Shapes, MNIST is a more challenging object because it has a distribution of patterns due to different hand-written styles as explained in A.6.1.

The lower performance on MNIST-related dataset may be caused by several aspects. First, the DAE in this paper is implemented as a MLP. We find that the denoising reconstruction performance for DAE differs among the five datasets (Bars/Shapes>Corners>Multi-MNIST>MNIST-Shape). Such difference influences the binding process. Second, the objects in these datasets are more likely to overlap. The overlap in MNIST has a larger impact since it may make one digit like another digit (8/6 to 0; 7 to 1), and even cause a "shape" to disappear, which is also challenging for a human observer. The DASBE seems to have the "multi-stability" property. However, such valuable "intelligence" is not considered in our current evaluation (AMI score) because the AMI compares the grouping with a

Table 2: hyperparameters in the experiment

| Dataset | Bars | Shapes | Corners | Multi-MNIST | MNIST+Shapes |
|---|---|---|---|---|---|
| T | 1080 | 840 | 640 | 870 | 1450 |
| Iteration step | 20 | 30 | 50 | 30 | 50 |
| $\tau_{delay}$ | 54 | 28 | 32 | 29 | 29 |
| $\tau_{rfr}$ | 6 | 8 | 9 | 12 | 9 |
| $\tau_w$ | 3 | 3 | 3 | 3 | 3 |
| $p$ | 0.5 | 0.5 | 0.5 | 0.5 | 0.5 |
| $V_{th}$ | 1 | 1 | 1 | 1 | 1 |
| back | 10 | 10 | 10 | 10 | 10 |
| $\tau_{smooth}$ | 1 | 1 | 1 | 1 | 1 |

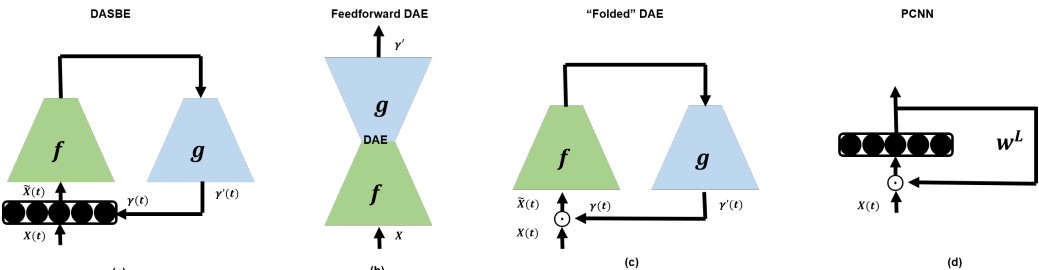

Figure 3: Comparison among **DASBE** and the baseline models.(a) **DASBE** composed by SCS and folded DAE (b) feedforward DAE (c) reduced ANN model: folded DAE, using its reconstructed output to modulate the input (d) reduced SNN model: PCNN, using fixed connection to feedback the output for modulation. $\odot$ refers to element-wise product

ground truth (pre-defined) and therefore lead to a pretty low score. Despite of this, combining the AMI score with qualitative results in A.10 (binding in five datasets), it can be seen that binding is still achieved in these dataset.

The hyper-parameters used in binding process are shown in the Table 2. T=iteration step$\times\tau_{delay}$, which are hyper-parameters for the simulation setting only. $\tau_{delay}, \tau_{rfr}, \tau_w,$, $V_{th}$ are the hyper-parameters for the model. $p$ and $V_{th}$ is the decay rate used in the coincidence detector. The "back" and "smooth" are hyper-parameters for k-means evaluations only. back is the number of last iteration considered and smooth is the timescale to smooth the spike train with an exponential kernel (decay rate is also 0.5).

### A.6.3 Training details

See Section A.4

### A.6.4 Baseline

**Folded DAE**

Different from the feedforward DAE (Fig.3b),the folded DAE are defined as using its reconstructed output to modulate its input at following time steps (Fig.3c). More formally,

$$\widetilde{x}(t) = (x(t) \odot \gamma(t))$$

$$\gamma(t) = f_{DAE}(\widetilde{x}(t - \tau_{delay}))$$

In the experiment, the folded-DAE is kept the same architecture as the DAE used in **DASBE**, all relevant parameter are also inherited. The clustering result is based on $\widetilde{x}(t)$.

As shown in Fig.3, folded DAE can be taken as a reduced ANN model. By the bootstrap effect of reconstructive feedback, the folded DAE can often capture one single object in the image. However, without spiking dynamics including spike generation and self-refraction as temporal competition mechanism, the folded-DAE is unable to switch between different objects Fig.4. Thus, all multi-object

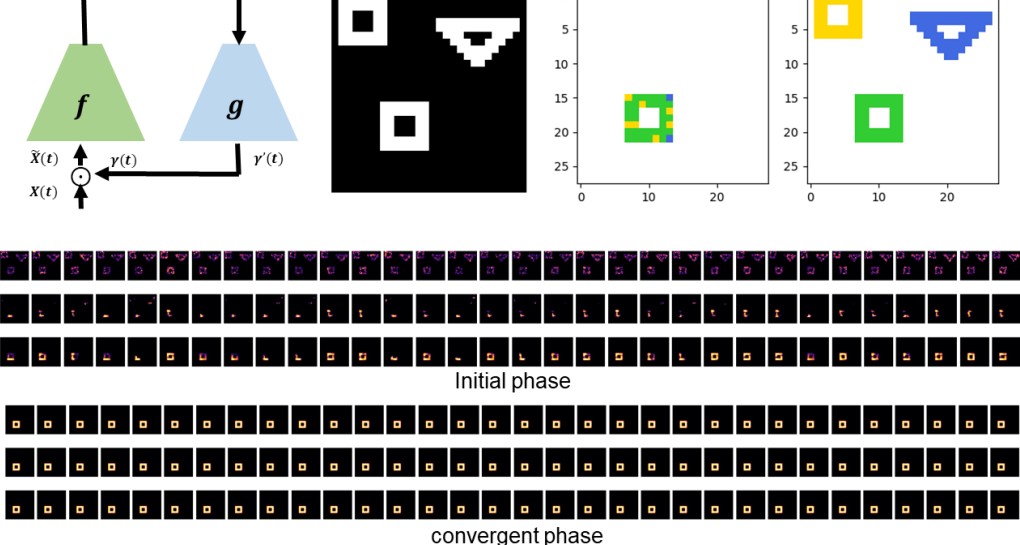

Figure 4: an example of binfing result of forlded DAE. upper panel (from left to right): network structure, input image, clustering result(based on the pattern in the bottom panel), ground truth. middle panel: temporal evolution of $\tilde{x}(t) = x \odot \gamma(t)$ ("non-spiking membrane potential") in the initial phase of binding. bottom panel: same as middle panel, in convergent phase of binding. The time step is arranged from left to right, from top to bottom.

datasets are challenging for it. However, by perfectly reconstructing one object, the AMI score can be much higher than random guess. But on dataset like Bars (much larger amount objects), they reach much lower AMI scores ( around 0.093 ).

**Pulse coupled neural network**

Pulse-coupled neural network (PCNN) was first introduced by [15, 16] and later developed for temporal binding through spiking behavior. In this paper, we inherit the model architecture in [17, 18]. Similar to **DASBE**, PCNN also use the two-compartment neuron model (Fig.5b), receiving driving input and modulatory input respectively (but they use alternative terms like feeding and linking). The two types of input also interact multiplicatively but the linking part is biased so as to be always larger than 1. The global inhibitory input is used to introduce the spatiotemporal competition among neurons. All connections in the PCNN are pre-defined, fixed weights with spatially exponential decay. As shown in Fig.3d, PCNN can be taken as a single-layer SNN that replaces the modulatory connections parameterized by the trainable DAE (in our **DASBE**) into a fixed, pre-designed spatial linking connections. Since it is a general framework of binding models in neuroscience and has close relation with our hybrid model, we choose this model as a reduced SNN baseline.

More formally, the PCNN we use can be described as the following equations.

$$F_{ij}(t) = x_{kl}$$

$$L_{ij}(t) = \sum_{kl=1}^{N} [w_{kl}^L \cdot s_{kl}(t)] * I(V_{kl}^L, \tau_j^L, t).$$

$$I(V, \tau, t) = \begin{cases} V \cdot exp(-t/\tau), & t \geq 0 \\ 0, & otherwise \end{cases}$$

$$w_{ijkl}^L = \begin{cases} exp(-\sqrt{(i-k)^2 + (j-l)^2}), & if\ |i-k| \leq 4, |j-l| \leq 4, |i-k| + |j-l| \neq 0 \\ 0, & otherwise. \end{cases}$$

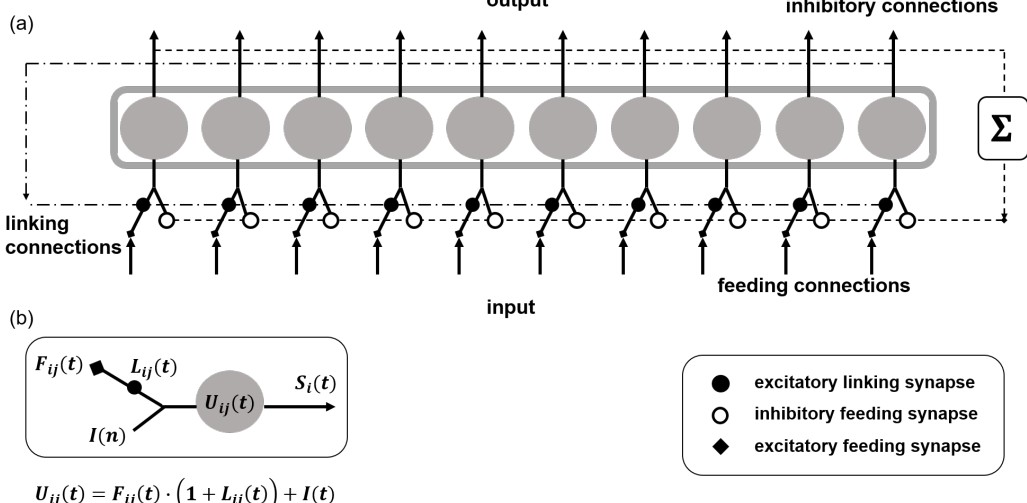

Figure 5: The pulse coupled neural network model (PCNN).(a) circuit diagram of PCNN. Three types of connection are identified: dashed line–inhibitory connection; dash-point line–linking connection; vertical-up black arrow–feeding connection. Three types of synapses are also identified. (b) single neuron model in PCNN. Each neuron has a 2-dimensional index (i,j) refering its location in a 2-dimensional lattice.

Table 3: hyperparameters in the PCNN experiment. "back" is the number of time-steps of latest recorded spikes being used for clustering evaluation. "$\tau_{smooth}$" is the number of time-steps being integrated by an exponential filter before clustering evaluation. Other parameters are for the PCNN model.

| Dataset | **Bars** | **Shapes** | **Corners** | **Multi-MNIST** | **MNIST+Shapes** |
|---|---|---|---|---|---|
| T | 500 | 500 | 500 | 500 | 500 |
| $\beta$ | 3 | 3 | 3 | 3 | 3 |
| $V^L$ | 1 | 1 | 1 | 1 | 1 |
| $\tau^L$ | 2 | 2 | 2 | 2 | 2 |
| $V^\Theta$ | 10 | 6 | 10 | 10 | 10 |
| $V_0^\Theta$ | 2 | 2 | 2 | 2 | 2 |
| back | 100 | 100 | 100 | 100 | 100 |
| $\tau_{smooth}$ | 2 | 2 | 2 | 2 | 2 |

$$U_{ij}(t) = F_{ij}(t) \cdot (1 + \beta \cdot L_{ij}(t)) + I(t), I(t) \leq 0$$
$$S_{ij}(t) = S(U_{ij}(t) - \Theta_{ij}(t)),$$

with

$$S(x) = \begin{cases} 1, & if x \geq 0 \\ 0, & otherwise. \end{cases}$$

$$I(t) = -w_I \cdot \sum_{ij} S_{ij}(t-1)$$

$$\Theta(t) = V^\Theta \cdot y(t) + \exp(-1/\tau^{Theta}) \cdot \Theta(t-1)$$

$*$ is the convolution. $\beta, V^L, \tau^L, V^\Theta, w_I$ are hyperparameters to tune, which are shown in the Table3 .The original PCNN has quite large amount of hyper-parameters to tune (more than 10!)[15, 16] and the binding result can highly dependent on the configurations of these parameters. We reduce the number of parameters based on [17, 18]. The $V^\Theta$ is initialized as $V_0^\Theta$, and feeding input are mixed with salt&pepper noise of probability 0.2 at each time step. The whole binding period is $T$.

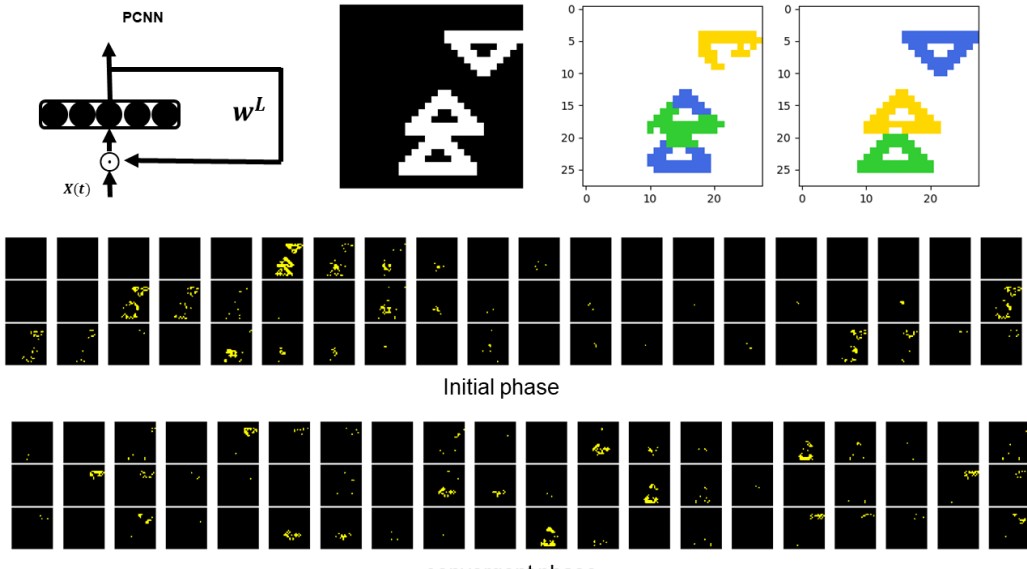

Figure 6: an example of binding result of PCNN. upper panel (from left to right): network structure, input image, clustering result(based on the pattern in the bottom panel), ground truth. middle panel: temporal evolution of spiking patterns in the initial phase of binding. bottom panel: same as middle panel, in convergent phase of binding. The time step is arranged from left to right, from top to bottom.

Note that, in principle, all the parameters of PCNN need to be tuned case-by-case on each "image" (in original paper), because the total excitatory strength (number of non-zero pixels) can possibly affect the dynamical behavior of the system. Only properly chosen configurations can result in the successful binding. Such a large amount of parameters are very time consuming to be tuned in practise for complex dataset like those we use. In **DASBE**, however, these modulatory connections can be learned automatically. Besides, the PCNN binds objects based on connectivity only. Thus, the dataset Bars (contain a large amount of interconnected objects), Corners (contain objects composed of unconnected parts) and Shapes, Multi-MNIST, MNIST+Shapes (contain frequent situations where objects can overlap badly) all challenge such traditional SNN method. As shown in Fig 6, for properly tuned PCNN model, the isolated object is separated but the connected objects are bound as a whole.

### A.7 Details for Fig2 & Fig3

The network and parameters are consistent with A.6. Total binding period $T = N_{delay} \times \tau_{delay}$. More details is shown in Fig.7

**Synchrony score**. The synchrony score, rate score and silhouette coefficient (See Section A.5) are measured for each iteration step of length $\tau_{delay}$ during the binding process. The length of latest spike train being considered is also $\tau_{delay}$ (in other words, $N_{back} = 1$)

Notably, the comparison between synchrony score with rate score (Fig.7a, Fig.8, Fig.9) shows that in each group acquired by K-means, the temporal coherence is positively correlated with the silhouette score along the binding period. This means that neurons in the same cluster has more synchronous firing pattern. The interesting point is that K-means compare each spiking pattern against an averaged firing density (firing rate) for grouping and does not explicitly take temporal structure into account. However, the comparison between synchrony score and clustering score confirms that the clustering result (eg. AMI score and visualization) reflects the timing structure of the representation. On the contrary, the rate score remains near zero and do not have salient correlation with the clustering score. Thus, the firing rate of spike trains in different clusters does not have visible distinctions. In other words, the grouping information is independent of firing rate. The firing rate just reflects the existence of feature that is being grouped.

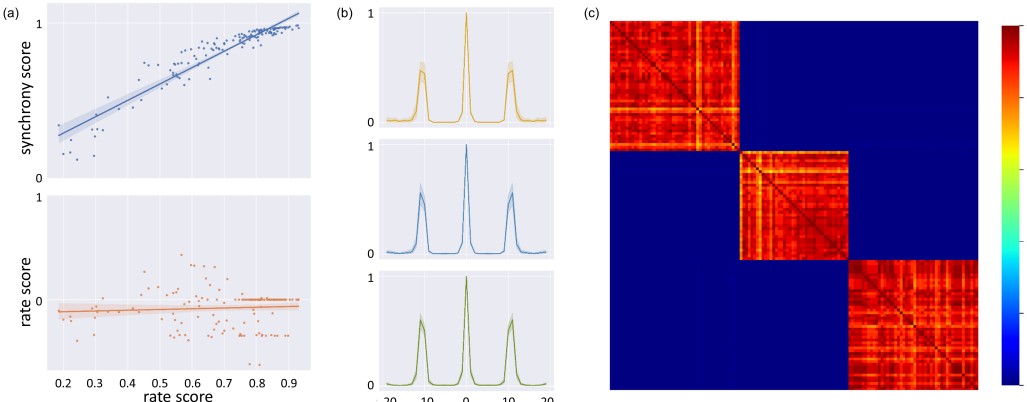

Figure 7: More analysis details for the example shown in Fig2 in the main text. (a) synchrony/rate score compared to clustering score, measured step-wise during the binding process. It can be seen that synchrony is consistent with clustering score, while rate score remains around zero. (b) averaged auto-correlation function in each group of neurons in Fig2 d in the main text. It can be seen that oscillatory behavior emerges at time scale around 10 time steps. (c) correlation matrix of neurons based on respective spike train in the bound phase. Three clear blocks can be observed.

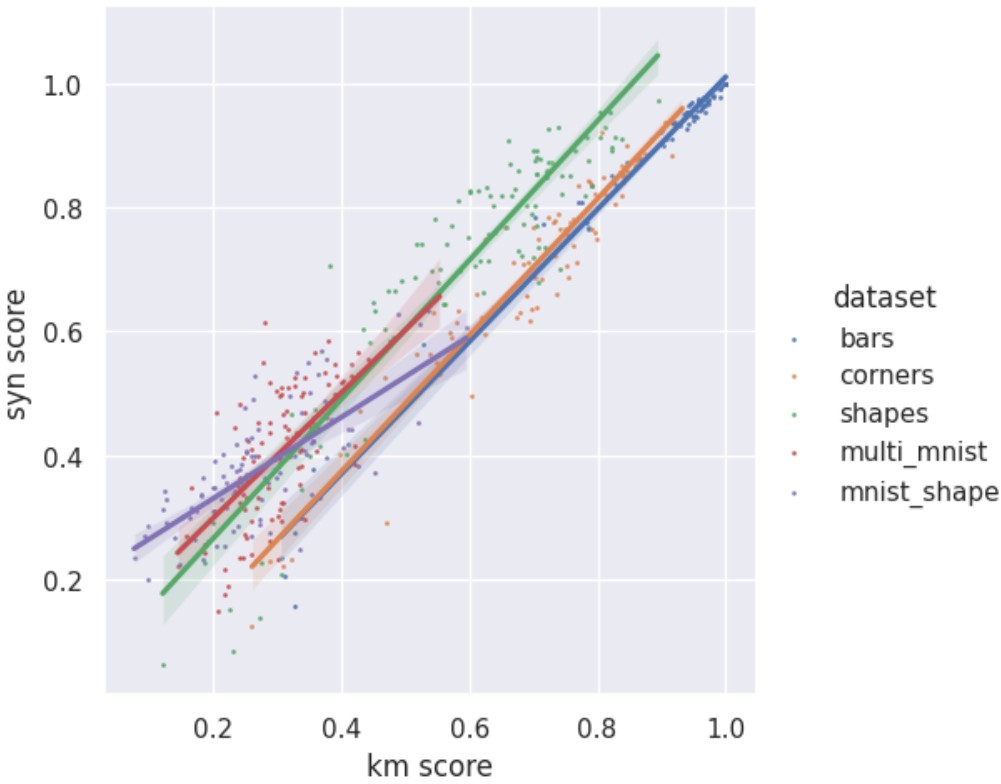

Figure 8: synchrony score vs clustering score. copied from Fig.2b in the main text

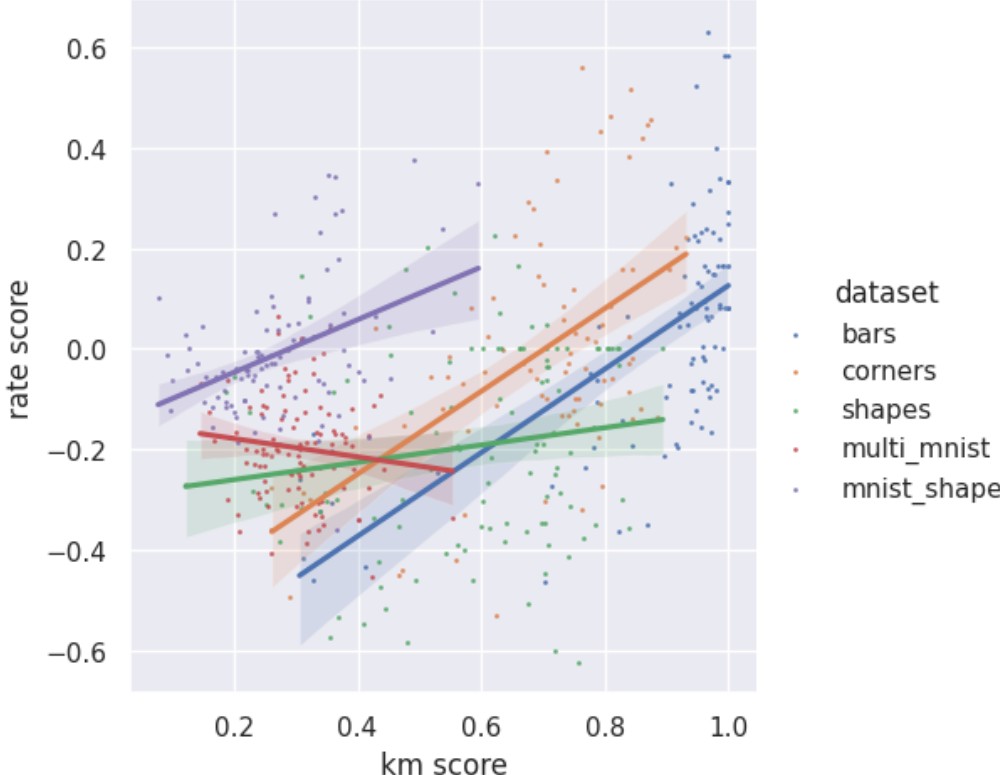

Figure 9: rate score vs clustering score. copied from Fig.2b in the main text. Note the different ticks on y axis

To show whether the coding scheme and synchrony behavior is consistent in other dataset, we compute synchrony score, rate score and silhouette score for 100 randomly selected samples from the other datasets (Fig.10,Fig.11). The general result holds but in the more challenging dataset like Multi-MNIST and MNIST+Shapes, the convergence and temporal coherence is less clear. As can be seen in the following section. Even in these cases, the temporal structure is still present, but the precision of the temporal structure is lower. For example, firing of different groups still alternate in time but with some overlap. Since the Victor-Purpura metric is sensitive to synchrony, such "non-perfect" temporal structure may cause a relatively underestimated synchrony score.

**Coloring**. We give color to each neuron in the SCS according to the spiking firing times. However, the challenge is that each neuron can fire multiple times instead of one. Thus, we assign the color according to an periodic mapping function, whose period is $\tau_{rfr}$. We choose this period because we find a gamma-like rhythm in the population activity and this mapping can roughly reflect the temporal structure of spikes relative to such a reference frame. The color is based on latest spikes of length $\tau_{delay}$

**Spiking pattern and spike recording**. Both are qualitative visualization of spiking representation in SCS. The former preserve the spatial structure of spikes within an 2-dimensional image while the latter concentrates on temporal structure.

**Correlation matrix and auto-correlation function**. Correlation matrix (Fig.7c) is derived by computing pair-wise Euclidean distance among neurons in SCS. The latest spike train of length $10 \times \tau_{delay}$ are considered. The spike train is smoothed by an one-step exponential filter of decay 0.5. Averaged auto-correlation function (Fig.7b) of population activity (summed spikes) inside each group of neurons is computed to measure the temporal correlation. The latest spike train of length $10 \times \tau_{delay}$ are considered.

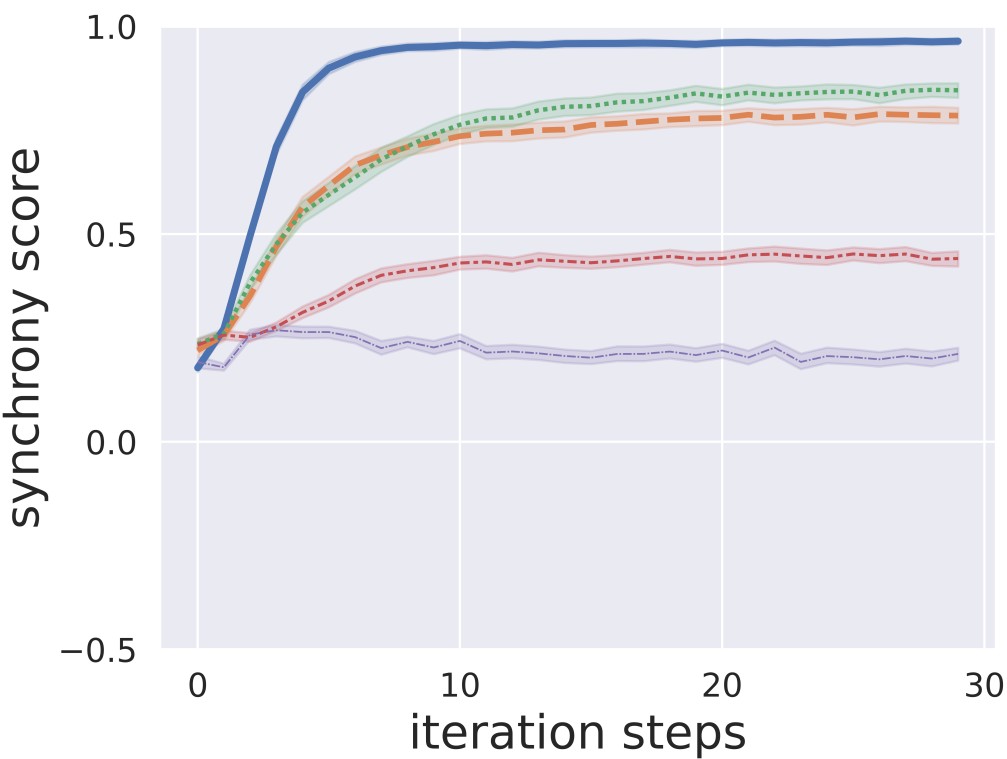

Figure 10: synchrony score and clustering score. copied from Fig.3.a in the main text.

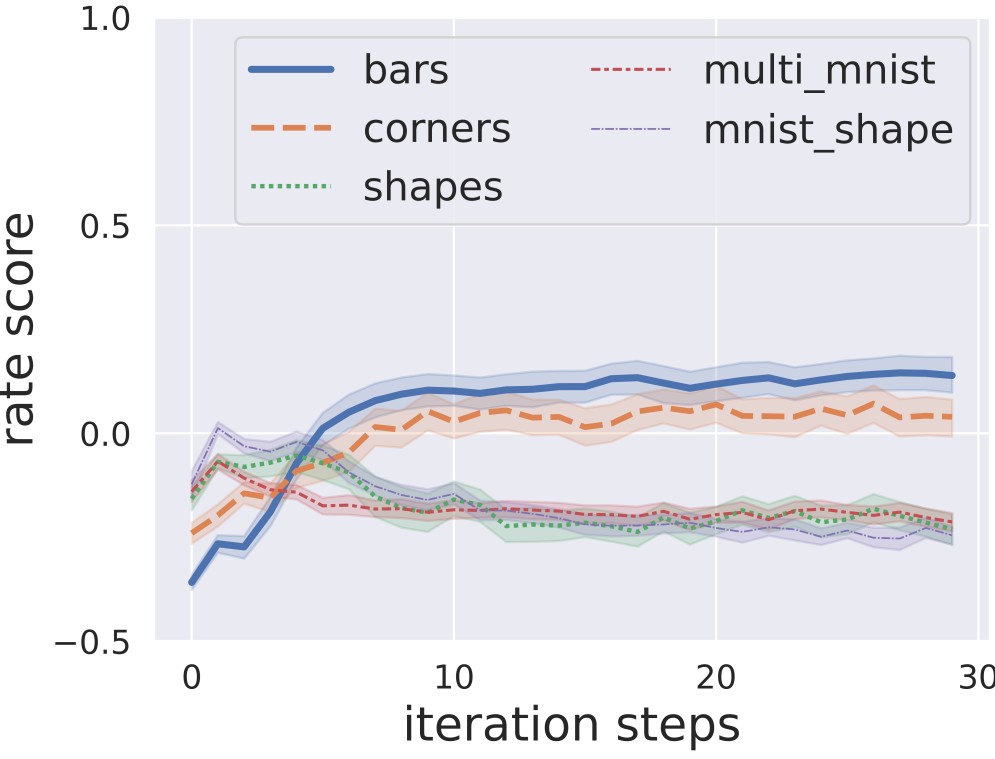

Figure 11: rate score and clustering score. copied from Fig.3.b in the main text.

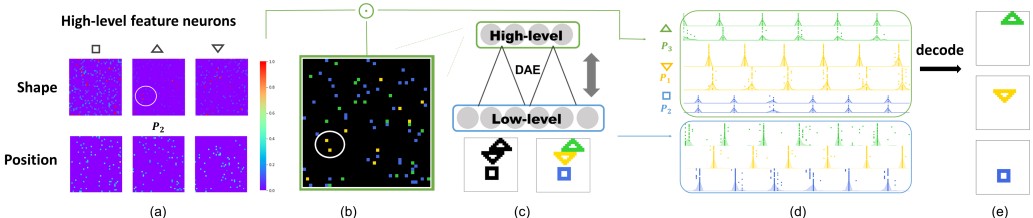

Figure 12: Hierarchical binding process. copied from Fig.4 in the main text

## A.8 Details for Figure 4

**General experimental setting**

In Figure4 in the main text, we study the hierarchical binding ability of **DASBE**. Low-level features is represented in SCS and high-level features is represented in the latent layer of DAE. Hierarchical binding requires four key elements: (1) **DASBE** has a spiking representation for higher level features, so that spike timing structure can be exploited to provide grouping information. (2) the latent representation should be disentangled and distributed, so that there are neurons invariantly respond to certain features (feature neurons). (3) We should be able to identify and track these feature neurons (4) The neurons can be bound in time alternatively according to which object they belong to.

For the first element, we minimally extend the DAE with an additional Bernoulli sampling process $T$ to transform the continuous encoding into spike encoding in the latent space (Fig.12c). For the second element, we use an additional contrastive loss, which is an unsupervised method to encourage a more factorized representation. For the third element, we estimate the receptive field of each spiking neuron in the latent space. Imagine that we are neuroscientists faced with a super-complex system like the brain. How can we known something about the responsive properties of the millions of neurons? We provides stimuli of all kinds of features and if there are neurons respond to one feature (like shape/position), a feature neuron can be identified (shape neuron/place cell). Note that feature neuron is identified within the experimental "stimuli set": they may also respond to other stimuli as well. And note that we do not need to find out all feature neurons, most of the time we only find some of them. So in our experiment, we inherit the idea in neuroscience. For example, to find out "triangle neuron", we provide all images (stimuli) with a triangle at random places to the DAE, and then count the firing activity of each neuron (Fig.12a upper). If there are neurons keep an consistent high firing rate across all "random-single-triangle" images, we identify these neurons as "triangle neurons" (eg. the neurons being circled in Fig.12a), because they have the receptive field of shape triangle and invariant to the changes in the opponent feature (position). The same process applies to the other shape neurons. For position neurons, we generate a large number of images with different shapes (mixed with salt&pepper noise) on certain position and also count the firing rate of each neurons. In the experiment, we find that neurons are not all perfectly disentangled into certain generative factor, but an intermediate state between pure-disentangle and pure-distribute ( especially for position neurons (Fig.12a bottom) partly because the disentangled encoding of position is more challenging in our unsupervised training). However, like neuroscientists, we can still find the feature neurons we want and get to know something about the super-complex representation within the latent space, even though these neurons may have other responsive properties and we do not capture all of them. For the fourth elements, with these feature neurons in hand, we compare the firing activity in the latent space during the binding process with all these candidates (Fig.12b. eg. the circled neurons). And we find out that the feature neurons we identified synchronize as expected (Fig.12d). We also decode the synchronized neurons (shape+position) in the latent space with the decoder of the **DASBE** and find that the reconstructed object is consistent with the input image (Fig.12e).

To be more specific, the network for *Hierarchy Feature Binding* is trained using a supervised contrastive learning method [19] to bias the disentanglement in the latent representation. In this section, we firstly introduce the extra loss function added to the reconstruction error and secondly show how to find the shape and the position feature neuron index sets.

**Contrastive loss** Taking dataset *Shape* as an example, a minibatch of training dataset is divided into three different sub-datasets $D_i$ ($i = 0, 1, 2$) each has a different shape (i.e., square, triangle facing up, triangle facing down). The three different sub-datasets $D_i$ have equal size. The similarity $s_{ij}$ of two

sub-datasets $D_i = \{d_k\}$ and $D_j = \{d_l\}$ is difined as:

$$s_{ij} = \sum_{\substack{d_k \in D_i \\ d_l \in D_j}} \frac{f(d_k)^\top f(d_j)}{\|f(d_k)^\top f(d_j)\|}$$

where $f(x)$ means the encoder network. The loss of contrastive learning we used is divided into two parts. The first part of loss is:

$$loss_{pos} = \frac{1}{|\{0,1,2\}|} \sum_{i=0}^{2} s_{ii}$$

The second part of loss is:

$$loss_{neg} = \frac{1}{|\{(i,j)|(i \neq j) \wedge (i,j \in \{0,1,2\})\}|} \sum_{\substack{i \neq j \\ i,j \in \{0,1,2\}}} s_{ij}$$

The overall contrastive learning loss is defined as

$$\mathcal{L}_{contrastive} = loss_{pos} + loss_{neg}$$

The learning loss of the contrastive autoencoder is the weighted sum of the contrastive learning loss and the reconstruction error $\mathcal{L}_{reconstruct}$ of the autoencoder:

$$\mathcal{L} = w_1 \cdot \mathcal{L}_{contrastive} + w_2 \cdot \mathcal{L}_{reconstruct}$$

where $\mathcal{L}_{reconstruct}$ refers to the reconstruct loss of the autoencoder, $w_1$ and $w_2$ are constant weight coefficients. In our experiments, $\mathcal{L}_{reconstruct}$ refers to the binary cross entropy loss, $w_1$ and $w_2$ are set to 2, 1 respectively.

**Finding the disentangled feature neuron** After training the constrastive autoencoder, we choose the shape feature neuron set $I_{shape}$ and position feature neuron set $I_{pos}$ of the hidden layer according to the following steps. First we generate sets of samples ($20000 \sim 60000$) either fixing the shape type or the position of the single object (termed as $D_{shape}$ or $D_{position}$). Second, we count the total firings of latent-layer neurons given $D_{shape}$ or $D_{position}$. Second, the firing rates of hidden neurons are sorted and neuron index set $I_{shape}$ with top $k$ firing rate are selected. $k$ is determined by the maximum value such that $\gamma \cdot f_k > f_{k+1}$, where $f_k$ is the firing rate of the $k^{th}$ largest neuron, $\gamma$ is an empirical constant usually set to 0.9 for finding shape feature neurons and 0.6 for finding position feature neurons. Third, following the similar method, we calculate the candidate position feature neuron index set $\tilde{I}_{pos}$. Forth, in the binding process, we compare the firing of hidden neurons $I$ against the candidate feature neurons. So, the set of final feature neurons are: $I_{shape} = \tilde{I}_{shape} \cap I$ and $I_{pos} = \tilde{I}_{pos} \cap I$.

Note that **DASBE** provides a network architecture different from classical feed-forward network of hierarchical feature representation[20]. Here, different levels of features communicate with each other to determine a final stable representation.

To compare with the temporal binding results of the constrastive learning autoencoder, we remove the constrastive loss $\mathcal{L}_{contrastive}$. The results are shown in Fig.13 and Fig.14.

On the one hand, disentanglement is much more visible in contrastive training cases(Fig.13 upper panel). Without contrastive loss or any other constraints, latent layer is not likely to learn a disentangled representation but a rarely complex representation(Fig.14 upper panel). On the other hand, either with or without contrastive loss, the model is able to bind the features at lower and higher levels, though in the later case, the representation is less explainable (but still can synchronize). It can also be seen that the synchrony is more salient with contrastive loss. The reason might be two-fold: 1. learning a compact latent representation is more challenging for spiking neurons due to binary coding. 2. contrastive loss biases the latent representation to be disentangled, which mitigate the challenge. Therefore, disentanglement in the higher-level representation seems to benefit temporal binding. Notably, though such disentanglement is introduced by a supervised contrastive loss in this model, many other unsupervised method[21, 22] is reported to also able to create such disentanglement.

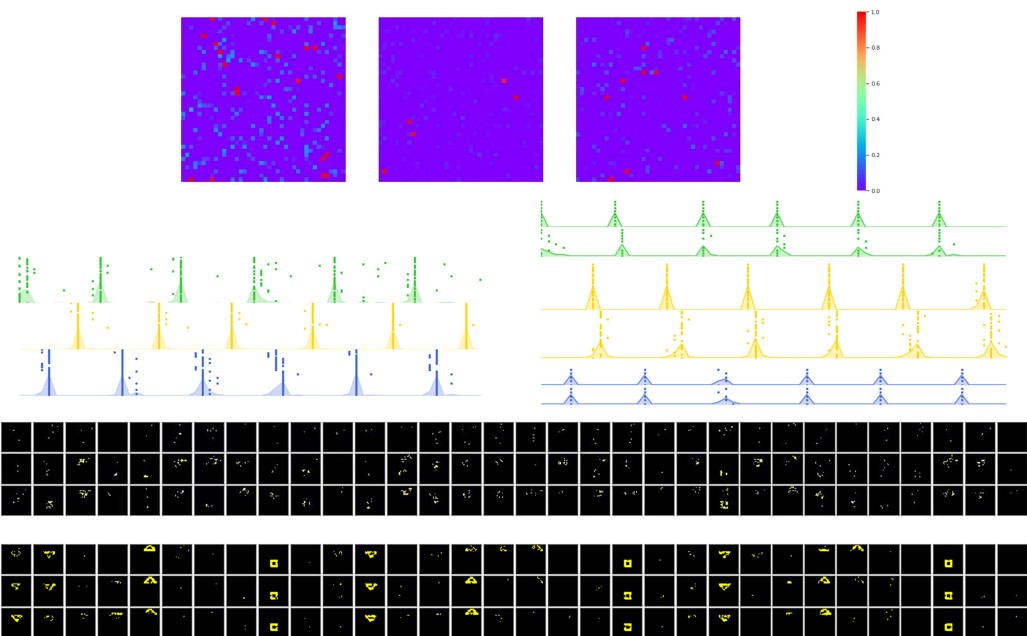

Figure 13: Results with contrastive loss. upper panel: statistical response to "shape" features of the latent layer neurons. Copied from the Fig4 in the main text. Middle panel: spiking plot for low-level features neurons(left) and high-level feature neurons(right).The neurons are arranged and colored based on which feature it encodes. Bottom panel: spiking pattern in initial phase (row1 row3) and convergent phase (row4 row6)

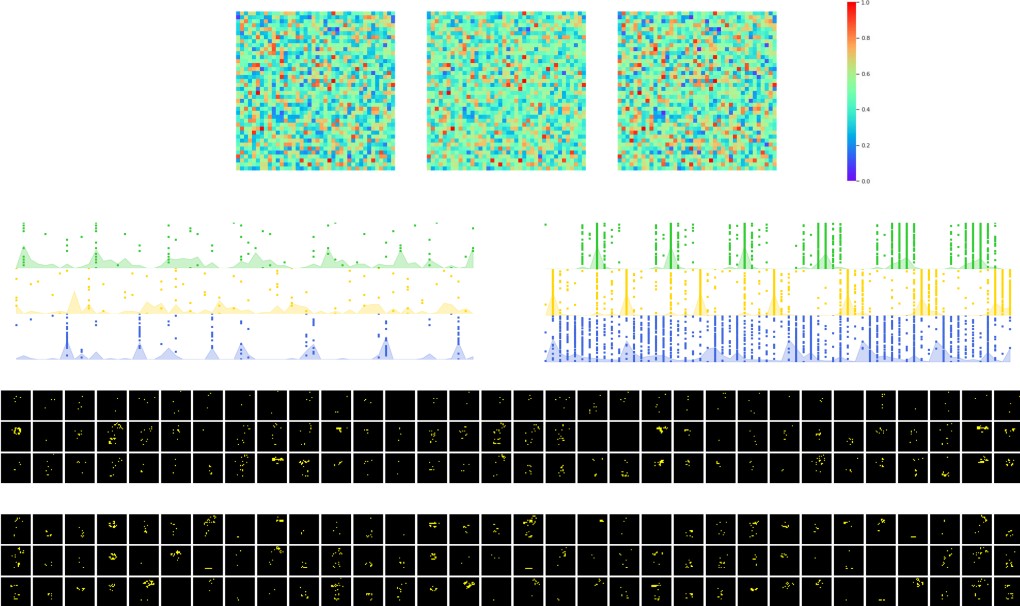

Figure 14: Results without contrastive loss.upper panel: statistical response to "shape" features of the latent layer neurons. Each sub-figure is the average firing rate of latent neurons given inputs of same shape but varied positions. The latent vector is reshaped to be a square for visualization. Middle panel: spiking plot for low-level features neurons(left) and high-level feature neurons(right). Since the features are not disentangled and there is no ground truth, we coloring the neurons based on k-means clustering of spiking patterns and no separation of features is provided. Bottom panel: spiking pattern in initial phase (row13) and convergent phase (row4 row6)

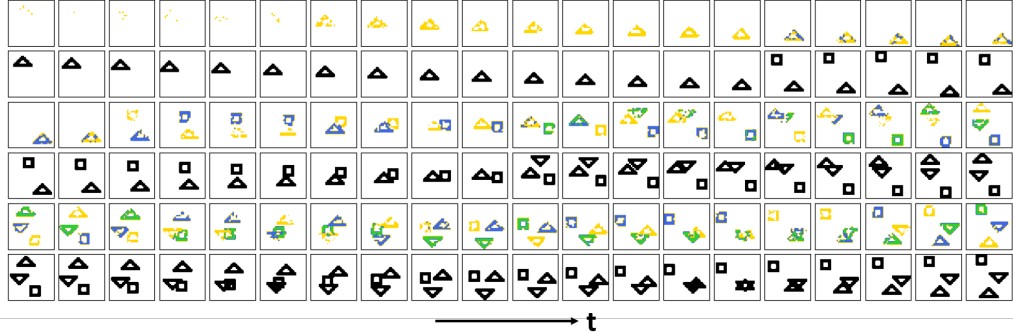

Figure 15: Binding moving objects. copied from Fig.5 in the main text. The demonstrated images are snapshot of input data and spiking representation being taken every $\tau_{delay}$ time steps. The black images are the generated data and the color image is visualization of temporal binding by clustering the latest spikes in the past $\tau_{delay}$ time steps. Time from left to right and from up to bottom.

## A.9   Details for Figure 5

Different from static situation, moving situation is more challenging for a temporal binding model. (1) Features are being constantly shifted in the changing world (eg. the object identity, position of the object, etc.), so that the internal representation need to be able to dynamically change at the same time. If features compete with each other in time dimension, then, this competition is required to influence next step, which may have a different firing structure. (2) **Timescale**. Temporal binding use time to provide grouping information, which means the grouping is entangled with time. This is fine in static situation because time is not a generative factor. However, in the moving situation, object MOVES. So here, time is used for two things: one for representation and one for describing the world. At each time step, to avoid superposition catastrophe, time is needed to group the object. But object also changes with time and their features is ever-changing. Now, how can temporal binding work? These are the motivations behind this experiment. In a word, moving situation challenges the temporal binding model by revealing the implicit entanglement between grouping and time.

Thus, in the experiment, there are two time-scales. The first is the macroscopic (or behavioral) timescale $\tau_M$ to describe the movement of the object. The second is the microscopic (or representational) timescale $\tau_m$ to bind object based on time code. $\tau_M >> \tau_m$. It is also the case in the brain. The timescale for temporal coding is around 10ms (or even less) and the behavioral timescale is around 500ms (0.5 second). Thus, a moving object is moving w.r.t $\tau_M$ but may be quasi-static with w.r.t. $\tau_m$. The object appears suddenly and moves fast w.r.t $\tau_M$. In this experiment $\tau_M = \tau_{delay}$ and $\tau_m = 1$. In this situation we study whether the **DASBE** can successfully bind multiple-moving objects in the time dimension.

More specifically, we extend DASBE to moving dataset in this section. Different from the static dataset, the objects in moving dataset appear one by one at regular intervals. Once an object appears in the image, it moves according to a randomly chosen constant direction and constant speed. Whenever the object reaches one of the image edges, it reverses its direction immediately (black images in Fig.15 are the input data). To deal with the moving situation, we modify the feed-forward DAE to have a recurrent latent layer. The network architecture is described in Section A.3. The recurrent DAE is trained to directly predict the next step of the single object given the current input image $x_t$, so that the DAE can learn a kind of predictive inference ability.

Using the recurrent DAE, we extend DASBE to the moving dataset (Algorithm 1). In each iteration step, the recurrent DAE gains a new image from moving dataset. Different from static dataset, in each time step, `latent` is calculated according to the hidden state of both current step $t$ and the previous step $t-1$. Thus, differently, the hidden state should be stored as `hidden`$_{pre}$ initialized as zero vector. And at each step, `hidden`$_{pre}$ is set to be the current hidden state `hidden`. At the end, `attn`$[t]$ stores the predictive results of the next $\tau_{delay}$ position of each detected shape.

To track the binding representation along the whole period, at each iteration step (length of $\tau_{delay}$), we use K-means to cluster the neurons in SCS based on the latest $\tau_{delay}$ spiking pattern (Fig.15

**Algorithm 1** Extend temporal binding process in **DASBE** to moving dataset. The input $x_t$ is a binary vector of dimension $D_{image}$ at time step $t$. The attention map (attn) is a real-valued vector of dimension $(T + \tau_{delay}) \times D_{image}$. $T$ is the length of binding process. We initialize attn $[t]$, $t \in [-\tau_{delay}, 0]$ as independent positive samples (attn $> 0$) from standard Gaussian distribution $\mathcal{N}(0, I)$. Refractory variable rfr is initialized as $0$. $\tau_{delay}, \tau_w, \tau_{rfr}$ are time-scale parameters. In our experiments, we set $\tau_{delay} \in [20, 60]$, $\tau_w = 3$, $\tau_{rfr} \in [0, 15]$, $T \approx 1000$.

---

1: **Input**: $x_t \in \{0, 1\}^{D_{image}}$, attn $[-\tau_{delay} : 0] \sim |\mathcal{N}(0, I)| \in \mathbb{R}^{\tau_{delay} \times D_{image}}$

2: **Layer params**: Norm; CD; Ber, Softmax; MLP$_f$, MLP$_g$: encoder and decoder of DAE; T: Bernuli sampling operation for binary latent space or identity map for real-valued latent space.

3:  $\quad$ attn $[-\tau_{delay} : 0]$ = Norm(attn $[-\tau_{delay} : 0]$);  $\qquad$ # normalization of initial attention map

4:  $\quad$ hidden$_{pre}$ = 0;

5:  **for** $t = 0 \ldots T$

6:  $\qquad$ context = attn $[t - \tau_{delay}]$;

7:  $\qquad$ firing_rate = $x_t \odot$ context $\odot$ (rfr == 0);  $\qquad$ # $\odot$: element_wise product

8:  $\qquad$ spike $[t] \sim$ Ber(firing_rate);  $\quad$ # Ber: sampling according to Bernoulli distribution

9:  $\qquad$ rfr+ = spike $[t] \cdot \tau_{rfr}$;  $\qquad$ # set spiking neuron into refractory period

10:  $\qquad$ rfr = max(rfr $- 1, 0$);  $\qquad$ # update refractory variable

11:  $\qquad$ input2dae = CD(spike $[t - \tau_w : t]$);  $\qquad$ # CD: coincidence detector

12:  $\qquad$ hidden = MLP$_f$(input2dae) + $W_h \cdot$ hidden$_{pre}$ + $b$;

13:  $\qquad$ encoder = Sigmoid(hidden);

14:  $\qquad$ hidden$_{pre}$ = hidden;

15:  $\qquad$ latent = T(encode);  $\qquad$ # latent space can be real-valued or binary

16:  $\qquad$ attn $[t]$ = Softmax(MLP$_g$(latent));

17:  **return** spike

---

color image). The clustering results shows that **DASBE** can quickly detect pop-up objects and track multiple moving objects.

### A.10 Visualization of results in other datasets

In this Section, we provide further temporal binding results on all dataset. According to the sorting of mean AMI score (Fig.2), we show the qualitative result of different AMI scores. We can see that some super low AMI score may due to the ill-generated ground-truth (Fig.18). And some super high AMI score may not guarantee perfect synchrony (Fig.32).

Interestingly, it can be seen in Bars dataset (Fig.16,Fig.17) that the temporal structure can be very complicated, not restricted to the homogeneous oscillatory activity. Some neurons may burst and then go to silence. Some neurons can fire more frequently than another neuron. The temporal structure is not predefined or supervised, but self-organized in an unsupervised manner. Despite of their complexity, it is clear that spiking representation still use temporal correlation to provide grouping information. Similar result can also be found in Corners dataset (Fig.22, Fig.23, Fig.24).

Besides, in Fig.26, we can see that the DASBE can bind objects creatively in unexpected ways. In this case, the 0 and the 4 overlap so that it looks like the 8. Then the DASBE flexibley bind them into a 8 and a 6. Of course it is evaluated as a unsuccessful binding based the ground truth (AMI=43.3). But actually such binding makes good sense and beneficial for remaining a self-consistent state (by continuous self-evidence process). It is the feature of both DASBE and the human perception. We constantly found those interesting cases during our experiment.

In figures on MNIST+Shape dataset (Fig.30,Fig.31,Fig.32), we find that the temporal structure of the spiking representation has a kind of "precession" (in a reversed direction): if we assume the delayed feedback provide a temporal coordinate in SCS (like periodic topdown modulation can induce oscillatory activity in neural circuit) the firing time shift in this temporal coordinate every feedback cycle. Similar neural behavior is observed in Hippocampus and believed to be essential for memory formation and recall[23].

It is notable that the synchrony is not absolute and the grouping information is continuous in time. Such property can be clear especially in samples that do not achieve a super high AMI score (Fig.19,Fig.32)

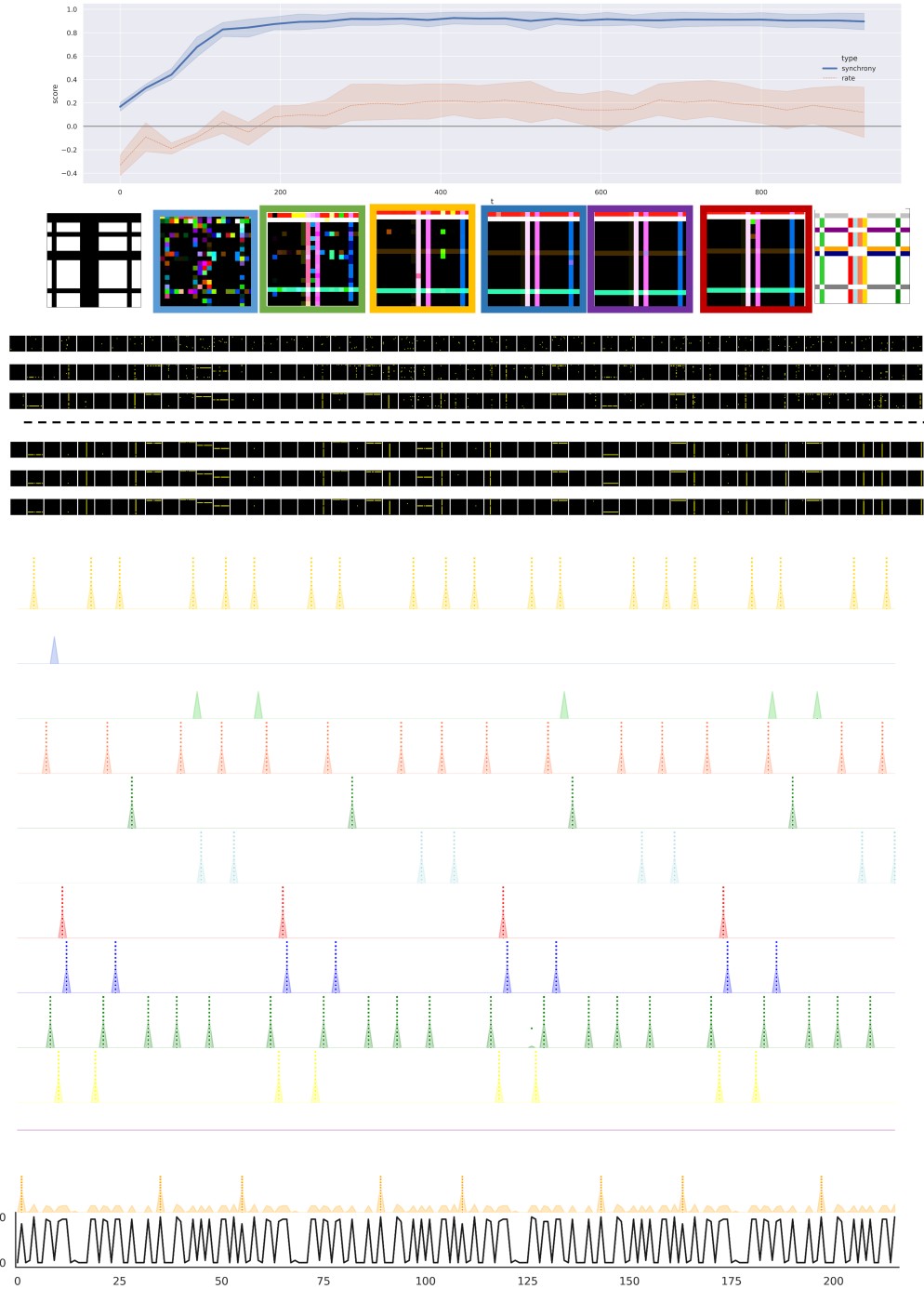

Figure 16: Visualization of temporal binding in Bars (i) AMI=84.5. Same settings as Fig2 in the main text. From the top to bottom: synchrony score and rate score; evolution of temporal structure indicated by color (left most–input; right most–ground truth middle–evolution of temporal structure from left to right); spiking pattern in initial phase; spiking pattern in convergent phase; spike recording in convergent phase.

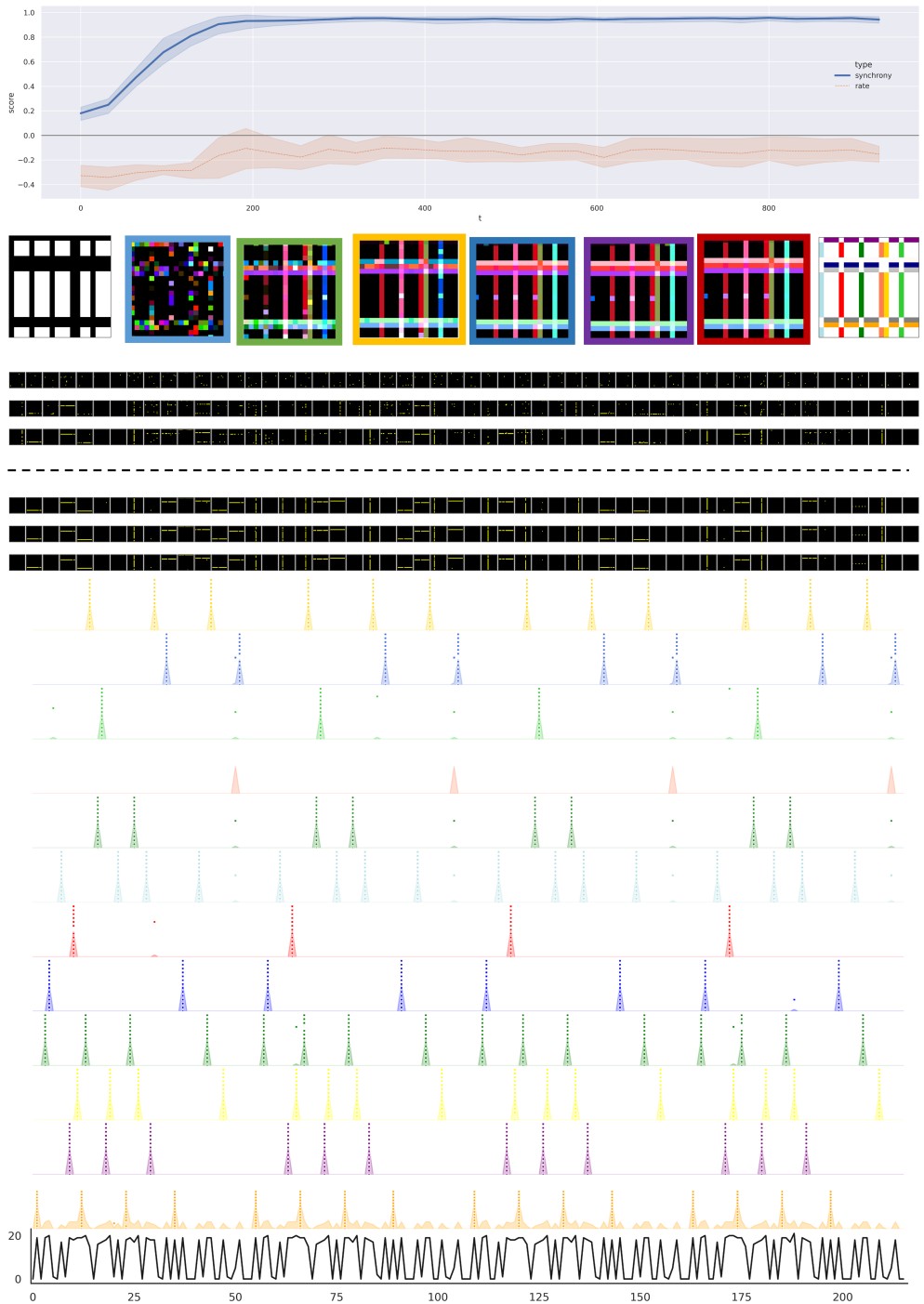

Figure 17: Visualization of temporal binding in Bars (ii) AMI=97.7. Same settings as Fig2 in the main text. From the top to bottom: synchrony score and rate score; evolution of temporal structure indicated by color (left most–input; right most–ground truth middle–evolution of temporal structure from left to right); spiking pattern in initial phase; spiking pattern in convergent phase; spike recording in convergent phase.

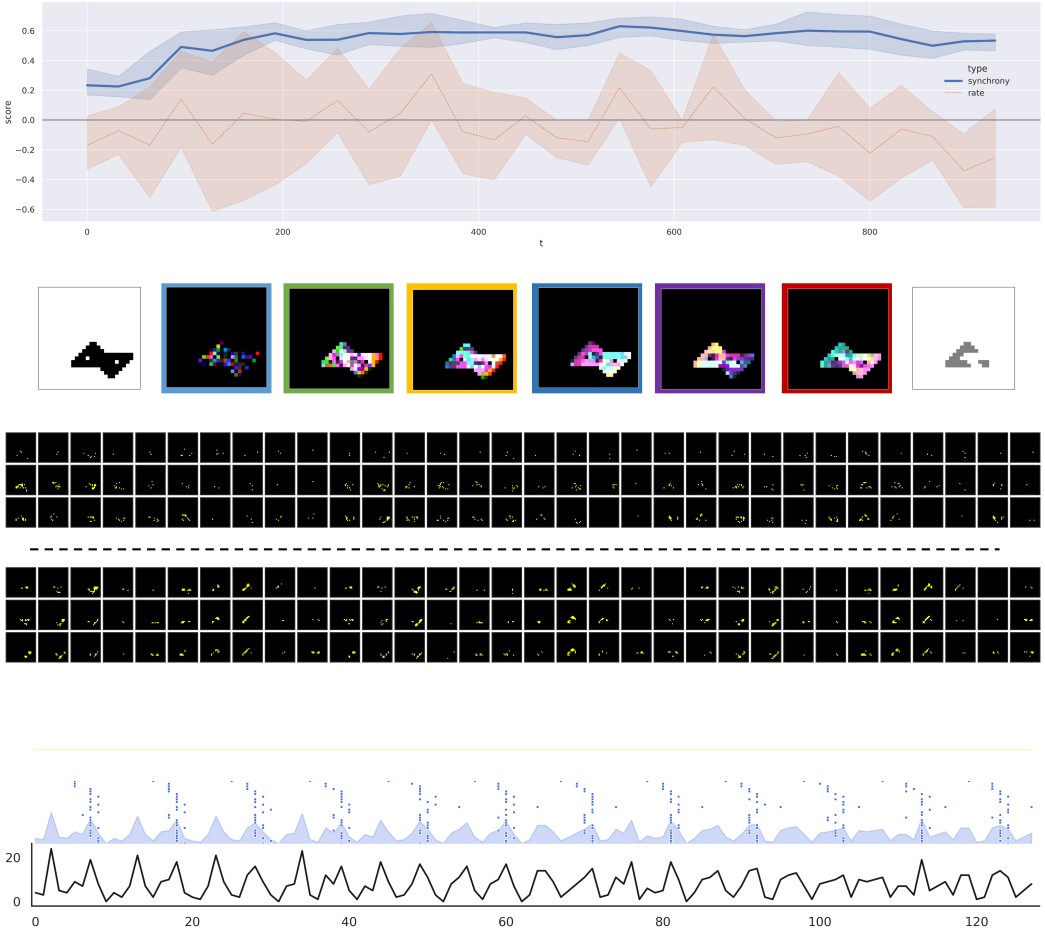

Figure 18: Visualization of temporal binding in Shapes (i) AMI=0. Same settings as Fig2 in the main text. From the top to bottom: synchrony score and rate score; evolution of temporal structure indicated by color (left most–input; right most–ground truth middle–evolution of temporal structure from left to right); spiking pattern in initial phase; spiking pattern in convergent phase; spike recording in convergent phase.

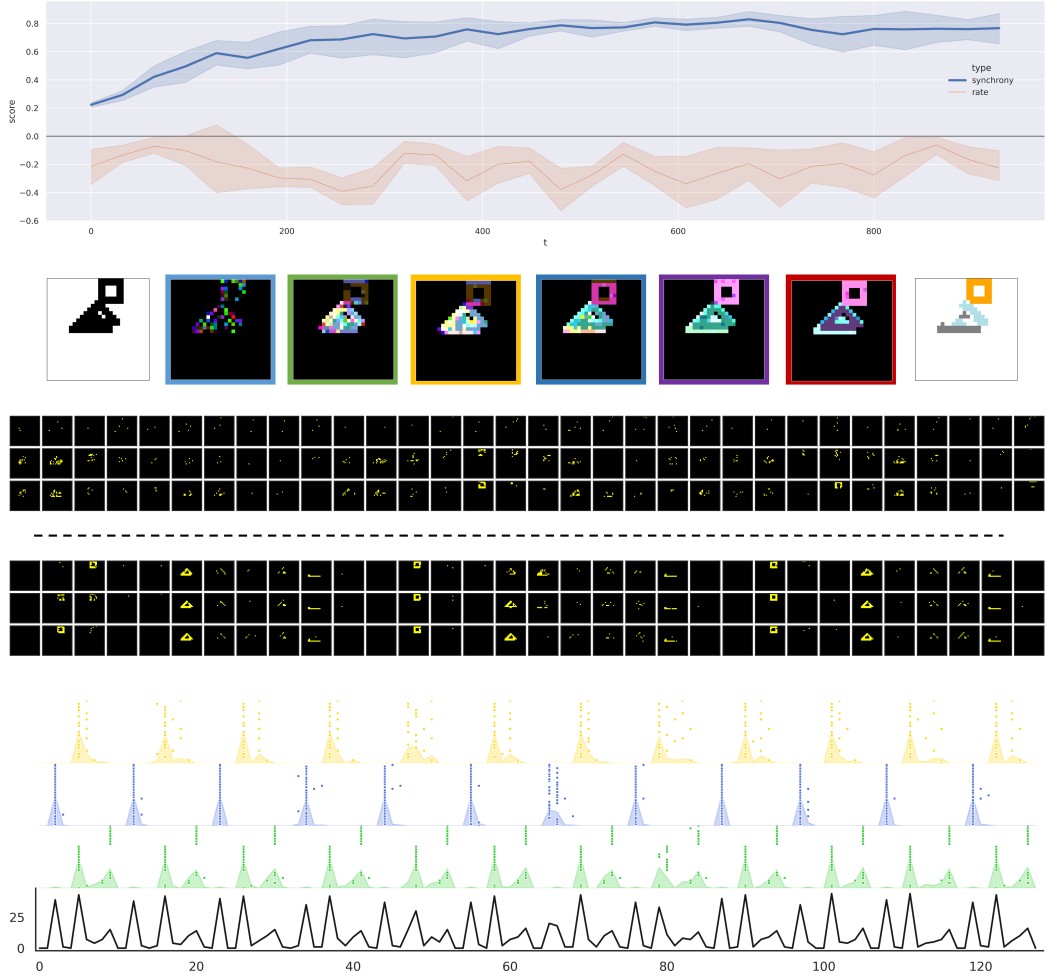

Figure 19: Visualization of temporal binding in Shapes (ii) AMI=81.6. Same settings as Fig2 in the main text. From the top to bottom: synchrony score and rate score; evolution of temporal structure indicated by color (left most–input; right most–ground truth middle–evolution of temporal structure from left to right); spiking pattern in initial phase; spiking pattern in convergent phase; spike recording in convergent phase.

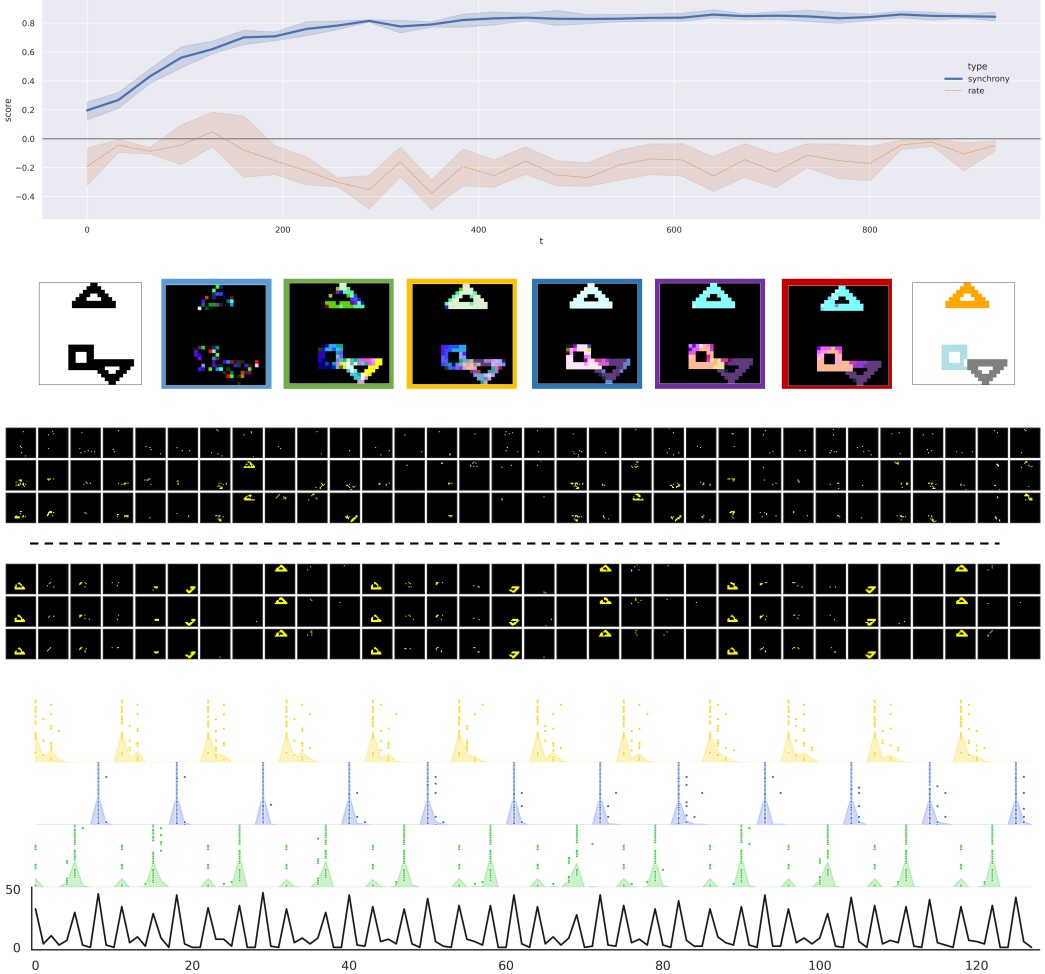

Figure 20: Visualization of temporal binding in Shapes (iii) AMI=89.0. Same settings as Fig2 in the main text. From the top to bottom: synchrony score and rate score; evolution of temporal structure indicated by color (left most–input; right most–ground truth middle–evolution of temporal structure from left to right); spiking pattern in initial phase; spiking pattern in convergent phase; spike recording in convergent phase.

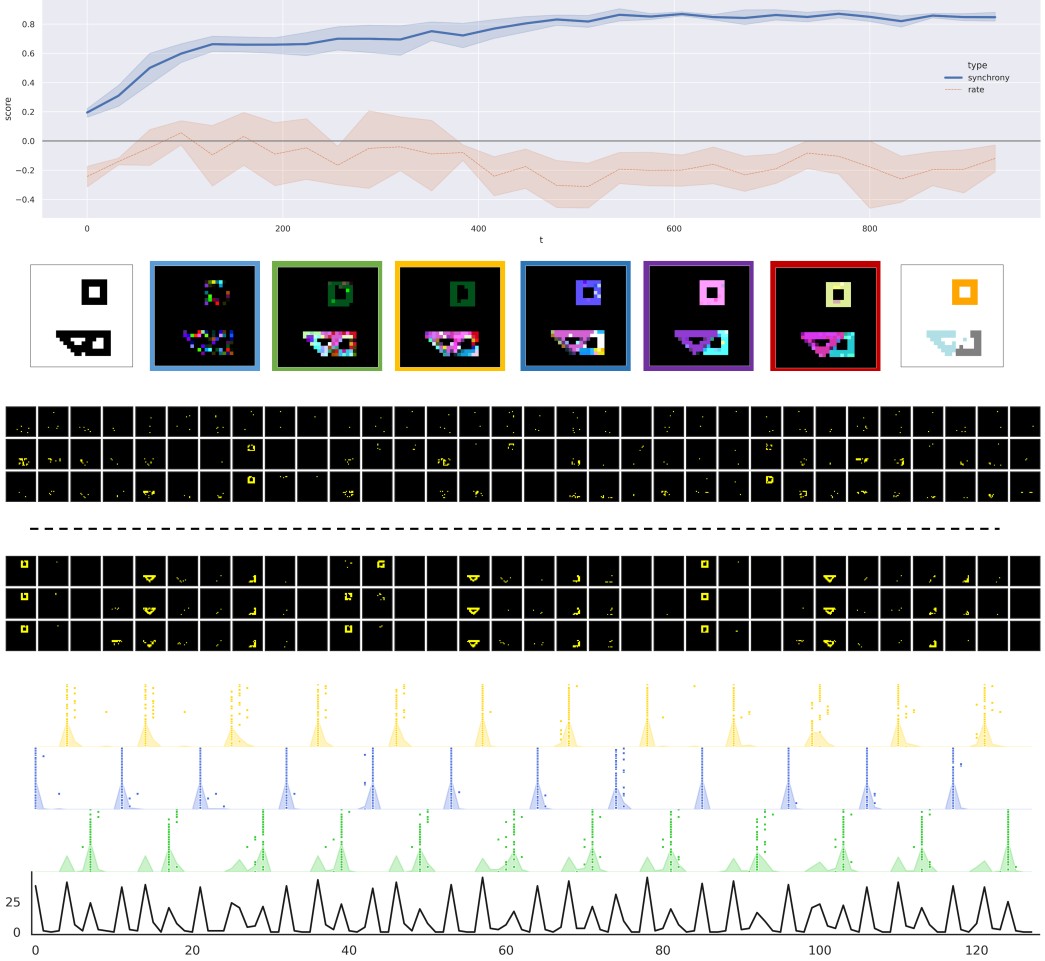

Figure 21: Visualization of temporal binding in Shapes (iii) AMI=98.9. Same settings as Fig2 in the main text. From the top to bottom: synchrony score and rate score; evolution of temporal structure indicated by color (left most–input; right most–ground truth middle–evolution of temporal structure from left to right); spiking pattern in initial phase; spiking pattern in convergent phase; spike recording in convergent phase.

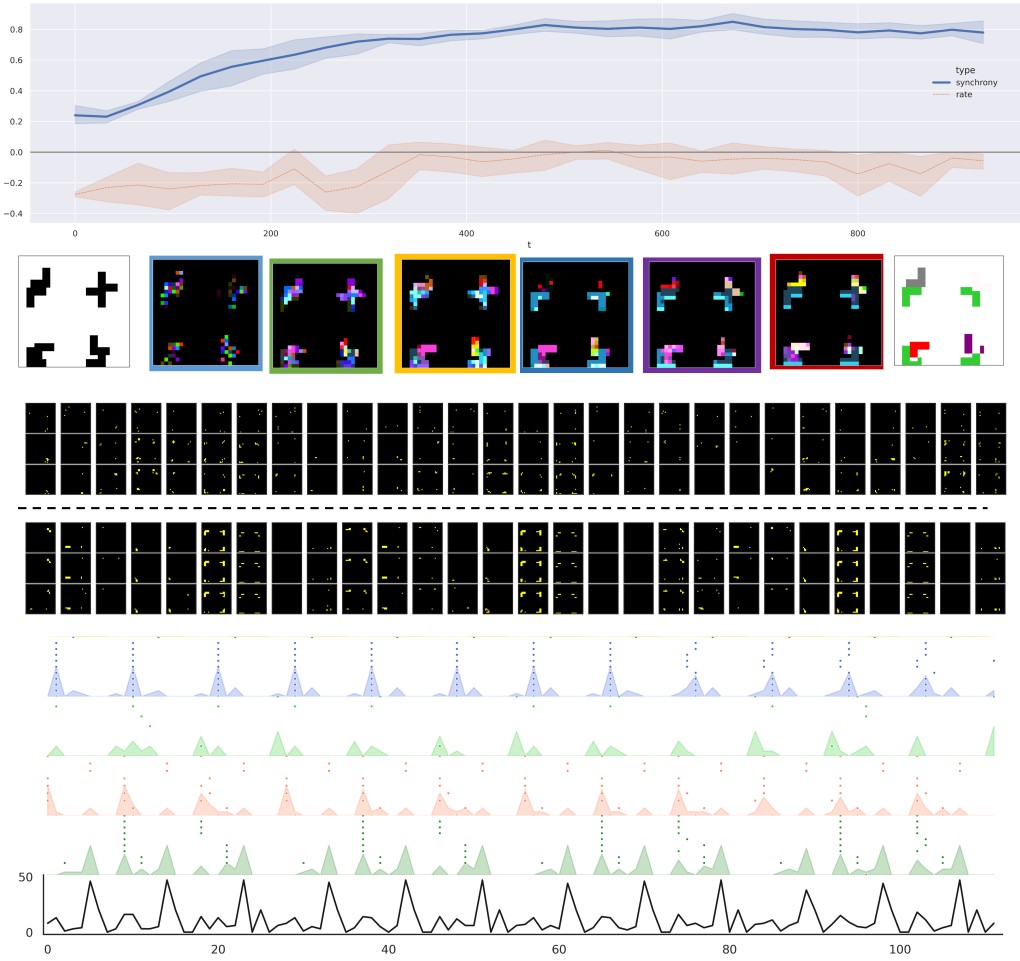

Figure 22: Visualization of temporal binding in Corners (i) AMI=43.2. Same settings as Fig2 in the main text. From the top to bottom: synchrony score and rate score; evolution of temporal structure indicated by color (left most–input; right most–ground truth middle–evolution of temporal structure from left to right); spiking pattern in initial phase; spiking pattern in convergent phase; spike recording in convergent phase.

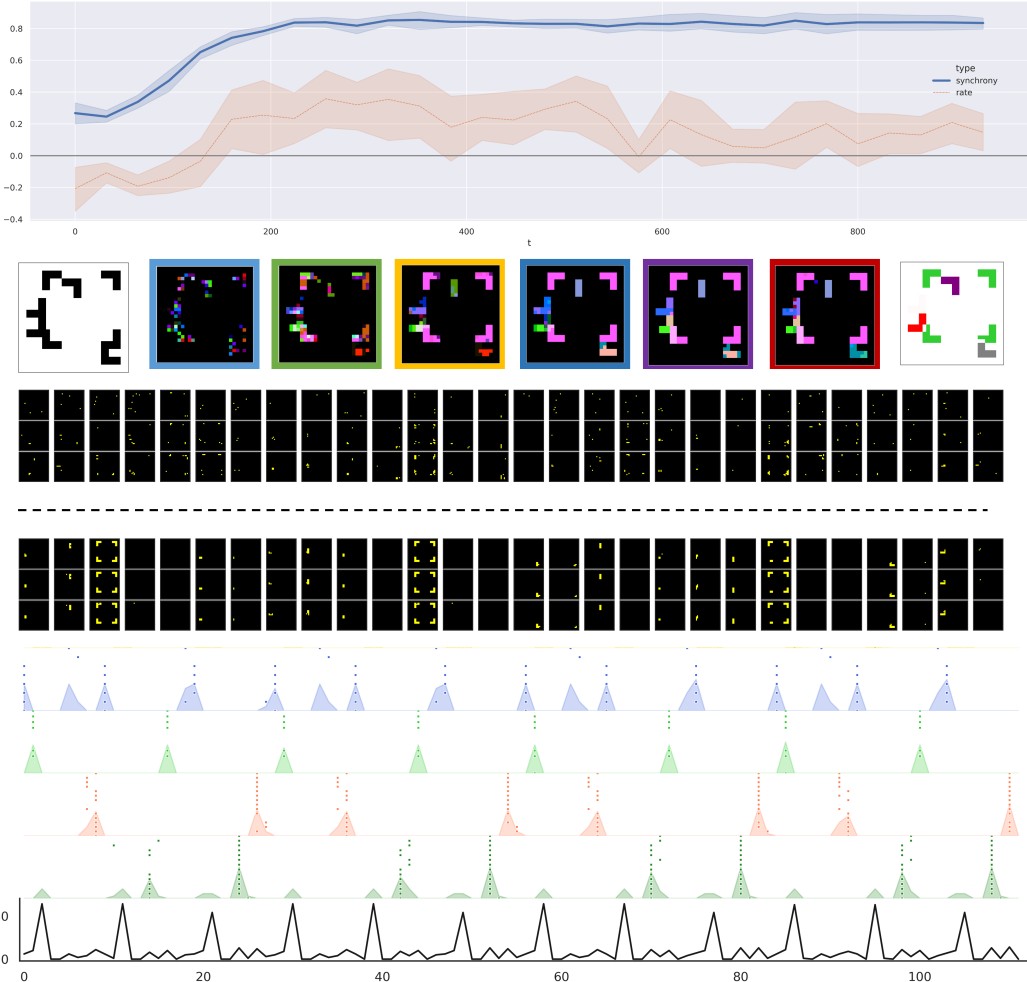

Figure 23: Visualization of temporal binding in Corners (ii) AMI=73.2. Same settings as Fig2 in the main text. From the top to bottom: synchrony score and rate score; evolution of temporal structure indicated by color (left most–input; right most–ground truth middle–evolution of temporal structure from left to right); spiking pattern in initial phase; spiking pattern in convergent phase; spike recording in convergent phase.

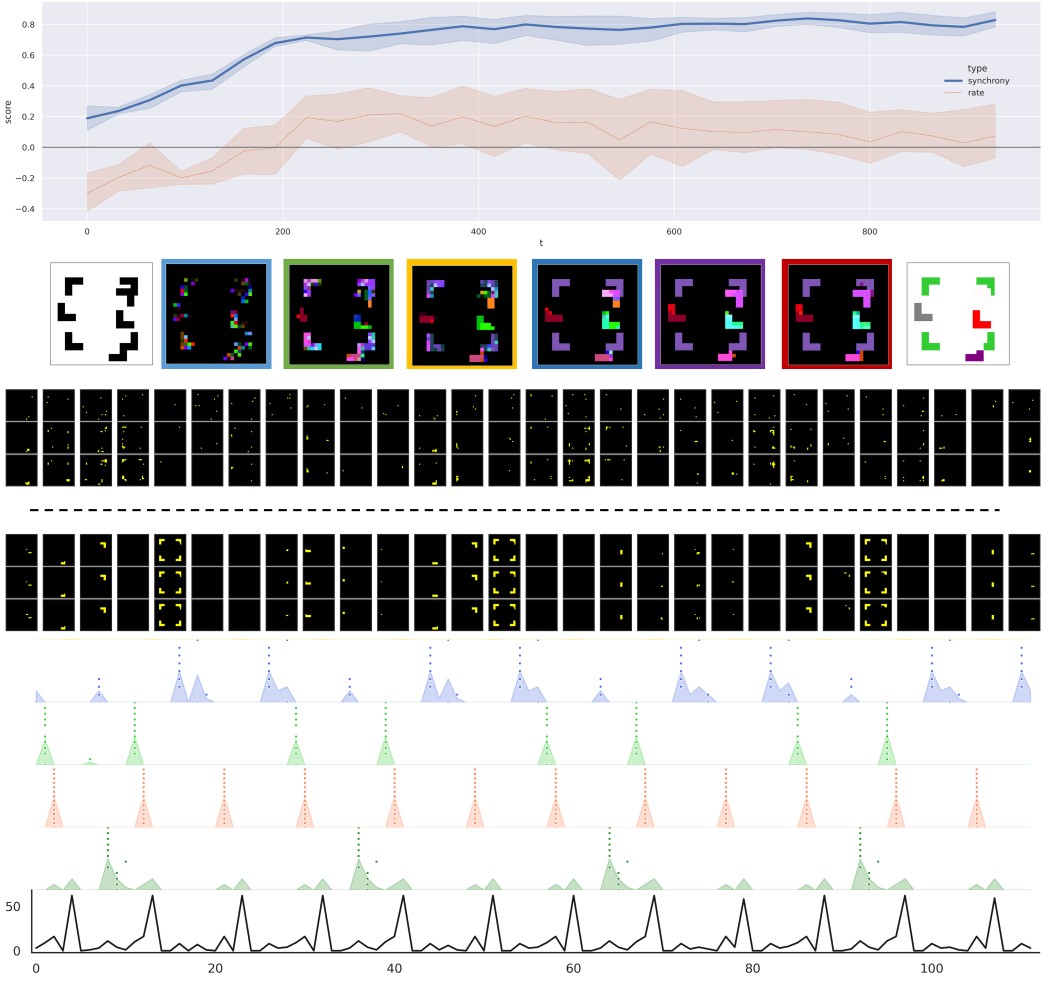

Figure 24: Visualization of temporal binding in Corners (vi) AMI=85.3. Same settings as Fig2 in the main text. From the top to bottom: synchrony score and rate score; evolution of temporal structure indicated by color (left most–input; right most–ground truth middle–evolution of temporal structure from left to right); spiking pattern in initial phase; spiking pattern in convergent phase; spike recording in convergent phase.

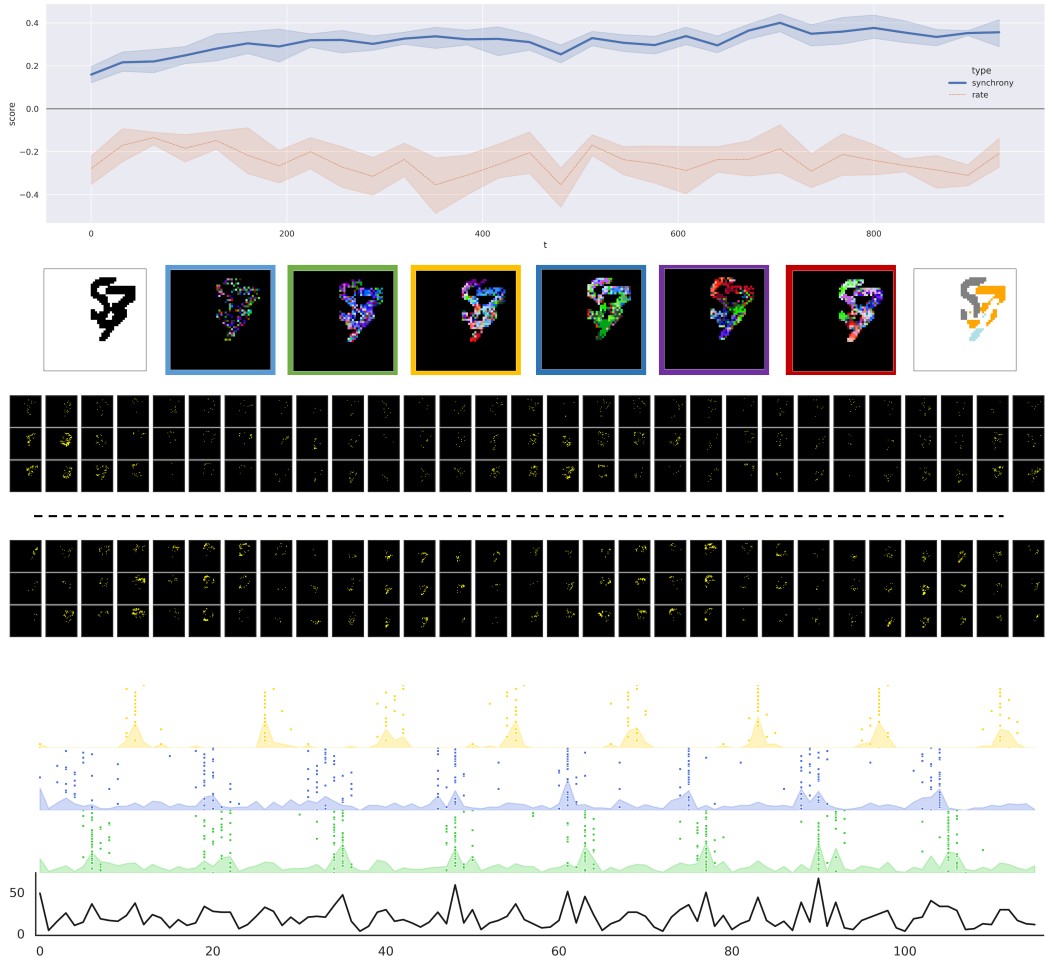

Figure 25: Visualization of temporal binding in Multi-MNIST (failure) (i). Same settings as Fig2 in the main text. From the top to bottom: synchrony score and rate score; evolution of temporal structure indicated by color (left most–input; right most–ground truth middle–evolution of temporal structure from left to right); spiking pattern in initial phase; spiking pattern in convergent phase; spike recording in convergent phase.

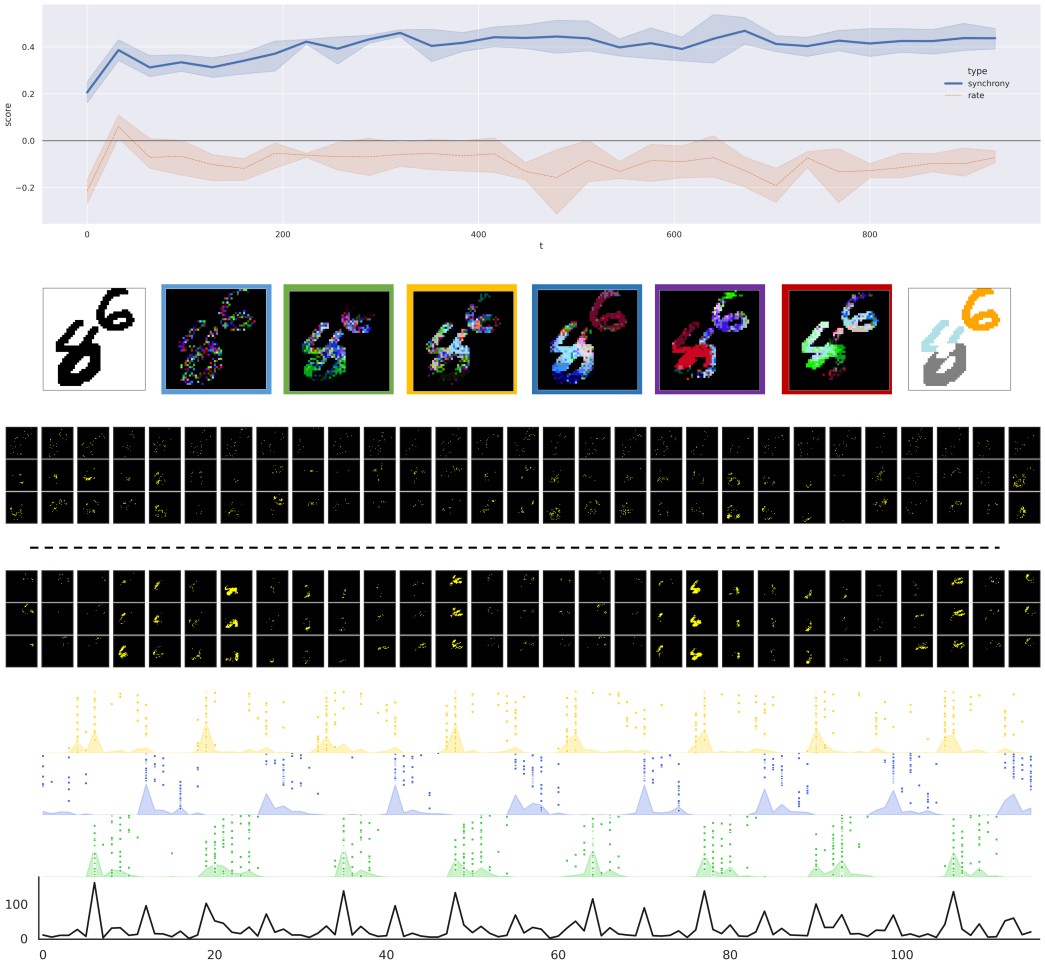

Figure 26: Visualization of temporal binding in Multi-MNIST AMI=43.3. Same settings as Fig2 in the main text. From the top to bottom: synchrony score and rate score; evolution of temporal structure indicated by color (left most–input; right most–ground truth middle–evolution of temporal structure from left to right); spiking pattern in initial phase; spiking pattern in convergent phase; spike recording in convergent phase.

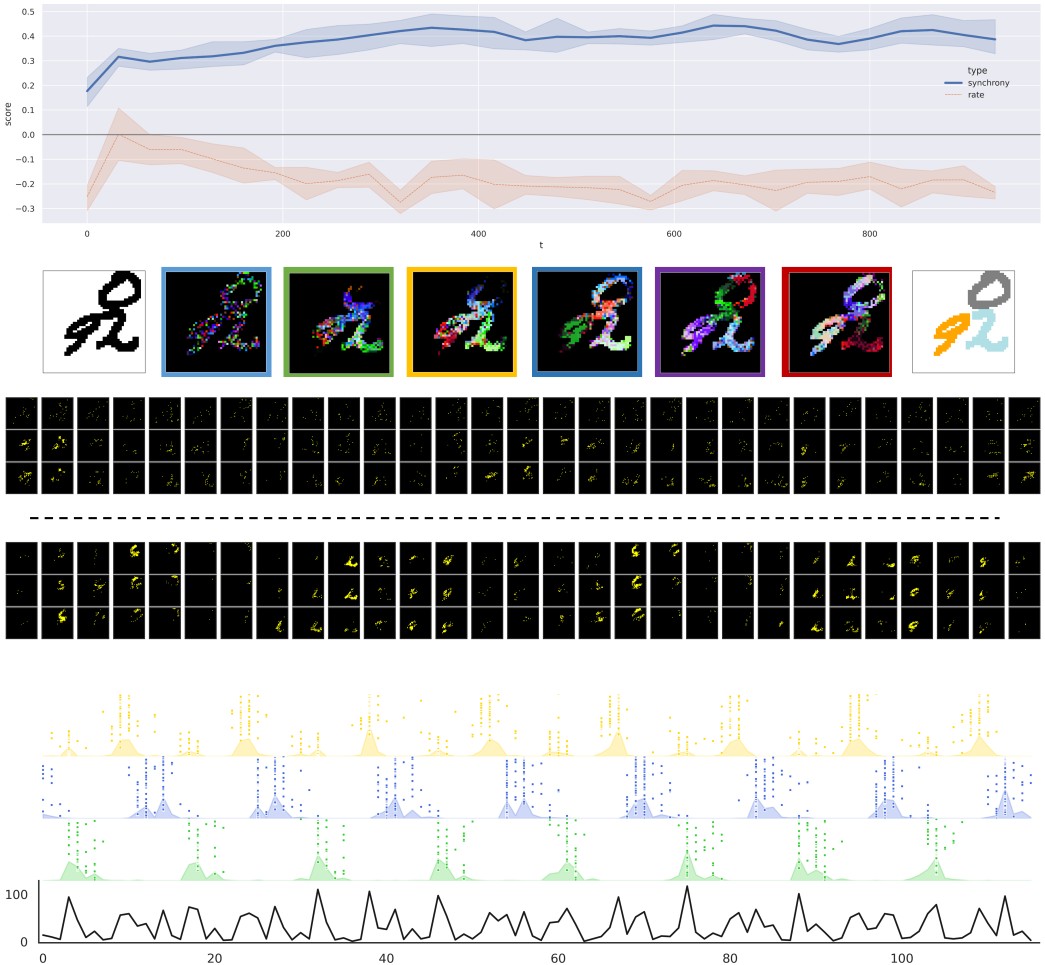

Figure 27: Visualization of temporal binding in Multi-MNIST (failure) (ii) AMI=67.9. Same settings as Fig2 in the main text. From the top to bottom: synchrony score and rate score; evolution of temporal structure indicated by color (left most–input; right most–ground truth middle–evolution of temporal structure from left to right); spiking pattern in initial phase; spiking pattern in convergent phase; spike recording in convergent phase.

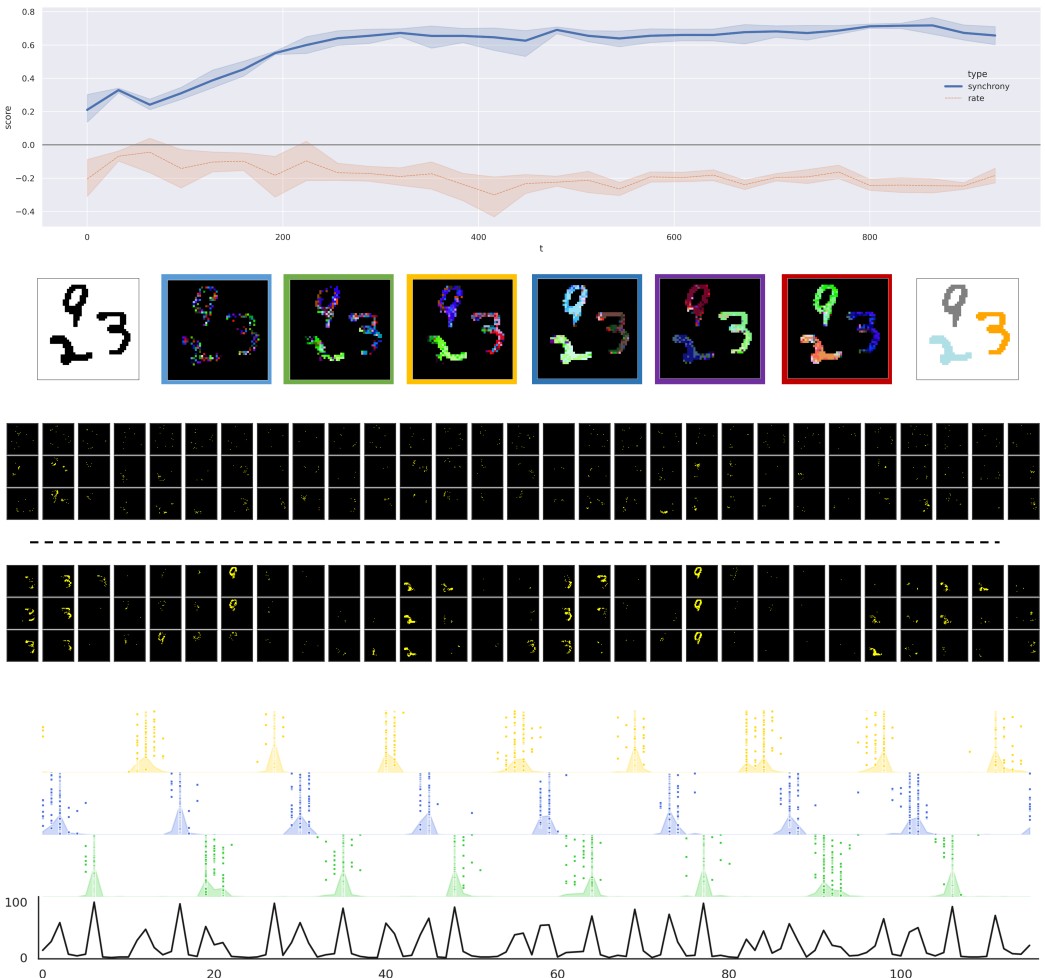

Figure 28: Visualization of temporal binding in Multi-MNIST (iii) AMI=94.4. Same settings as Fig2 in the main text. From the top to bottom: synchrony score and rate score; evolution of temporal structure indicated by color (left most–input; right most–ground truth middle–evolution of temporal structure from left to right); spiking pattern in initial phase; spiking pattern in convergent phase; spike recording in convergent phase.

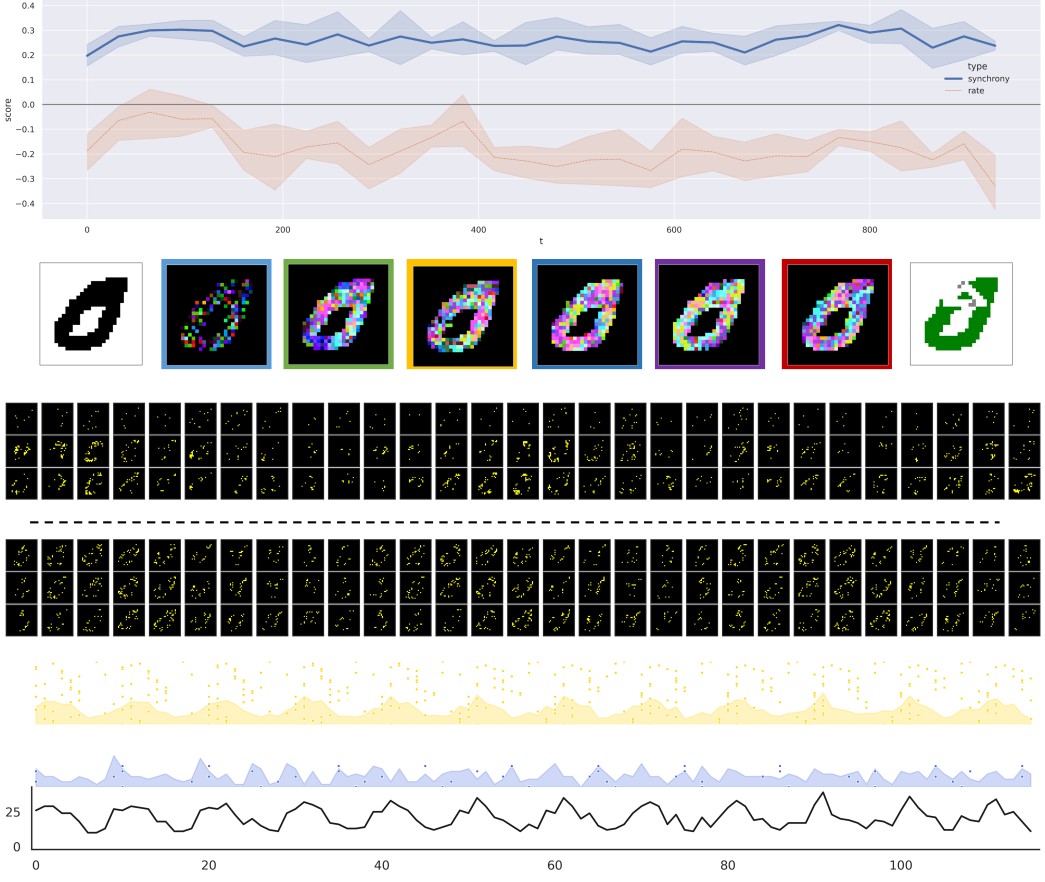

Figure 29: Visualization of temporal binding in MNIST+Shape (failure) (i). Same settings as Fig2 in the main text. From the top to bottom: synchrony score and rate score; evolution of temporal structure indicated by color (left most–input; right most–ground truth middle–evolution of temporal structure from left to right); spiking pattern in initial phase; spiking pattern in convergent phase; spike recording in convergent phase.

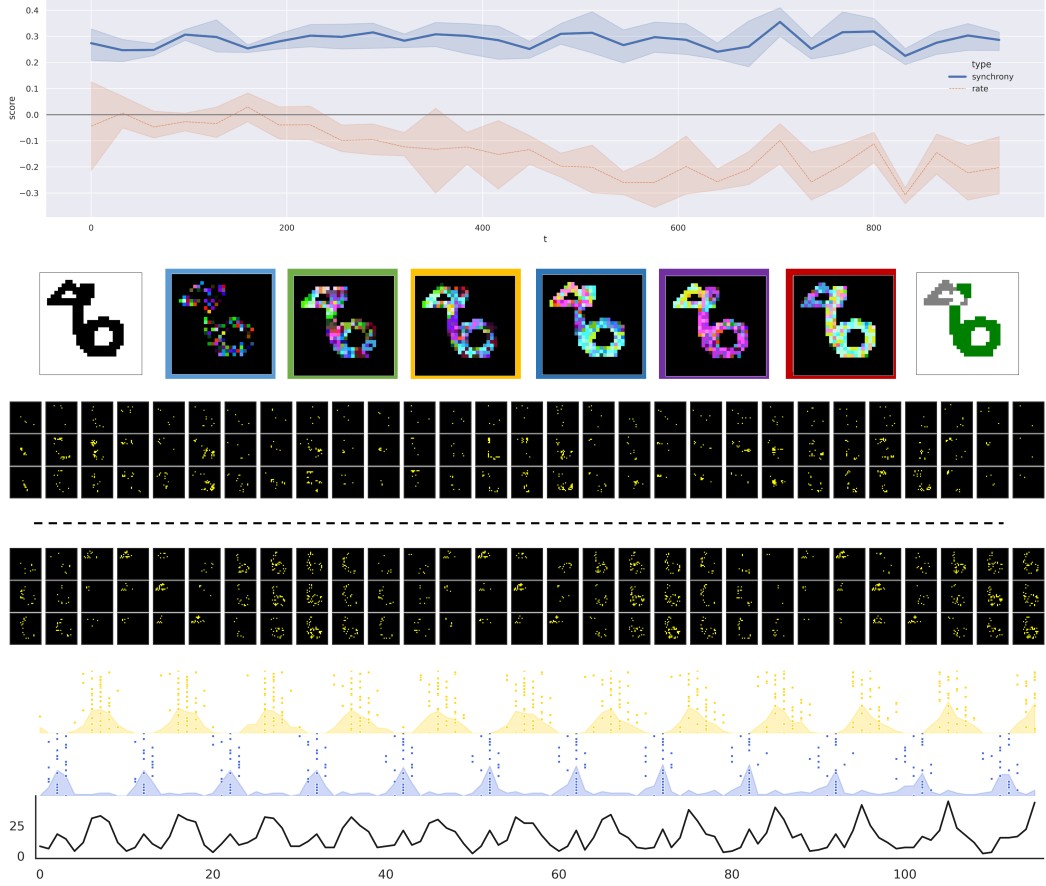

Figure 30: Visualization of temporal binding in MNIST+Shape. AMI=76.4. Same settings as Fig2 in the main text. From the top to bottom: synchrony score and rate score; evolution of temporal structure indicated by color (left most–input; right most–ground truth middle–evolution of temporal structure from left to right); spiking pattern in initial phase; spiking pattern in convergent phase; spike recording in convergent phase.

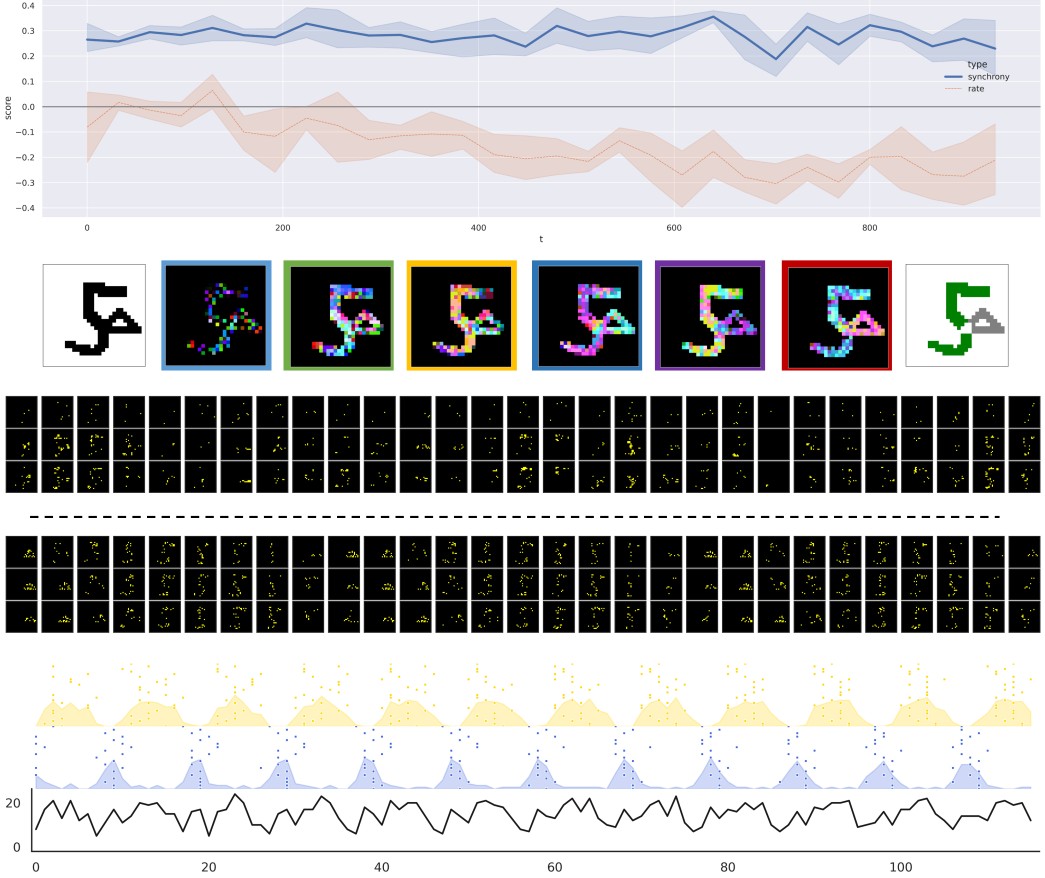

Figure 31: Visualization of temporal binding in MNIST+Shape. AMI=88.7. Same settings as Fig2 in the main text. From the top to bottom: synchrony score and rate score; evolution of temporal structure indicated by color (left most–input; right most–ground truth middle–evolution of temporal structure from left to right); spiking pattern in initial phase; spiking pattern in convergent phase; spike recording in convergent phase.

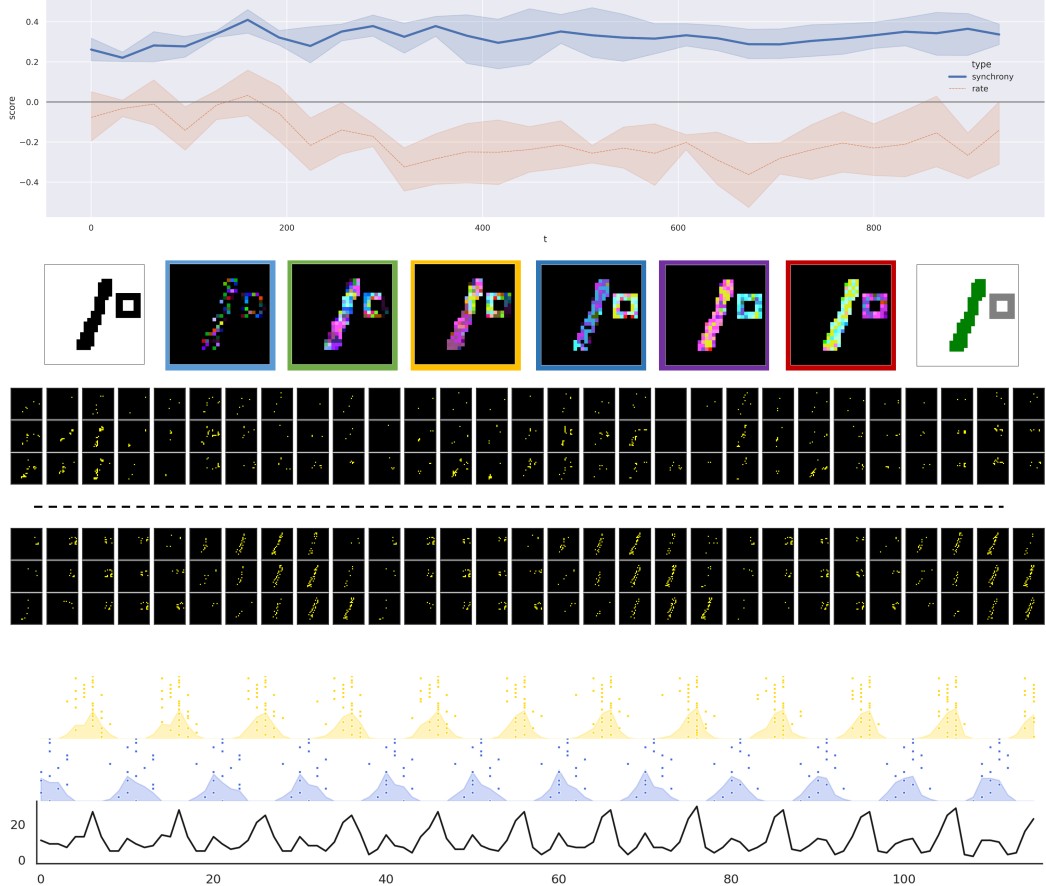

Figure 32: Visualization of temporal binding in MNIST+Shape. AMI=100. Same settings as Fig2 in the main text. From the top to bottom: synchrony score and rate score; evolution of temporal structure indicated by color (left most–input; right most–ground truth middle–evolution of temporal structure from left to right); spiking pattern in initial phase; spiking pattern in convergent phase; spike recording in convergent phase.

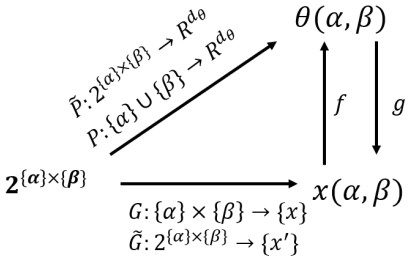

Figure 33: proof2

## A.11    Proof

This section contains assumptions and detailed derivations of the properties demonstrated in the Model Section. Here, we first clarify several assumptions:

1. The single object representation $x_n$ is binary, the disturbance variable $\epsilon$ is also binary. And the disturbance is restricted to random removal of bits of $x_n$ instead of adding bits (consistent with the training in the experiment)

2. By "perfectly", we assume the $f_{DAE}$ not only (a) learns a perfect denoising map $f_{DAE}$: $\forall \epsilon \in \{0,1\}, s.t. \|\epsilon\| < \delta$, and $\forall x_n \in \{x_n\}$, then, $x_n = f_{DAE}(x_n - \epsilon)$; but also (b) learns to represent the stimulus $\{x_n\}$ in the most efficient way, with its latent layer learns a perfect disentangled representation of the generative factors.

3. We assume all the "+","−" on $x_n$ are all constraint in binary space, which means: $1+1 = 1+0 = 0+1 = 1, 0+0 = 0, 1-1 = 0 = 0-1 = 0-0 = 0, 1-0 = 1$.

The proof of Proposition1 is straightforward and hold by definition of DAE.

**Proposition1**

*proof.* (1) Assume the maximum salt & pepper noise strength is set to be $\delta$ in the training process. Then $\forall \epsilon \in \{0,1\}, s.t. \|\epsilon\| < \delta, \forall x_n \in \{x_n\}$, we have, $x_n = f_{DAE}(x_n - \epsilon)$ □

*proof.* (2) Given $\alpha < 1$, $\|f_{DAE}(x_n - \epsilon) - x_n\| = \|x_n - x_n\| = 0 < \alpha\epsilon = \alpha\|x_n - \epsilon - x_n\|$ □

The proof of the second Proposition is based on the fact that a perfect DAE can suffer from the binding problem, so that it can not reconstruct a superposed input. More formally:

**Proposition2**

*proof.* By assuming learning a perfect disentangled latent representation, we have two interchangeable paths in the diagram in Fig.33 We first describe each part in the diagram.

(a)let $f_{DAE} = g \circ f$, the f and g are the mapping realized by the encoder and decoder respectively. $\theta$ is the latent representation in the DAE.

(b) $\{\alpha\}, \{\beta\}$ are the set of values of different types of generative factors of the objects (eg. $\{\alpha\}$={up-triangle,down-triangle,square}; $\{\beta\} = \{(x,y)|x \in [0, 28], y \in [0, 28]\}$ in Shapes dataset, here we only consider two types of features for simplicity). Given set X, $2^X$ is the power set of X. $2^{\{\alpha\}\times\{\beta\}}$ contains all possible combinations of features to generate a scene with single or multiple objects.

(c) We assume there is an implicit single object generation process formulated by

$$G : \{\alpha\} \times \{\beta\} \to \{x\}.$$

Further, the generation of superposed object x' is induced by $G$,

$$\widetilde{G} : 2^{\{\alpha\}\times\{\beta\}} \to x'$$

and $\widetilde{G}$ has the property:

$$\forall A, B \in 2^{\{\alpha\}\times\{\beta\}}, \widetilde{G}(A \cup B) = \widetilde{G}(A) + \widetilde{G}(B)$$

$$\forall(\alpha, \beta) \in \{\alpha\} \times \{\beta\}, \widetilde{G}(\alpha, \beta) = G(\alpha, \beta)$$

(d) To describe disentangled representation, we assume there is an imaginary alternative mapping $P$,

$$P : \{\alpha\} \cup \{\beta\} \to R^{d_\theta}$$

where $d_\theta$ is the dimension of latent space. $P$ has the property,

$$\forall a, b \in \{\alpha\} \cup \{\beta\}, a \neq b \Rightarrow \langle P(a), P(b) \rangle = 0$$

$\widetilde{P}$ is induced by $P$,

$$\widetilde{P} : 2^{\{\alpha\} \times \{\beta\}} \to R^{d_\theta}$$

$\widetilde{P}$ has the property,

$$\forall A, B \in 2^{\{\alpha\} \times \{\beta\}}, \widetilde{P}(A \cup B) = \widetilde{P}(A) + \widetilde{P}(B)$$

$$\forall(\alpha, \beta) \in \{\alpha\} \times \{\beta\}, \widetilde{P}(\alpha, \beta) = P(\alpha) + P(\beta)$$

(e) Since DAE can perfectly reconstruct the input $x(\alpha, \beta)$, we have $\forall(\alpha, \beta) \in \{\alpha\} \times \{\beta\}, \theta(\alpha, \beta) = f(x(\alpha, \beta))$ and $g(\theta(\alpha, \beta)) = x(\alpha, \beta)$

OK, all elements in the diagram have been defined.

Perfect disentanglement of objects tells us that the representation of features contained in a scene is independent of the specific object that instantiates the feature. The former is the lower path: from generative factor, to object by generation process and finally to latent representation by encoding. The latter is imaginary upper path: directly from the set of feature combinations to the representation, described by $P$ and $\widetilde{P}$. Thus, disentanglement claim that the upper path and lower path are interchangeable. Start from this, we can generally construct at least one example x' that satisfy the Proposition2: $\|f_{DAE}(x') - x'\| > 0$. This is the direct outcome of the binding problem.

In this spirit, consider two objects pairs: $(x(\alpha_1, \beta_1), x(\alpha_2, \beta_2))$ and $(x(\alpha_1, \beta_2), x(\alpha_2, \beta_1))$

then superposition catastrophe will prevent the accurate reconstruction of both superposed cases.

From the diagram we have,

$$g \circ f(x(\alpha_1, \beta_1) + x(\alpha_2, \beta_2)) = g \circ [P(\alpha_1) + P(\beta_1) + P(\alpha_2) + P(\beta_2)]$$
$$g \circ f(x(\alpha_1, \beta_2) + x(\alpha_2, \beta_1)) = g \circ [P(\alpha_1) + P(\beta_2) + P(\alpha_2) + P(\beta_1)]$$

if

$$g \circ f(x(\alpha_1, \beta_1) + x(\alpha_2, \beta_2)) = x(\alpha_1, \beta_1) + x(\alpha_2, \beta_2)$$

and

$$g \circ f(x(\alpha_1, \beta_2) + x(\alpha_2, \beta_1)) = x(\alpha_1, \beta_2) + x(\alpha_1, \beta_2)$$

hold at the same time. Then,

$$
\begin{aligned}
x(\alpha_1, \beta_1) &+ x(\alpha_2, \beta_2) \\
&= g \circ f(x(\alpha_1, \beta_1) + x(\alpha_2, \beta_2)) \\
&= g \circ [P(\alpha_1) + P(\beta_1) + P(\alpha_2) + P(\beta_2)] \\
&= g \circ [P(\alpha_1) + P(\beta_2) + P(\alpha_2) + P(\beta_1)] \\
&= g \circ f(x(\alpha_1, \beta_2) + x(\alpha_2, \beta_1)) \\
&= x(\alpha_1, \beta_2) + x(\alpha_1, \beta_2)
\end{aligned}
\tag{1}
$$

However,

$$x(\alpha_1, \beta_1) + x(\alpha_2, \beta_2) \neq x(\alpha_1, \beta_2) + x(\alpha_1, \beta_2)$$

Contradiction arises and we find at least one x' in the proposition2

$\square$

The proof of Proposition3 and Proposition4 is straightforward by construction.

**Proposition3**

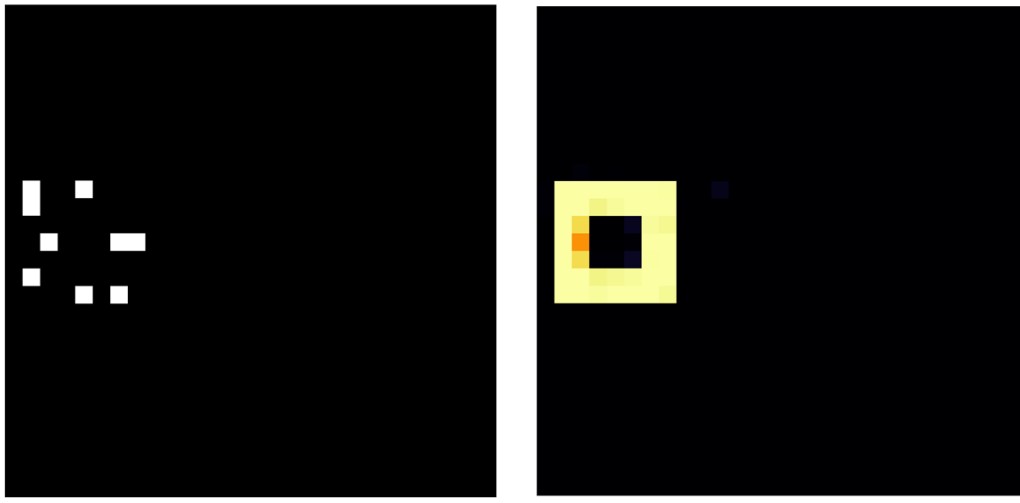

Figure 34: The experimental evidence of proposition1. left: input image with single objects; right: output of DAE trained on single object dataset. It can be seen that output is almost identical to input for superposed input.

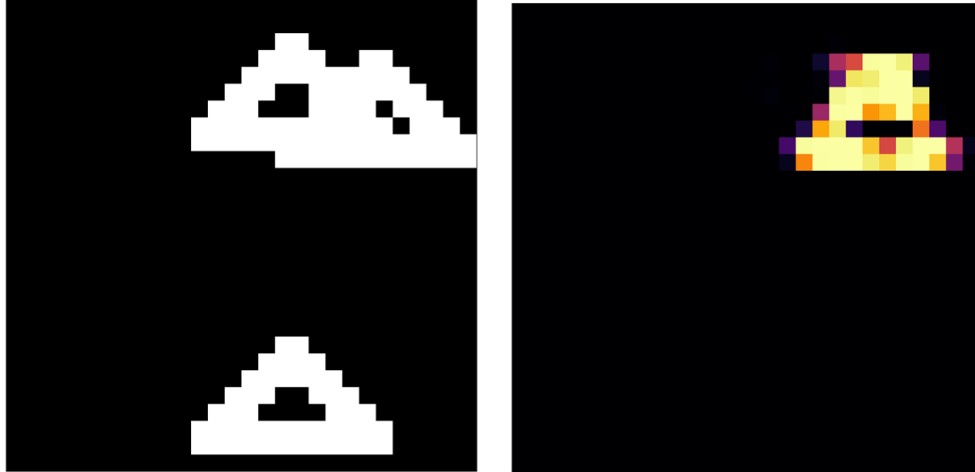

Figure 35: The experimental evidence of proposition2. left: input image with multiple objects; right: output of DAE trained on single object dataset. It can be seen that output is not identical to input for superposed input.

*proof.* Since $\tau_{delay} \in N$ in our case.

if $x(0) = x(1) = x(2) = ... = x(\tau_{delay}) = x_n$,

then, $\forall t = n \cdot \tau_{delay} + m, (m < \tau_{delay})$

$$\begin{aligned} x(t+1) &= x(n \cdot \tau_{delay} + m + 1) \\ &= f_{DAE}(x(m+1)) \\ &= x_{m+1} = x_n \end{aligned} \quad (2)$$

Thus, $x(t) \equiv x_n$ is a possible solution of the dynamics.

Since $f(x_n - \epsilon) = f(x_n) = x_n$, the trajectory is attractive:

let $\widetilde{x}(t) = (x(t), x(t+1), ..., x(t+\tau_{delay}))$, and $\widetilde{x}_n = (x_n, x_n, ..., x_n)$, if $\widetilde{x}(t) = \widetilde{x}_n$

then under small disturbance, the fixed point will be arrived after $\tau_{delay}$:

$$\widetilde{x}(t + \tau_{delay}) = f_{DAE}(\widetilde{\boldsymbol{x}}(t) - \boldsymbol{\epsilon}) = \boldsymbol{x_n}$$

$\square$

**Proposition4**

*proof.* (1) If $x(t) = x(t+1) = x_n$, then due to refraction, $x(t + \tau_{delay}), x(t + \tau_{delay} + 1)$ can not be $x_n$ at the same time. Because, if $x(t + \tau_{delay}) = 0$, then $rfr = 0$ at $t + \tau_{delay} + 1$, thus, $x(t + \tau_{delay} + 1) = f_{DAE}(0) \neq x_n$. Thus, $x(t) \equiv x_n$ does not hold in this case.

(2) Given the number of superposed objects is $K$. Due to time dimension is continuous (in practice, it is discrete but of high resolution), assume $K << \tau_{delay}$

Then, consider the following solution. $w = [\tau_{delay}/K] > 0$

$$x(m \cdot \tau_{delay} + i \cdot w) = x_{n_i}, m \in N, 0 < i < K \quad (3)$$

Otherwise, $x(t) = 0$. Since $\forall t_1, t_2, 0 < t_2 - t_1 < \tau_{rfr} \Rightarrow x(t_1) = x(t_2) = 0$ or $x(t_1) \neq x(t_2)$, no interference among x(t) by $rfr$. Thus obviously, x(t) satisfy the equation in proposition 4.

In this way, we construct a periodic solution of x(t) that represent K object alternatively. Similar to the proof of proposition 3, the solution is attractive.

$\square$

We are aware that the proof above depend on very strong assumptions (like perfect DAE) and extremely idealized situation. Thus, reality may deviate from the proved result. However, we find experimentally that these results are generally hold in reality.

For proposition 1 3, we can see from experiments that single objects can indeed be a convergent state, eg. Fig.34 for proposition 1 and Fig.4 for proposition 3

For proposition 2, in general, the mapping that can reconstruct the superposed input can certainly exist without further assumptions (eg. identity map). However, in practise, it is highly possible that the superposed input will not be reconstructed, simply because the decoder has a strong prior of single objects so that the range of $g$ is constraint in single-object related patterns Fig.35. Besides, the denoising auto-encoder can avoid learning an identity map.

For proposition 4, we can see from experiments (Section A.10) that alternative states is indeed emerged.