# OpenReview forum: "Dance of SNN and ANN: Solving binding problem by combining spike timing and reconstructive attention"
_NeurIPS.cc/2022/Conference — NeurIPS 2022 Accept_

### Official Review · Reviewer_1EXz · 2022-07-06

**Rating:** 5
**Confidence:** 2
**Soundness:** 2 fair
**Presentation:** 3 good
**Contribution:** 3 good

**Summary:**

In this article, the authors propose to combine Artificial Neural Network (ANN) and Spiking Neural Network (SNN) to solve the binding problem. In their model the ANN is a Denoising Autoencoder (DAE) that has been trained to remove salt and pepper noise on images of single object. The SNN is a coincidence detector that is feeding the encoder of the DAE. In return, the decoder of the DAE is providing an attention map used by the SNN. This is defining a dynamical loop between SNN and ANN. The authors show that the combination of both (ANN and SNN) allows to solve the binding problem on 5 different datasets (Bars, Shapes, Corners, Multi-MNIST, MNIST+Shape).


**Questions:**

* The training procedure is never really detailed. I have understood that only the ANN was trained (as a DAE trained to remove salt and pepper noise in images of single objects), not the SNN. Is it correct ? If so could you explain if the ANN is trained separately ? Including a proper mathematical formulation of the loss function might improve the understanding of the training procedure

* In general, the value of the hyper-parameters are not detailed. Sometimes, I don't even know if they are parameters (updated through a learning process) or hyper-parameters (i.e. set by hand). For example, what about p in eq 3 (Is it learned ) ? I would suggest the authors to update their article to make this specific point clearer.

* In figure 3 (and in table 1), we observe that the performances are lower for Multi-MNIST (and MNIST + shape). This is not properly discussed in the article.

* In section 5: [1] should be added in the reference (this is one of the most recent model that solve the binding problem)

* The proposed network should be compared to other state of the art networks. For exemple other complex auto encoder [1] are known to also solve the binding problem.

Minor:
Line 107 : N is not defined (I guess this is the size of the image, but it should be explicitly said)

Line 115: peroid —> period

Line 142 dynamcs —> dynamics

Line 209: On the contrast —> "In Contrast" or "On the contrary"

Line 255 Equivarince —> Equivariance


[1] : Löwe, Sindy, et al. "Complex-Valued Autoencoders for Object Discovery." arXiv preprint arXiv:2204.02075 (2022).

**Limitations:**

The authors have properly discussed the limitations and the societal of their work in their article.

**Strengths And Weaknesses:**

The paper is clear and well written. To the best of my knowledge the type of model presented in this article is novel. Thus, this is a valuable contribution to the field. Another strength of the article lies in the method to evaluate the synchrony level (which is also a novelty)

The main weakness of this article is the lack of comparison with other related models. Even if the principle behind the presented model is novel, other models also claim to solve the binding problem (e.g. complex autoencoders in [1]...).

Overall, I would tend to accept the article but it is borderline because there is no comparison with other algorithms solving a similar tasks (so it is hard to say if the proposed model is bringing a better binding ! )

[1] : Löwe, Sindy, et al. "Complex-Valued Autoencoders for Object Discovery." arXiv preprint arXiv:2204.02075 (2022).

---

> ### Author Response · Authors · 2022-08-02
> **Response to reviewer4 (1EXz) part 1**
>
> # **Response**
>
> Thanks for your comments and appreciation. For concerns of comparison, we added in response to Q5
>
> ## Q1.
>
> In this paper, only DAE is trained (separately) to remove salt and pepper noise in images of single objects. The loss function is the binary cross-entropy loss between input image and reconstruction output of DAE. We have added more descriptions in SI A.4.
>
> However, we would like to highlight that the training scheme in this paper is just an example and there are many ways for training. For instance, **we have studied directly training the whole model (as an RNN) either on single-object dataset or on multiple-object dataset, with a similar denoising loss and the DASBE still learns to autonomously group the objects.**
>
> ## Q2.
>
> In this model, the only learnable parameters are the weights in the DAE. “p” in equation 3 is a hyperparameter (set as 0.5) but its exact value has little effect on binding as long as p<1.
>
> We have provided an illustration of hyper-parameters (and the value) in Table 2 SI in the latest SI and Alg1 in the main text.
>
> ## Q3.
>
> We explain it in three aspect, which is added in SI A.6
>
> 1. The MNIST has more **diverse** pattern thus is more challenging (A.6.1 SI). Eg. The denoising performance for MLP-DAE is different in five datasets (Bars/Shapes>Corners>Multi-MNIST>MNIST-Shape). We are convinced that the issue can be mitigated by introducing more advanced ANN models.
>
> 2. “**Overlap**” is more likely to happen and may have larger effects in MNIST-related datasets. In Multi-MNIST, occluding a local patch can lead to ambiguity of the digit (7 to 1 or 2 to 7). Such confusion can be seen in the attention map during the binding that the DASBE is always trying to predict the stimulus but not completely correct and thus cannot lead to the coherence (high synchrony score). In MNIST-Shape, overlap may even cause a “shape” to disappear.
>
> 3. **Evaluation**. Interestingly, some overlapped pattern is challenging even for a human observer. In these cases, however, the DASBE might create a novel segmentation of the image, which is quite reasonable(multistability). However, such valuable “intelligence” is not captured in our current evaluation (AMI score) because the AMI compares the pre-defined grouping with a ground truth and therefore lead to a pretty low score.
>
> ## Q4.
>
> We thank the reviewer’s advice and have added in the reference.
>
> ## Q5.
>
> The reviewers seem to concern a lot about this issue. We would like to address this concern from two aspects, highlighting our novelties.
>
> **Qualitative comparison**
>
> Compared with traditional supervised algorithm, DASBE do not need grouping level supervision, thus is consistent with “multistability” property of binding.
>
> Compared with (pre-defined) slot-based ANN models, DASBE do not need such explicit separation and therefore grouping “slots” are self-organized.
>
> Compared with complex-value based models (eg. CAE[2]), which can only represent a small number of objects at a time due to limited range of phase values[2], DASBE can flexibly use time dimension for grouping. This difference is more obvious in Bars dataset, where 12 objects needs to be bound. DASBE seems to use a very flexible way to correlate bars in time dimension (bind each bar to a unique composition of (multiple) timing points), See SI A10 Fig 17,24.
>
> **Quantitative comparison**
>
> The RC-bind model in [1] is one of the most preliminary works on binding in ANNs and can act as a baseline because they also use relatively simple network models and were evaluated by AMI score (but its slot is still pre-defined for each dataset, and background is not considered in evaluation). The comparison is shown in another seperate response.
>
> It can be seen that the overall score is generally competitive to benchmark (RC-bind) and can exceed the benchmark on certain dataset (eg. Bars, partly due to its extendable binding capability). Note that the evaluation based on real-value pattern / pre-defined slots / ignoring background may bias the AMI to a higher value than that based on binary pattern and self-organized slots.
>
> Comparison with CAE based on paper[2](code not available). On shapes dataset, the performance is comparable(ARI=0.98), while on MNIST-Shape dataset, CAE is much better(ARI=0.78). However, CAE uses an essential trick: the MNIST and Shapes is set to have different greyscales (SI A.4 in [2]) and this leaks the information about binding by the amplitude (firing rate). This makes the problem much easier. We guess if given identical greyscale to MNIST and Shapes (as we do), CAE might not work at all.
>
> Besides, in [2] the performance of another state of art model SlotAttention[3] is reported to perform much worse than DASBE on Shapes dataset(nearly chance level)
>
> [1] Klaus Greff, Jürgen Schmidhuber. Binding via reconstruction clustering.2015.
>
> [2] Löwe "Complex-Valued Autoencoders for Object Discovery." 2022.
>
> [3] Locatello “Object-Centric Learning with Slot Attention.” 2020

---

> > ### Comment · Reviewer_1EXz · 2022-08-08
> > **response to authors**
> >
> > I am still not fully convinced by the author's response (I still think the comparison with other networks is weak, and should deserve a more careful analysis in the paper). Even If do think that the idea behind the article is valuable and novel for the ML/neuroscience community, I have the feeling that the current state of the article is not mature enough. I encourage the authors to rearticulate the article such that the differences/similarities with other networks are better highlighted, and better understood.

---

> > > ### Author Response · Authors · 2022-08-09
> > > **Response to reviewer4 (1EXz)**
> > >
> > > We thank the reviewer's acknowledgement of the value and nolvelty behind the article. Meanwhile, we also understand the reviewer's concerns. We are aware that the current work is still in a preliminary stage on the broader issues of combining spiking synchrony / iterative attention mechanism / deep learning framework to solve the visual binding problem and also designing the general methodology to evaluate the binding quality. We point these issues out in the limitation section and will continue to explore these valuable issues in a line of future works.
> > >
> > > On the one hand, the potential advantages of the temporal binding solution has been pointed out in [1] as "natural common format","shared feature set","soft-assignment","meta-stability" and also "bio-plausibility" and are regarded as a promissing extra ingredient for ANNs. However, the core difficulty for combining the both sides is the incompatibility between the deep learning framwork and the traditional temporal binding approach. The aim of this article is to propose such a framework that is conceptually consistent with both sides, so that a line of "hybrid" models could be built. The model demonstrated in the article qualitatively confirms the above desirable properties. The qualitative difference with other approaches (eg. complex-value based approach / slot-based approach) is discussed in the related work in the main text.
> > >
> > > On the other hand, we totally agree that a more thorough analysis and quantitative comparison are also essential. Again, we thank the reviewer's valuable advice and we will continue to explore these important issues in a line of future works, within the framework proposed in this article.
> > >
> > > [1] Klaus Greff, On the binding problem in artificial neural networks 2020.

---

> > > > ### Comment · Reviewer_1EXz · 2022-08-09
> > > > **Response to authors**
> > > >
> > > > Thank for the answer,
> > > >
> > > > I have no doubt about the advantages of temporal bindings in general ! I would just like to see a clear comparison between your way of doing temporal binding and the ways that have been proposed by other article (e.g. complex values, slot-based...). The comparison is only discussed, but is not quantitative. I think this point is crucial because it would allow to identify the interesting mechanisms for including temporal component in ANN. I'll keep my original rating.

---

> > > > > ### Author Response · Authors · 2022-08-09
> > > > > **Thanks you for your response**
> > > > >
> > > > > Thanks for the reviewer's constructive comments. We agree with the reviewer and will work on more elaborative comparative analysis.

---

> ### Author Response · Authors · 2022-08-02
> **Response to reviewer4 (1EXz) part2 (Comparison)**
>
> ## Response for the comparison with benchmark.
>
> The RC-bind model in [1] is one of the preliminary works on binding in ANNs and can act as a baseline because they also use relatively simple network models and were evaluated by AMI score. However, this model and a line of later developed generative binding models all uses an explicitly designed latent structure to seperate the objects (the slots are pre-defined). Besides, the background is not considered in RC-bind[1]. Since the results in the main text considers the background, we add DASBE results where background is ignored in the evaluation (-bg, the results are based on a single random seed due to limited time for running the evaluation). In general, the AMI can have a 1% increase without background.
> Besides, for MNIST-Shape we use a CNN instead of MLP for the DAE and the performance get further increased (from 0.53 to 0.58).
>
> |     AMI score        | RC-bind (-bg) | DASBE-bind (+bg) | DASBE-bind(-bg) |
> | ----------- | ----------------------- | ---------------- | -------- |
> | Bars        | 0.95                    | 0.97             | **0.99**        |
> | Shapes      | 0.93                    | 0.90             | 0.91     |
> | Corners     | 0.85                    | 0.81             |    0.82      |
> | Multi-MNIST | 0.65                    | 0.56             |     0.58     |
> | MNIST-Shape | 0.55                    | 0.58             | **0.59**     |
>
> It can be seen that, the overall AMI score is comparable to the benchmark and can even exceed in certain dataset (eg. Bars). For bars, visualization in SI A.10 Fig17 shows that DASBE has quite special and flexible way of binding (even longer term correlation). As far as we know, such way of binding is not seen in any other models (either slot-based or complex-value based), but is reported to be possible in the brain, eg[2].
>
> [1] Klaus Greff, Jürgen Schmidhuber. Binding via reconstruction clustering.2015
>
> [2] Izhikevich, Eugene M.. “Polychronization: Computation with Spikes.” Neural Computation (2006)

---

### Official Review · Reviewer_QMxx · 2022-07-07

**Rating:** 5
**Confidence:** 3
**Soundness:** 3 good
**Presentation:** 2 fair
**Contribution:** 3 good

**Summary:**

In the paper "Dance of SNN and ANN: Solving binding problem by combining spike timing and reconstructive attention" the authors propose a hybrid architecture consisting of a spiking neural networks and an attention map modulating denoising autoencoder to address the binding problem. They test their approach by encoding input images consisting of binary images and evaluate binding properties of the network. Notably, they observe gamma-like oscillations. Moreover, they prove four propositions pertaining to convergence of the network.

**Questions:**

Line 50: The statement is too strong. Continuous does not necessarily imply representation of uncertainty.

What value was used for V_th?

Algorithm 1, line 2: misspelled Bernoulli.

Fig 2a: Why is the rate score approximately zero? When the units start firing, the rate should increase, shouldn't it?

Fig 2, panels a, b and h: The authors used an extremely tiny font size. It is not legible even when zooming in a bit.

Fig 3: This figure wastes a lot of whitespace and unnecessarily repeats the legend.

Table 1: It should be explained that the values are percentages. The main text describes values between 0 and 1.

**Limitations:**

The authors discuss potential societal impacts in the supporting information. The authors also discuss limitations with respect to missing biological components and competitiveness when compared to segregation methods.

**Strengths And Weaknesses:**

The paper describes an interesting idea to address the binding problem: the combination of a spiking neural network with a denoising autoencoder. To the best of my knowledge, this is a novel approach in the context of the binding problem.

However, the significance of the paper is lacking. The authors compare their approach to a pure-ANN and a pulse-coupled neural network (PCNN) as a pure-SNN, which do not perform well. However, in the PCNN the authors do not tune all hyperparameters (lines 206ff in the supporting information). Indeed, in the Discussion the authors note that their approach does not compete with a dedicated segmentation algorithm. Hence, significance to the machine learning community is clearly limited. On the other hand, with a denoising autoencoder as an ANN and missing biological ingredients such as STDP, relevance to the neuroscience community is limited too.

Regarding the proven propositions, the authors admit that their propositions do not guarantee convergence of their network in practice. Therefore, their propositions seem to be of limited value.

I could not find conceptual or technical mistakes in the main paper. I did not check the proofs in the supporting information.

The clarity of the main text is fine. The figure quality could be improved. The labels are often small and sometimes not legible (see Questions).

EDIT: The authors clarified some misunderstandings and I increased my score from 3 to 5.

---

> ### Author Response · Authors · 2022-08-02
> **Response to reviewer3 (QMxx)**
>
> # Response
>
> We thank the reviewer for assessing our work and describing it as “interesting idea” and “a novel approach”. However, we found several misunderstandings in these comments which may have misled the reviewer’s assessment of our work. We will start with general response to these comments (A~C), and then address the remaining questions.
>
> **A** “However, in the PCNN the authors do not tune all **hyperparameters** (line 206ff SI).”
>
> In the PCNN, we indeed have tuned all hyperparameters. The misunderstanding may be caused by the line 206 SI “The original PCNN has quite large number of hyper-parameters to tune [15, 16] and the binding result can highly dependent on the configurations of these parameters. We reduce the number of parameters based on [17, 18].”. The models in [15,16,17,18] are all variants of PCNNs. Our purpose is to clarify the model that we used for comparison instead of stating that “we do not tune all parameters”. We have rewritten this part to make it more clear(SI A 6.4).
>
> **B** “Indeed, in the Discussion the authors note that their approach does not compete with a dedicated segmentation algorithm. Hence, significance to the **machine learning community** is clearly limited.”
>
> The “dedicated segmentation algorithm” refers to those traditional clustering methods or supervised methods and models that require expert knowledge. Although these models may have higher performance on certain tasks, the design principles are conceptually inconsistent with the core properties of the binding problem (eg. multi-stability) [1]. Combining synchronization with ANNs is a non-trivial and essential issue in machine learning community. For instance, Jürgen Schmidhuber discussed its importance in [1], “…Likely, the brain does not rely on a single mechanism for addressing the binding problem but on a combination of several. In either case, it is clear that temporal synchronization plays an important role in neural information processing, and perhaps one that is still unaddressed in current artificial neural networks….”. To the best of our knowledge, this is the first model that explicitly combines spiking synchrony and ANN, which could be an important step for the machine learning community.
>
> **C** “On the other hand, with a denoising autoencoder as an ANN and missing biological ingredients such as STDP, relevance to the **neuroscience community** is limited too.”
>
> We are convinced that this work is highly relevant to the neuroscience community, which can be elaborated from two aspects. First, deep learning is considered a promising modeling framework for neuroscience, and various comparable properties have been discovered [2]. By combining spiking representation and dynamics with ANNs, synchrony (and relevant mechanisms) can be further modeled and observed in DASBE, which extends the modeling capability of pure-ANN. Furthermore, the DAE is consistent with neuroscience models. Top-down prediction of input stimulus is an important part of the brain. The predictive coding or active inference theory, raised by Rao and Ballard and further developed by Karl Friston, is considered as a canonical model of neocortex[3], which is “autoencoder like”.
>
> Second, this work was inspired by many neuroscience findings. In our model, the biological ingredients include refractoriness, delay in the top-down/bottom-up loop, coincidence detector, stochasticity and synchrony. Although STDP is not directly involved, it is consistent with this model. Gradient-based learning in autoencoder has been shown to be consistent with Hebbian plasticity (Hebbian descent) [4]. STDP is just one of many essential neuronal ingredients, which cannot function alone. Our model can be regard as a pre-wired network that helps maintain the stability and basic performance, to which STDP can be added (in SCS). This work shows that bringing together artificial intelligence and neuroscience promises to yield benefits for both fields.
>
> For concerns of **propositions**, we have updated main text and added experimental supports (SI A.11) to make it clearer.
>
> Back to questions.
>
> ## Q1.
>
> We rewrote this sentence: “the grouping information is usually continuous thus may **help** encode uncertainty about the grouping”.
>
> ## Q2.
>
> Vth=1 (Alg1 and SI table2 updated)
>
> ## Q3.
>
> Corrected in the revised version.
>
> ## Q4.
>
> The firing rate for each neuron is measured in a relatively long interval (30 steps). And it is measured one interval after another. Thus, firing rate is possible to be independent of detailed spike train structure during the binding, which is used for clustering. Due to the independency, rate score can be zero.
>
> ## Q5 & Q6
>
> Figures are reedited.
>
> ## Q7.
>
> Yes, thanks for the advice.
>
> [1] Klaus Greff, On the binding problem in artificial neural networks 2020.
>
> [2]Richards, A deep learning framework for neuroscience. Nat Neurosci 2019.
>
> [3] Friston. Canonical microcircuits for predictive coding. Neuron 2012
>
> [4] Melchior “Hebbian-Descent.” (2019)

---

> > ### Comment · Reviewer_QMxx · 2022-08-06
> > **Thank you for the clarifications**
> >
> > I thank the authors for their clarifications.
> >
> > I misunderstood the hyperparameter tuning procedure and the new phrasing can help to avoid such misunderstandings.
> >
> > Further explanations of the significance to the machine learning and neuroscience communities are helpful. Additional explanations about the propositions are much appreciated too.
> >
> > I increased my rating from 3 to 5 and also increased soundness, presentation and contribution scores.

---

> > > ### Author Response · Authors · 2022-08-09
> > > **Response to reviewer3 (QMxx)**
> > >
> > > Many thanks to the reviewer's appreciation of our clarification and explaination.

---

### Official Review · Reviewer_Ybbh · 2022-07-10

**Rating:** 7
**Confidence:** 4
**Soundness:** 3 good
**Presentation:** 2 fair
**Contribution:** 4 excellent

**Summary:**

This paper proposes a neural network architecture for solving the visual binding problem. The proposed architecture is comprised of a single layer of spiking neurons and a denosing autoencoder. Importantly, the top-down feedback from the output of the autoencoder modulates the spike timing of the spiking neurons. The solution to the binding problem for every visual input is formed through the iterative top-down and bottom-up processing.


**Questions:**

Major comments:

1- It is not clear in the paper how the k-means clustering is applied to the pure-ANN model. For DASBE, as mentioned in the paper, the clustering is applied to the spike trains. Without the spiking output of the first layer, how is the clustering method implemented? Also, for sake of clarity, it would be helpful to include an example result of the pure-ANN-based clustering in the figures. Moreover, it seems that pure-ANN and folded-DAE are used interchangeably throughout the paper referring to the same type of model. I suggest that the authors use a single term to avoid confusion.

2- In the formulation of DASBE, the output of the DAE decoder is used to module spike timing and it's called $\gamma$ in equation (4). In folded-DAE what seems to be the same variable is called X'. If they're referring to the same variable, and there is no difference between the two in the two models, I strongly suggest that the authors use the same notation in the two models to avoid any confusion. If there is any difference in how the output of the DAE decoder is used in the two models, it should be explained.

3- In section 4.2.4, the contrastive loss that is used for training the model is a *supervised* contrastive loss, as explained in the supplementary materials. Given that contrastive losses are mostly associated with self-supervised methods, it'd be more clear if the authors would clearly explain that in the case of this problem, a supervised contrastive loss is used to learn a disentangled representation of object shapes. Also, How does the results compare with the results of using only a reconstruction loss?

Minor comments:

1- How are the number of clusters set in k-means?

2- Figure 2b is too small and the text is not readable. It's not clear what the rows and columns and each square are showing in the zoomed-in views of figure 2c.

3- In the last paragraph of page 6, pulse-coupled neural network (PCNN) is introduced, but it's referred to as pure-SNN in the rest of the paper. I suggest that the authors use only one term for referring to this model to avoid confusion.

4- In line 212, "It is interesting to note that the grouping information is continuous (Fig.2 d.g)" How are the plots in Figures 2 d-g related to what is stated in this sentence?

**Limitations:**

Given that the authors are mostly relying on qualitative comparisons to present the performance of the model, it's important to also include cases where the model doesn't perform very well. Ideally, one would like to see some statistics on the number of cases in each dataset where the model doesn't outperform other approaches or even possibly performs worse. In the absence of such statistics, at least I need to see a comparison between successes and failures of the proposed model.


**Strengths And Weaknesses:**

Strengths: I enjoyed reading this paper and very much liked the proposed idea. A combination of top-down modulation and spike timing is an intriguing idea as a solution to the binding problem. I believe that both neuroscience and AI communities would, in general, benefit from the publication of this paper as a proof of concept for the potential role of spike timing in binding different properties of an object.

Weakness: The paper could benefit a lot from some clarification in the method. Also, some of the figures need further explanation and editing. For example, it's not clear what the rows and columns are representing in the zoomed in views in figure 2c. Moreover, the details in some figures are not readable, e.g. in figure 3. I have a few concerns and comments that I elaborate below.

---

> ### Author Response · Authors · 2022-08-02
> **Response to Reviewer2 (Ybbh)**
>
> # Response
>
> We thank the reviewer’s genuine appreciation of the proposed idea and elaborative advice to improve the paper. We have uppdated the main text and SI to make it more clear.
>
> ## Major 1
>
> (a)  Thank you for your question. We are sorry for not precisely describing how k-means is applied to the folded DAE, although it is very briefly mentioned in line184 of the supplementary material. Here the folded DAE is a reduced DASBE with spiking dynamics (spiking non-linearity, stochastic noise and refractoriness) removed and the “membrane potential”   $\widetilde{x}(t)$ directly fed to the DAE (SI Fig.4). To evaluate binding in **spiking models**, we treat each neuron as a sample and cluster each neuron according to its temporal “activities” as features of the sample. In this way, “shorter distance” between feature patterns can be interpreted as higher similarity between spike timings. **For folded DAE**, although there is no spike train, it still has similar temporal activity patterns or “reduced membrane potential train ($\widetilde{x}(t)$)”. Thus, only a minor change of spike train to “real-value train” is used to cluster the neurons, while all other settings remain the same (SI Fig.4 a.d).
>
> (b)  Thank you for your advice and we have added an example in SI Fig.4.
>
> (c)  We have removed the term “pure-ANN” and unified them as “folded-DAE”
>
> ## Major2
>
> Thanks for your insightful advice. We agree that the output of DAE/ folded-DAE should be noted as $\gamma$, and have corrected it in Fig3 of SI.
>
> ## Major3
>
> We thank the reviewer’s important advice to avoid conceptual confusion. The intention to use contrastive loss is to learn feature neurons invariantly response to certain feature(shapes/position). In principle, such disentanglement might be more implicitly achieved by unsupervised regularization like $\beta$-VAE, or self-supervised contrastive loss, but “supervised” contrastive loss seems to be more efficient. Although the loss is supervised for learning disentangled features, it does not directly guide the grouping of features. We update the main context and provide discussion about it in SI A.8.
>
> For the comparison between the results trained by pure-reconstruction loss and that by augmented contrastive loss. We add it in the SI Fig13,14. The difference is clear: without further constraint (either contrastive loss or regularizations like $\beta$-VAE ), the disentanglement is not visible. Thus, the representation in the hidden layers is less explainable and may not directly comparable to the results in neuroscience (binding of high-level feature neurons). However, if we take the distributed and **“entangled”** pattern still as high-level features (but represented more complexly), they are still being bound in synchronous timings. Thus, the contrastive loss is mainly for better explanation and visualization of latent layers (visible high-level feature neurons) instead of a necessary requirement of hierarchical binding itself.
>
> ## Minor 1
>
> The number of clusters is set to be the same as the ground truth (updated in line173SI A.6.2), which is K (the number of objects) + 1(background).
>
> ## Minor 2
>
> We are sorry for the unclear figure. Figure 2b compares the synchrony/rate score (y-axis) with clustering score of k-means (x-axis) during the binding process from random firing to convergence. We have reedit the Fig2 in the main text and leave those detailed analysis to SI Fig7
>
> ## Minor 3
>
> We have remove the term “pure-SNN”.
>
> ## Minor 4
>
> We thank the reviewer’s insight question. The continuous grouping information refers that, due to the time-dimension being continuous or having high temporal resolution, the information of each object (in this case, these explainable spikes of feature neurons) is grouped in a **neighborhood** of the time axis, instead of a “single discrete slot”. In other words, the (“continuous”) span of synchronized spikes can reflect the quality/confidence of binding. In figure 2d, it can be seen that neurons related to pixels of one object can fire at **adjacent** time steps (patterns in bound phase). In figure 2g, it can also be seen that the neurons are not locked in a single time step, but have a **span** around 3 timesteps, which is also reflected by the wider shadow (gaussian-shape) behind the spikes. In the revised manuscript, we have added more explanations. This property can be seen more clearly in other examples in SI A.10 (which may get a lower AMI score)
>
> ## Comments in the Limitation
>
> We fully agree with the reviewer. We therefore have provided additional examples of binding results in five datasets of different AMI (from low to high) in SI A.10 (eg Fig18,22,25,29), and discussed the results. The statistics at the dataset level is shown in SI Fig 2. In sum, the model can fail due to more challenging input (special MNIST digit) or very severe overlap (MNIST+Shape/Multi-MNIST). Lastly, we provide a comparison to other works in the response to reviewer4.

---

> > ### Comment · Reviewer_Ybbh · 2022-08-08
> > **Thanks for the response**
> >
> > I thank the authors for their comprehensive response to my questions and concerns. Given the explanations in the authors’ rebuttal and the additional results added to the paper, I’m now more convinced of the solidity of the proposed method. I’ll also increase my score for this paper to 7. As I mentioned in my initial comments, I believe that this paper is an important contribution to both fields of neuroscience and machine learning.

---

> > > ### Author Response · Authors · 2022-08-09
> > > **Thanks for the comments**
> > >
> > > We thank the reviewer for the appreciation of the article and our response. We also thank the reviewer's for the elaborative advices and comments that are very helpful to improve the article.

---

### Official Review · Reviewer_2YkF · 2022-07-11

**Rating:** 6
**Confidence:** 3
**Soundness:** 3 good
**Presentation:** 3 good
**Contribution:** 3 good

**Summary:**

This paper proposed a hybrid neural network of SNN and ANN to solve the binding problem. This hybrid network on the one hand expands the representation space with additional temporal dimension, and on the other hand holds the computational power of ANNs. The effectiveness of the model is evaluated on several artificially generated datasets, showing its availability, though not concrete enough. The limitations of the current paper were well illustrated (in the appendix), however, the proposed model could be a significant step towards a broader work.

**Questions:**

The current model only deals with the binary images, which is quite compatible with the binary spike in the spike coding space (SCS), when this input extends to real-valued pixels, how would it influence the performance of this model, in particular, how would this influence the SCS (I guess you will then need an additional coding mechanism which will then violate the one-to-one correspondence described in line 107)?

Based on the paper, the SNNs provide an additional temporal domain by inducing the neuron dynamics, what is the function of the “spikes” in this model, i.e., what if we replace the spiking neuron models with real-valued dynamic neuron models, which still provide the temporal dimension but then remove the binary coding mechanism?


**Limitations:**

please see above

**Strengths And Weaknesses:**

Strengths: The idea of combing SNN and ANN in binding problem is quite novel. The theory (although with idealized conditions) is built very solidly. The experiments show the advances of this combined model compared to pure ANN or SNN.

Weakness: As mentioned above, the complexity of the implemented experiments are not significant enough.

---

> ### Author Response · Authors · 2022-08-02
> **Response to Reviewer1 (2YkF)**
>
> # Response
> We thank the reviewer's appreciation and comments.
>
> ## Q1
>
> This problem can be addressed in a way inspired by the brain processing the rich information of the real-world. To be specific, each pixel is not topologically mapped into a single neuron, but rather a “column” of a **neuronal population**. Therefore, RGB values can be encoded by a binary population code. For example, in each column, an “array” of neurons has receptive fields for different colors (RGB values) of the “corresponding pixel”. More specifically, we can divide (quantize) RGB values into finite regions and utilize a one-hot encoding strategy. Thus, real values are converted into binary values and the overall mechanism in the DASBE remains. And the one-to-one correspondence (pixel to column) is not violated, which resembles the topographical mapping in the brain. Such strategy has been reported to bridge neural mechanism and machine learning task. For example, a general real-value to binary value encoding strategy was developed to apply the cortical neuron inspired hierarchical temporal memory mechanism to solve anomaly detection in real situations [1,2]. **Thus, a variety of encoding strategies of different complexity can be used to solve this problem without violating the topographical mapping**[2].
>
> ## Q2
>
> As far as we understand, the usefulness of additional temporal dimension is dependent on the concept of temporal correlation (eg. synchrony or polychrony) in the temporal binding theory. Thus, **“synchrony” needs to be identified whether it is a spike or non-spike model**.
>
> In a general perspective of binding in temporal dimension [3], two types of traditional models for binding have been proposed: one based on LEGION (local excitatory and global inhibitory oscillatory neural network) and one based on PCNN (pulse coupled neural network). The former uses real-valued dynamical neuron models, which are described by well-designed continuous dynamical equations to generate “smooth pulses”. The latter uses spiking neurons. With proper configuration of parameters, both models have oscillatory behavior and use timing or phase to bind objects. Here, the real-valued neuron, although not firing spikes, still has “smooth pulses”, which is the premise to identify “synchrony”. Thus, the “use” of temporal dimension and general grouping behavior is not the privilege restricted to certain neuron types, spike or real-valued. However, it is notable that since synchrony is essential in these models (or any binding in time dimension models), we need the **“indicator”** of timing, which is best described by spikes. The “smooth pulse” is just an approximation of such indicator. As a direct consequence, **temporal precision** is limited in real-valued “pulse” models and a limited number of objects can be bound in these models [3]. In other words, spikes can provide binding of a higher resolution. Besides, the synchrony of (binary) spikes is easier to be readout (by downstream coincidence detectors), because the indicators of timing (spike) do not interfere with amplitude information (the amplitude of “pulse”) or time-duration (temporal span of “smooth pulse”).
>
> Without any constraints, we argue that no arbitrary real-valued dynamical system can use temporal dimension as our model does. In these models, “synchrony” is usually not definable at all (arbitrary temporal pattern is unable to function as a timing indicator). Besides correlation, there may be other interesting ways to use temporal dimension for binding, which are worthy for exploration in the future.
>
> In our model, the function of “spike” can be attributed to three basic elements: **threshold, stochasticity and refractoriness**. Threshold and sub-threshold stochasticity provide the neural basis for a **sampling** process based on top-down active attention (like Boltzmann machine or MCMC). In this way, several stable states, such as the local minima in the Boltzmann distribution, might be formed by DAE and found during the sampling. Refractoriness enforces states to **switch** and avoids equilibrium, which is the basis for binding different objects. In brief, the spiking neuron model is indispensable for the dynamical behavior of this model.
>
> [1] Ahmad, Subutai, Alexander Lavin, Scott Purdy and Zuha Agha. “Unsupervised real-time anomaly detection for streaming data.” Neurocomputing 262 (2017): 134-147.
>
> [2] Y. Cui, S. Ahmad, J. Hawkins, The HTM Spatial Pooler: a neocortical algorithm for online sparse distributed coding, bioRxiv, 2016, doi:http://dx.doi.org/10.1101/085035.
>
> [3] DeLiang Wang. The time dimension for scene analysis. IEEE Transactions on Neural Networks, 16(6):1401–1426, 2005

---

> > ### Comment · Reviewer_2YkF · 2022-08-09
> > **thank you for the response**
> >
> > Many thanks to the authors for the response to my questions,  which enhances the novelty of the proposed architecture.
> > On the other hand, I still holds the concern on more general and complex tasks, thus I will keep my original rating.

---

> > > ### Author Response · Authors · 2022-08-09
> > > **Response to reviewer1 (2YkF)**
> > >
> > > We thank the reviewer for appreciating the novelty of the proposed architecture. Meanwhile, we can understand the the reviewer's concerns on more complex tasks.
> > >
> > > As pointed out in [1], the binding problem is solved at different levels. For example, supervised methods can lead to sucessful segementation of a practical scene but fails to have the multistability property (due to labeling), which is an essential feature of binding. Thus, these method solves the binding problem at a "eaiser" level.
> > >
> > > For the "harder" level solutions, the generative-model-based method (take for example) is more promissing, because it is unsupervised, requiring less artificial knowledge, and consistent with multistability. However, the **shared shortcomings** of these approaches is that these methods are mostly tested on artificially generated datasets and have a long way to  go to more complex scenes.
> > >
> > > Thus, it seems that complex task is a common challenge for all "harder" level binding solutions. There seems a trade-off between the "level" and "complexity". The preliminary work on ANN binding problem [2], and several following works also uses simple binary dataset for the evaluation.
> > >
> > > As a preliminary attemp to combine the extra ingredient of spike timing into ANNs, we can understand the concern of more general and complex tasks, since as previous works, we do not provide the evaluation on very complex and general scenes. However, it is also notable that the generality of the proposed architecture lies in its **conceptual consistency** with various concerns in the binding problem (common format, multistability, soft-assignment, dynamical structure, unsupervision, bio-plausibility, etc).
> > >
> > > Besides, the concerns may also come from the "visible" coding space (eg. SCS in this paper), which was also used by previous binding methods [2]. However, the additional hierarchical binding property in the proposed model implies the potential for binding in arbitary **latent space**, which might generalize to more complex scenes.
> > >
> > > [1] Klaus Greff, Jürgen Schmidhuber. On the binding problem in artificial neural networks 2020.
> > >
> > > [2] Klaus Greff, Jürgen Schmidhuber. Binding via reconstruction clustering.2015.

---

### Meta-Review · Area_Chair_8x1k · 2022-08-23

**Recommendation:** Accept
**Confidence:** Less certain

**Metareview:**

This paper presents a solution to the binding problem that uses a combination of spiking and reconstructive attention. The authors show how spike timing can be used to group sensory inputs into objects and then have attention alter the synchrony of firing to bind multiple objects. They demonstrate this on a variety of visual datasets including basic shapes and MNIST.

The reviews for this paper were borderline. The reviewers felt that the paper was novel, technically sound, and and well-written, but they were not completely convinced by the paper, in particular there were worries that the experiments did not provide adequate comparisons to other techniques to make the significance of this approach clear, nor did they test appropriately complex situations. Nonetheless, the authors were able to partially alleviate the reviewers concerns, and the final scores were 7,6,5,5. Given these scores, and the general agreement that the paper was technically sound and novel, an 'accept' decision was made.

**Award:**

No

---

### Decision · Program_Chairs · 2022-09-14

Accept